# Analysis of CDPK1 targets identifies a trafficking adaptor complex that regulates microneme exocytosis in *Toxoplasma*

**Alex W Chan[1,2], Malgorzata Broncel[3], Eden Yifrach[1], Nicole R Haseley[1], Sundeep Chakladar[1], Elena Andree[1], Alice L Herneisen[1,2], Emily Shortt[1], Moritz Treeck[3], Sebastian Lourido[1,2]***

[1]Whitehead Institute for Biomedical Research, Cambridge, United States; [2]Biology Department, Massachusetts Institute of Technology, Cambridge, United States; [3]Signaling in Apicomplexan Parasites Laboratory, The Francis Crick Institute, London, United Kingdom

*For correspondence:
lourido@wi.mit.edu

**Abstract** Apicomplexan parasites use $Ca^{2+}$-regulated exocytosis to secrete essential virulence factors from specialized organelles called micronemes. $Ca^{2+}$-dependent protein kinases (CDPKs) are required for microneme exocytosis; however, the molecular events that regulate trafficking and fusion of micronemes with the plasma membrane remain unresolved. Here, we combine sub-minute resolution phosphoproteomics and bio-orthogonal labeling of kinase substrates in *Toxoplasma gondii* to identify 163 proteins phosphorylated in a CDPK1-dependent manner. In addition to known regulators of secretion, we identify uncharacterized targets with predicted functions across signaling, gene expression, trafficking, metabolism, and ion homeostasis. One of the CDPK1 targets is a putative HOOK activating adaptor. In other eukaryotes, HOOK homologs form the FHF complex with FTS and FHIP to activate dynein-mediated trafficking of endosomes along microtubules. We show the FHF complex is partially conserved in *T. gondii*, consisting of HOOK, an FTS homolog, and two parasite-specific proteins (TGGT1_306920 and TGGT1_316650). CDPK1 kinase activity and HOOK are required for the rapid apical trafficking of micronemes as parasites initiate motility. Moreover, parasites lacking HOOK or FTS display impaired microneme protein secretion, leading to a block in the invasion of host cells. Taken together, our work provides a comprehensive catalog of CDPK1 targets and reveals how vesicular trafficking has been tuned to support a parasitic lifestyle.

## eLife assessment

This **important** study in the protozoan parasite *Toxoplasma gondii* significantly advances our understanding of calcium signaling mediated by the kinase CDPK1 in this species. The authors' conclusions are supported by **convincing** evidence, with rigorous biochemical experiments and microscopy analysis. The work will be of broad interest to researchers in the fields of signal transduction and protozoan biology.

## Introduction

$Ca^{2+}$-regulated exocytosis is ubiquitous among eukaryotes. This signaling paradigm regulates an array of processes such as neurotransmitter release in neurons, hormone secretion in endocrine cells, and histamine secretion in mast cells (*Pang and Südhof, 2010*). $Ca^{2+}$-regulated exocytosis is also

critical for apicomplexan parasites that are the causative agents of rampant, life-threatening diseases, including malaria, toxoplasmosis, and cryptosporidiosis (*Havelaar et al., 2015*). Central to their pathogenesis is their ability to transition from intracellular replicative stages to extracellular motile stages, which involves a unique form of cellular movement called gliding, egress from the primary host cell, and invasion into a new host. These processes are driven by the $Ca^{2+}$-regulated exocytosis of apicomplexan-specific membrane-bound organelles called micronemes and rhoptries.

The sequential exocytosis of micronemes and rhoptries is required to promote extracellular motile stages of the parasite (*Blader et al., 2016*; *Carruthers and Sibley, 1997*). Micronemes are localized to the parasite apex and their positioning is dependent on cortical microtubules, ultrastable filaments that polymerize down the length of the parasite (*Chen et al., 2015*; *Leung et al., 2017*; *Wang et al., 2021*). Exocytosis of microneme cargo enables host cell rupture by releasing perforin-like proteins during egress and translocation of exposed adhesins required for gliding and attachment to new host cells (*Carruthers and Sibley, 1997*; *Carruthers and Tomley, 2008*; *Kafsack et al., 2009*). Multiple microneme proteins, including the associated cysteine repeat modular proteins (CRMP) complex, are also required to trigger the exocytosis of rhoptries upon host cell contact (*Kessler et al., 2008*; *Sidik et al., 2023*; *Singer et al., 2023*; *Sparvoli et al., 2022*). Rhoptry proteins include effectors that modulate host responses and transmembrane proteins that are embedded into the host plasma membrane to enable active invasion (*Bradley and Sibley, 2007*; *Lamarque et al., 2011*; *Lamarque et al., 2014*; *Ong et al., 2010*; *Tyler et al., 2011*). Intracellular $Ca^{2+}$ release is necessary and sufficient to trigger the rapid trafficking and exocytosis of micronemes (*Carruthers et al., 1999a*; *Carruthers et al., 1999b*; *Endo et al., 1982*; *Sidik et al., 2016b*). While $Ca^{2+}$ is also necessary for rhoptry discharge, discharge relies on additional cellular processes such as microneme exocytosis (*Coleman et al., 2018*; *Segev-Zarko et al., 2022*). While the exocytosis of micronemes and rhoptries is known to be critical for parasite motility, the mechanisms linking $Ca^{2+}$ signaling to their trafficking and fusion to the plasma membrane are still unclear.

$Ca^{2+}$ signals in apicomplexans are primarily transduced by $Ca^{2+}$-dependent protein kinases (CDPKs) (*Billker et al., 2009*; *Farrell et al., 2012*; *Garrison et al., 2012*; *Kumar et al., 2017*; *Lourido et al., 2012*; *Lourido et al., 2010*; *Lourido and Moreno, 2015*; *Luo et al., 2005*; *Márquez-Nogueras et al., 2021*; *McCoy et al., 2012*; *Sebastian et al., 2012*). CDPKs are serine/threonine protein kinases that are unique to apicomplexans and plants. CDPKs are activated by directly binding to $Ca^{2+}$, in contrast to $Ca^{2+}$/calmodulin-dependent protein kinases (CaMKs) found in animals, which are indirectly regulated by $Ca^{2+}$-bound calmodulin (CaM) (*Ojo et al., 2010*; *Wernimont et al., 2010*). CDPKs are crucial for apicomplexan infection—yet are absent from mammals—making them attractive drug targets; however, their mechanisms of action are still not well understood at a cellular and molecular level. In *Toxoplasma gondii*, $Ca^{2+}$-dependent protein kinase 1 (CDPK1) is required for the $Ca^{2+}$-regulated exocytosis of micronemes, impacting all steps of parasite motility, including egress, gliding, and invasion (*Lourido et al., 2010*). Small-molecule competitive inhibitors of CDPK1 have been identified and have shown some activity against *T. gondii* (*Doggett et al., 2014*; *Johnson et al., 2012*; *Lourido et al., 2013b*; *Lourido et al., 2010*; *Winzer et al., 2015*). Identifying the signaling pathways regulated by CDPK1 could reveal the pathways controlling microneme and rhoptry exocytosis.

The $Ca^{2+}$ that activates CDPK1 and other cellular processes is released from intracellular stores following cyclic nucleotide–mediated activation of protein kinase G (PKG) (*Bisio and Soldati-Favre, 2019*; *Brown et al., 2017*; *Sidik et al., 2016a*). This process can be artificially induced by treating parasites with cGMP specific phosphodiesterase (PDE) inhibitors zaprinast or BIPPO that indirectly activate PKG (*Figure 1A*; *Lourido et al., 2012*; *Nofal et al., 2022*; *Sidik et al., 2016a*; *Yuasa et al., 2005*). PDE inhibition activates PKG within seconds and triggers the $Ca^{2+}$ and lipid signaling nodes controlling parasite motility, including the exocytosis of micronemes (*Figure 1A*; *Lourido et al., 2012*; *Yuasa et al., 2005*). The identity of the CDPK1 substrates that contribute to the regulation of microneme exocytosis remains unknown. Determining CDPK1-dependent phosphorylation is challenging because the signaling pathways controlling parasite motility are rapid and integrate signals from multiple kinases.

Global phosphoproteomic studies of apicomplexans found that $Ca^{2+}$-dependent phosphorylation encapsulated proteins involved in signal transduction, motility, exocytosis, and the cytoskeleton, as well as those with unknown functions; however, the functional relevance and organization of these proteins—especially CDPK1—in regulating exocytosis remain unclear (*Herneisen et al., 2022*;

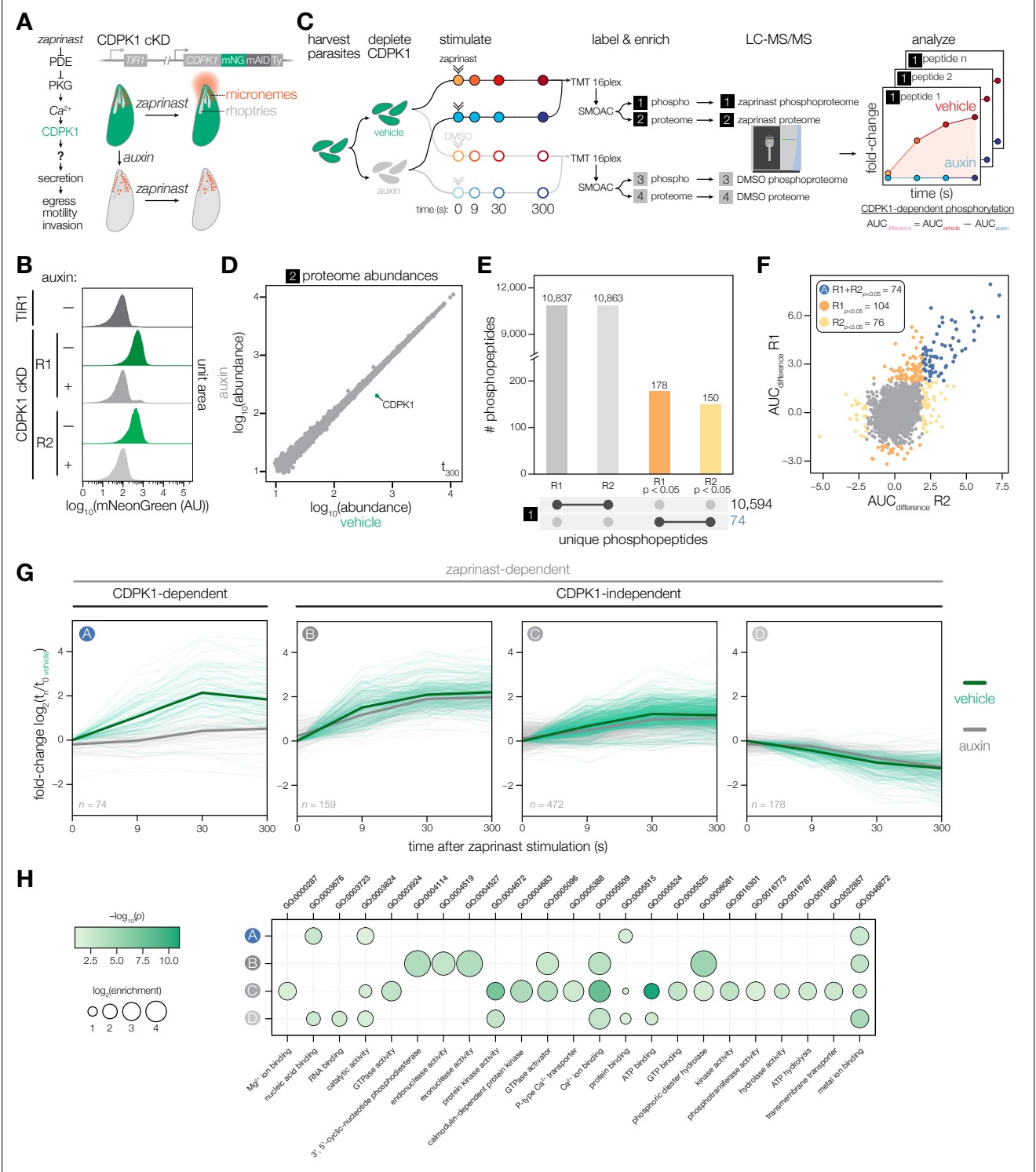

**Figure 1.** Identifying CDPK1-dependent phosphorylation with sub-minute resolution. (**A**) Stimulating parasites with zaprinast triggers $Ca^{2+}$-mediated activation of CDPK1, resulting in the secretion of microneme proteins (red) required for motile stages of the parasite. Conditional knockdown (cKD) of CDPK1 endogenously tagged with mNeonGreen-mAID-Ty (green) after auxin treatment. (**B**) Flow cytometry of mNeonGreen (mNG) fluorescence in extracellular CDPK1 cKD or parental TIR1 parasites treated with vehicle or auxin for 3.5 hr. (**C**) Schematic of phosphoproteomic time course. Parasites were harvested prior to CDPK1 depletion with auxin for 3.5 hs, followed by stimulation with zaprinast or vehicle (DMSO). Samples were collected at 0, 9, 30, and 300 s. The experiment was performed in biological replicates. Samples were labeled with TMTpro, pooled for analysis, and phosphopeptides were enriched using sequential metal-oxide affinity chromatography (SMOAC) prior to LC-MS/MS. Four sets of samples were generated: enriched phosphoproteomes for zaprinast [1] and DMSO [3], and proteomes for zaprinast [2] and DMSO [4]. Mock reporter ion intensities enabling relative

*Figure 1 continued on next page*

*Figure 1 continued*

quantification for a given peptide are shown to illustrate fold-change of unique phosphopeptide abundances during zaprinast stimulation. CDPK1-dependent phosphorylation is determined by calculating the area under the curve (AUC) difference between vehicle and auxin treatment conditions. (**D**) Protein abundances in the zaprinast proteome set [2] at 300 s comparing vehicle- and auxin-treated CDPK1 cKD parasites. (**E**) UpSet plot for the number of phosphopeptides identified in the enriched zaprinast phosphoproteome [1] across individual replicates. Phosphopeptides exhibiting CDPK1-dependent phosphorylation with p<0.05 are indicated. (**F**) Scatter plot of $AUC_{difference}$ values of enriched zaprinast phosphopeptides [1] across biological replicates. Significance was determined by comparing the distribution of $AUC_{difference}$ values from zaprinast phosphopeptides [1] to a null distribution of DMSO phosphopeptides [3] for individual replicates. (**G**) CDPK1-dependent and zaprinast-dependent phosphopeptide abundances over time. Ratios of zaprinast-treated samples relative to the vehicle-treated (no auxin) $t = 0$ samples. Median ratios of a group (solid lines). Individual phosphopeptides (opaque lines). CDPK1-dependent phosphopeptides (Group A) determined as described in (**F**). Zaprinast-dependent phosphopeptides (Groups B–D) were determined by comparing the distribution of $AUC_{vehicle}$ values from zaprinast phosphopeptides [1] to a null distribution of DMSO phosphopeptides [3]. Groups were determined by projection-based clustering. (**H**) GO terms enriched among phosphopeptides undergoing a significant change after zaprinast stimulation. Significance was determined using a hypergeometric test.

The online version of this article includes the following figure supplement(s) for figure 1:

**Figure supplement 1.** Zaprinast-dependent phosphoproteome.

*Invergo et al., 2017*; *McCoy et al., 2012*; *Nofal et al., 2022*; *Treeck et al., 2014*). Here, we utilized time-resolved phosphoproteomics and chemical genetics to identify 163 proteins phosphorylated by CDPK1. Our comprehensive analysis of CDPK1 not only identified phosphoregulation of known factors involved in parasite motility, but also revealed new regulators of exocytosis. We identified a conserved HOOK complex that is phosphorylated by CDPK1 and is required for microneme exocytosis by mediating the rapid trafficking of micronemes during parasite motility.

## Results

### Identifying CDPK1-dependent phosphorylation with sub-minute resolution

We examined the effect of CDPK1 on the phosphoproteome. To control the expression of CDPK1, we endogenously tagged the kinase with a C-terminal auxin-inducible degron (AID) for rapid conditional knockdown upon treating parasites with auxin (*Figure 1A*; *Brown et al., 2018a*; *Shortt et al., 2022*; *Smith et al., 2022*). CDPK1 was robustly depleted from extracellular parasites following auxin treatment for 3.5 hr (*Figure 1B*). We compared vehicle- and auxin-treated parasites at four time-points in the 5 min following zaprinast stimulation (0, 9, 30, and 300 s) to capture the earliest changes following $Ca^{2+}$ flux (*Figure 1C*). We used TMTpro labeling to multiplex 16 samples enabling analysis of a complete time course comparing vehicle- and auxin-treated parasites in biological duplicate in a single LC-MS/MS experiment (*Li et al., 2020*). Samples treated with DMSO instead of zaprinast served as a control. We enriched for phosphopeptides using sequential metal-oxide affinity chromatography (SMOAC) (*Tsai et al., 2014*). In total, we generated four datasets: an enriched zaprinast phosphoproteome [1], a zaprinast proteome (unenriched) [2], an enriched DMSO phosphoproteome [3], and a DMSO proteome (unenriched) [4]. Of the 4255 parasite proteins quantified in the zaprinast proteome by LC-MS/MS, CDPK1 was the only protein with a greater than two-fold depletion in parasites treated with auxin (*Figure 1D*). The remainder of the proteome was largely stable (median $\log_2$-fold change = –0.002 ± 0.089 M.A.D).

We first quantified peptide abundances across time relative to the 0 s vehicle-treated time point to identify CDPK1-dependent phosphorylation, which yielded kinetic profiles of individual phosphopeptides in vehicle and auxin conditions (*Figure 1C*). The zaprinast phosphoproteome included 2570 phosphorylated proteins, represented by 10,594 unique phosphopeptides quantified across both biological replicates (*Figure 1E*). We calculated the area under the curve (AUC) of individual phosphopeptide profiles for vehicle ($AUC_{vehicle}$) and auxin ($AUC_{auxin}$) conditions. To identify the subset of phosphopeptides exhibiting CDPK1-dependent phosphorylation, we calculated the difference between AUC values ($AUC_{difference}$) by comparing the distribution of $AUC_{difference}$ values in the zaprinast phosphoproteome to a null distribution derived from the DMSO phosphoproteome (*Figure 1—figure supplement 1*). This approach allowed us to account for the variability among AUCs under conditions that did not exhibit dynamic changes. We identified 74 unique CDPK1-dependent phosphopeptides across both biological replicates (Group A), belonging to 69 proteins (*Figure 1E and F*). Additionally,

we identified peptides that were zaprinast responsive—despite being CDPK1 independent—by comparing the distribution of AUC$_{vehicle}$ values in the zaprinast phosphoproteome to a null distribution derived from the DMSO phosphoproteome, resulting in 809 unique phosphopeptides representing 501 proteins. We utilized projection-based clustering to sort kinetic profiles of CDPK1-independent peptides into three groups, in which Groups B and C contained phosphopeptides that increased in abundance and Group D contained phosphopeptides that decreased in abundance (*Thrun and Ultsch, 2021*; *Figure 1G*, *Figure 1—figure supplement 1*; *Thrun and Ultsch, 2021*). Significant changes at the peptide level are attributed to altering levels of phosphorylation as opposed to protein levels. Of the 543 proteins exhibiting zaprinast-dependent phosphorylation, 484 were quantified at the proteome level. When comparing the 300–0 s time points, zaprinast-dependent proteins displayed no significant changes (median log$_2$-fold change = 0 ± 0.074 M.A.D). This was consistent with the rapid time scale of these experiments and the overall stability of the 4255 proteins quantified in the proteome (median log$_2$-fold change = 0.01 ± 0.089 M.A.D).

Gene ontology analysis identified several classes of genes that may be relevant to the regulation of zaprinast responses within the different groups (*Figure 1H*). CDPK1-independent phosphorylation and dephosphorylation was prevalent amongst proteins regulating cyclic nucleotide signaling (ACβ, GC, UGO, PDE1, PDE2, PDE7, PDE9, PDE10, and PKG), lipid signaling (PI4K, PI4P5K, PI-PLC, DGK1, and PAP1), and Ca$^{2+}$ signaling (CDPK2A and CDPK3). CDPK1-independent phosphorylation was also observed on downstream proteins regulating microneme exocytosis (APH and DOC2.1), rhoptry exocytosis (ARO, AIP, PL3, and NdP2), gliding motility (AKMT and MyoA), and homeostasis/biogenesis (NHE3, VHA1, NST2, and ERK7) (*Bullen et al., 2019*; *Lourido and Moreno, 2015*). MyoA phosphorylation was observed at S20 and S21, which was previously shown to be phosphorylated by CDPK3 and required for gliding motility (*Gaji et al., 2015*). SCE1 phosphorylation was observed at S225, which was previously shown to be phosphorylated by CDPK3 and required to relieve inhibition during Ca$^{2+}$-stimulated egress (*McCoy et al., 2017*). Our analysis highlights temporally resolved changes in the phosphoproteome in response to zaprinast stimulation. We identify a subset of proteins phosphorylated by CDPK1 that likely represent factors involved in motility-related exocytosis, which we discuss in further detail below.

## Myristoylation modulates CDPK1 activity and alters its interacting partners

CDPK1 was found to be myristoylated on Gly2 (*Broncel et al., 2020*). As myristoylation results in lipid modifications on proteins and can impact membrane targeting and protein–protein interactions (*Martin et al., 2011*; *Wright et al., 2014*), we validated myristoylation of CDPK1 using myristoylation-dependent pulldowns followed by immunoblot detection and MS analysis (*Figure 2—figure supplement 1*). To investigate the role of myristoylation on CDPK1 function, an inducible knock-down strain (iKD) was generated by introducing a mAID-Myc tag at its C terminus (*Figure 2—figure supplement 1*). Auxin-dependent depletion was confirmed using immunoblot (*Figure 2—figure supplement 1*). Depletion of CDPK1 abolished ionophore-induced egress, as expected from previous results (*Figure 2—figure supplement 1*; *Lourido et al., 2010*). Next, we complemented the iKD parasites by introducing HA-tagged WT (cWT) or myristoylation defective (cMut, G2A) copies of *CDPK1* into the *UPRT* locus (*Figure 2A*). We verified correct integration of both complementation constructs (*Figure 2—figure supplement 1*) and confirmed their equivalent and constitutive expression, as well as the auxin sensitivity of the endogenous mAID-tagged CDPK1 (*Figure 2B*). Additionally, we compared endogenous CDPK1 expression to mAID-tagged, cWT, and cMut strains (*Figure 2—figure supplement 1*). Appending an mAID tag on CDPK1 led to a modest reduction in CDPK1 levels, but these levels were equivalent to complementation constructs in cWT and cMut parasites. As predicted, the cMut allele was not myristoylated, based on acylation-dependent pulldowns and immunoblotting (*Figure 2C*).

Given that myristoylation is frequently reported to facilitate membrane association, we examined the localization of cWT and cMut CDPK1 by immunofluorescence (*Figure 2D*). No clear differences were detected between the punctate cytosolic patterns of cWT and cMut. We therefore explored the possible effects of myristoylation on the subcellular fractionation of CDPK1, which we resolved using differential centrifugation (*Figure 2E*). First, we evaluated the fractionation pattern of the endogenous, myristoylated CDPK1. YFP-expressing parasites were metabolically labeled with Myr or YnMyr

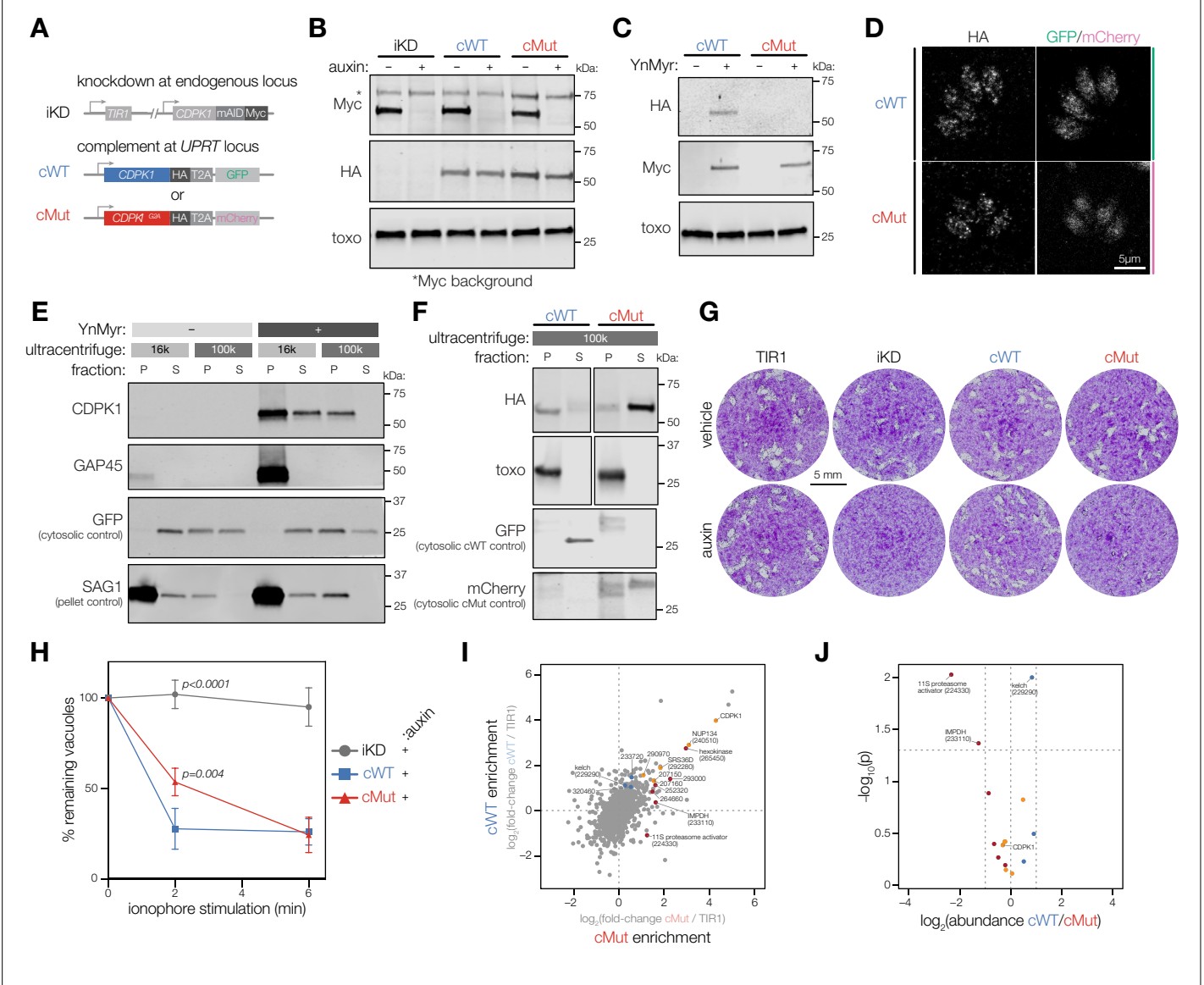

**Figure 2.** Myristoylation modulates CDPK1 activity and alters its interacting partners. (**A**) Complementation strategy used to evaluate the functional importance of CDPK1 myristoylation. See *Figure 2—figure supplement 1* for the construction of the inducible knock-down (iKD) (iKD) line. (**B**) Immunoblot demonstrating the auxin-dependent depletion of endogenous CDPK1 in the iKD, cWT, and cMut parasites (Myc) as well as equivalent expression of the complements (HA). *T. gondii* (toxo) antibody was used as a loading control. (**C**) Biochemical validation of complemented lines by YnMyr-dependent pulldown. Enrichment of WT and Mut complements (HA). The inducible endogenous CDPK1 (Myc) and *T. gondii* (toxo) antibody was used as enrichment and loading controls, respectively. (**D**) Localization of the complemented versions of CDPK1 and corresponding cytosolic reporters within cWT (GFP) and cMut (mCherry) by immunofluorescence. (**E**) Myristoylation-dependent subcellular partitioning of CDPK1. Localization of YnMyr-enriched CDPK1 was evaluated using differential centrifugation. The partitioning into pellet [P] and supernatant [S] fractions was detected by immunoblot (CDPK1) and compared to doubly acylated GAP45. GFP and SAG1 were used as S and P controls, respectively. As only half of the supernatant fraction was removed from the high-speed pellet (100,000 × *g*), the GFP signal is present in the latter. (**F**) Partitioning of complemented WT and mutant CDPK1 after high-speed centrifugation (HA). *T. gondii* (toxo) antibody was used as a P control, whereas GFP and mCherry were used as S controls for cWT and cMut, respectively. (**G**) Plaque assays demonstrating that myristoylation of CDPK1 is important for the lytic cycle of *T. gondii*. (**H**) Lack of CDPK1 myristoylation delays ionophore-induced egress from host cells. Each data point is an average of n = 3 biological replicates, error bars represent standard deviation. Significance calculated using one-way ANOVA with Tukey's multiple-comparison test. See *Figure 2—figure supplement 1* for vehicle controls. (**I**) Immunoprecipitation-MS (IP-MS) of CDPK1-HA in cWT, cMut, and untagged TIR1 parasites across n = 2 biological replicates. Significantly enriched proteins with at least one unique peptide are highlighted based on the following thresholds: significant enrichment in both cWT and cMut (orange) with $p_{cWT} < 0.05$, $p_{cMut} < 0.05$, cWT $\log_2$ fold-change > 1, cMut $\log_2$ fold-change > 1, significant enrichment in exclusively cWT (blue) or cMut (red) with similar criteria across both pulldowns; *t*-tests were Benjamini–Hochberg corrected. (**J**) Fold-enrichment comparing cWT and cMut

*Figure 2 continued on next page*

*Figure 2 continued*

pulldowns of significantly enriched proteins from I; *t*-tests were Benjamini–Hochberg corrected.

The online version of this article includes the following source data and figure supplement(s) for figure 2:

**Source data 1.** This file contains source data that was used to generate the blot in *Figure 2B* (Myc, LI-COR).

**Source data 2.** This file contains source data that was used to generate the blot in *Figure 2B* (HA, LI-COR).

**Source data 3.** This file contains source data that was used to generate the blot in *Figure 2B* (toxo, LI-COR).

**Source data 4.** This file contains source data that was used to generate the blot in *Figure 2C* (HA, LI-COR).

**Source data 5.** This file contains source data that was used to generate the blot in *Figure 2C* (Myc, LI-COR).

**Source data 6.** This file contains source data that was used to generate the blot in *Figure 2C* (toxo, LI-COR).

**Source data 7.** This file contains source data that was used to generate the blot in *Figure 2E* (CDPK1, GFP, LI-COR).

**Source data 8.** This file contains source data that was used to generate the blot in *Figure 2E* (GAP45, SAG1, LI-COR).

**Source data 9.** This file contains source data that was used to generate the blot in *Figure 2F* (HA, LI-COR).

**Source data 10.** This file contains source data that was used to generate the blot in *Figure 2F* (toxo, LI-COR).

**Source data 11.** This file contains source data that was used to generate the blot in *Figure 2F* (GFP, LI-COR).

**Source data 12.** This file contains source data that was used to generate the blot in *Figure 2F* (mCherry, LI-COR).

**Source data 13.** This file contains the annotated source data that were used to generate blots in *Figure 2*.

**Source data 14.** This file contains source data that was used to make the graph presented in *Figure 2H* (GraphPad Prism data).

**Figure supplement 1.** CDPK1 myristoylation, inducible knockdown, and complementation.

**Figure supplement 1—source data 1.** This file contains source data that was used to generate the blot in *Figure 2—figure supplement 1* (CDPK1, LI-COR).

**Figure supplement 1—source data 2.** This file contains source data that was used to generate the blot in *Figure 2—figure supplement 1* (GRA2, LI-COR).

**Figure supplement 1—source data 3.** This file contains source data that was used to generate the blot in *Figure 2—figure supplement 1* (Myc, LI-COR).

**Figure supplement 1—source data 4.** This file contains source data that was used to generate the blot in *Figure 2—figure supplement 1* (toxo, LI-COR).

**Figure supplement 1—source data 5.** This file contains source data that was used to generate the DNA gel in *Figure 2—figure supplement 1* (iKD integration).

**Figure supplement 1—source data 6.** This file contains source data that was used to generate the DNA gel in *Figure 2—figure supplement 1* (cWT, cMut integration).

**Figure supplement 1—source data 7.** This file contains source data that was used to generate the blot in *Figure 2—figure supplement 1* (CDPK1, LI-COR 700).

**Figure supplement 1—source data 8.** This file contains source data that was used to generate the blot in *Figure 2—figure supplement 1* (CDPK1, LI-COR 800).

**Figure supplement 1—source data 9.** This file contains the annotated source data that were used to generate blots in *Figure 2—figure supplement 1*.

**Figure supplement 1—source data 10.** This file contains source data that was used to make the graph presented in *Figure 2—figure supplement 1E* (GraphPad Prism data).

**Figure supplement 1—source data 11.** This file contains source data that was used to make the graph presented in *Figure 2—figure supplement 1G* (GraphPad Prism data).

and lysed in a hypotonic buffer to preserve intact membrane structures. Next, lysates were fractionated to generate a low-speed pellet and supernatant at 16,000 × *g*. The low-speed supernatant was fractionated further into a high-speed pellet and supernatant at 100,000 × *g*. Click reaction-based pulldown and immunoblotting were used to resolve myristoylation-dependent partitioning. In contrast to the doubly acylated GAP45, which was present exclusively in the low-speed pellet, myristoylated CDPK1 was observed in the (i) low-speed pellet, (ii) the supernatant, and (iii) the high-speed pellet (*Figure 2E*). This suggests a potential association with membranous structures or higher molecular weight complexes. We next used both the cWT and cMut lines to elucidate any myristoylation-dependent changes to CDPK1 localization. While cWT could be found predominantly in the high-speed

pellet, loss of myristoylation released cMut into the high-speed supernatant confirming an association with membranous structures or larger protein complexes (*Figure 2F*).

To evaluate the role of CDPK1 myristoylation in parasite fitness, we performed plaque assays comparing the various strains (*Figure 2G*). In the presence of the endogenous copy of CDPK1, both complemented lines developed normally. However, upon auxin-mediated depletion of endogenous CDPK1, cMut plaque size substantially decreased. This finding demonstrates that one or more steps of the *T. gondii* lytic cycle are negatively affected by the loss of CDPK1 myristoylation. In light of CDPK1's known function, we explored whether CDPK1 myristoylation might impact the parasite's ability to egress from host cells. In the absence of auxin, cWT, cMut, and the parental line (iKD) egressed within 2 min of ionophore stimulation (*Figure 2—figure supplement 1*). While cWT parasites maintained similar egress kinetics following auxin treatment, cMut parasites showed a significant delay after 2 min of treatment (*Figure 2H*). This egress delay was overcome by 6 min, suggesting that CDPK1 myristoylation is important for rapid ionophore-induced egress, but not essential.

Finally, we wanted to examine whether myristoylation of CDPK1 affects its interactions with other proteins, which may occur by direct binding to the kinase or by indirect association with a shared membrane structure. We performed immunoprecipitation mass spectrometry (IP-MS) on HA-tagged CDPK1 from cWT, cMut, and untagged TIR1 parasites to control for unspecific binding after hypotonic lysis. CDPK1 was the most significantly enriched protein across both cWT and cMut pulldowns when compared to untagged controls (*Figure 2I*). Fourteen other proteins were enriched at least twofold over untagged controls with an adjusted p-value<0.05 in one or both CDPK1 pulldowns. The greatest enrichment was observed for NUP134 (TGGT1_240510), hexokinase (TGGT1_265450), and SRS36D (TGGT1_292280). Comparing the relative abundance of the 15 proteins enriched in either cWT or cMut pulldowns, we found one protein preferentially associated with cWT: the kelch-repeat containing protein TGGT1_229290 that has been linked to the 19S proteasome activator (*Barylyuk et al., 2020*). Two other proteins preferentially associated with cMut: IMPDH (TGGT1_233110) and a putative subunit of the 11S proteasome activator (TGGT1_224330). While none of the observed associations clearly relate to the function of CDPK1 in parasite motility or microneme discharge, the differential association of cWT and cMut with different proteasome activators suggests myristoylation may affect the choice of degradation pathway (*Stadtmueller and Hill, 2011*).

## Identification of direct CDPK1 targets through thiophosphorylation

Our phosphoproteomic time-course experiment identified proteins exhibiting CDPK1-dependent phosphorylation in live parasites, which includes direct and indirect substrates of the kinase. CDPK1 is unusual among apicomplexan and metazoan kinases in that it contains a glycine at its gatekeeper residue, resulting in an expanded ATP-binding pocket that can accommodate bulky ATP analogues. CDPK1 can use bulky analogues like *N*6-furfuryladenosine (kinetin)–5′-*O*-[3-thiotriphosphate] (KTPγS) to thiophosphorylate its substrates in parasite lysates (*Lourido et al., 2013a*). Thiophosphorylated substrates can subsequently be enriched with an iodoacetyl resin (*Blethrow et al., 2008*). Specificity can be assessed by comparison to mutant parasites harboring a G128M gatekeeper mutation in CDPK1 (CDPK1$^M$) that retains kinase activity but prevents it from using KTPγS. Previous studies using lysates identified a small set of six putative CDPK1-dependent substrates; however, these studies lost the target specificity conferred by the subcellular context and lacked the accuracy and sensitivity of quantitative proteomic approaches (*Lourido et al., 2013a*).

We implemented several modifications to the thiophosphorylation procedure that improved identification of CDPK1 targets (*Figure 3A*; *Rothenberg et al., 2016*). First, we used stable isotope labeling of amino acids in cell culture (SILAC) to directly compare thiophosphorylation in WT (CDPK1$^G$) and mutant (CDPK1$^M$) parasites. Second, we used parasites semi-permeabilized with the bacterial toxin aerolysin—as opposed to the preparations of detergent-lysed parasites used in prior methods. Aerolysin forms 3 nm pores in the plasma membrane which permit the diffusion of small molecules but not proteins, enabling us to perform labeling reactions without drastically disrupting the concentration or localization of proteins (*Iacovache et al., 2006*). Additionally, we prevented non-specific extracellular substrate labeling due to premature lysis by treating semi-permeabilized parasites with 1B7, a nanobody that allosterically inhibits CDPK1 but does not enter the cytosol of semi-permeabilized parasites (*Figure 3B*; *Ingram et al., 2015*).

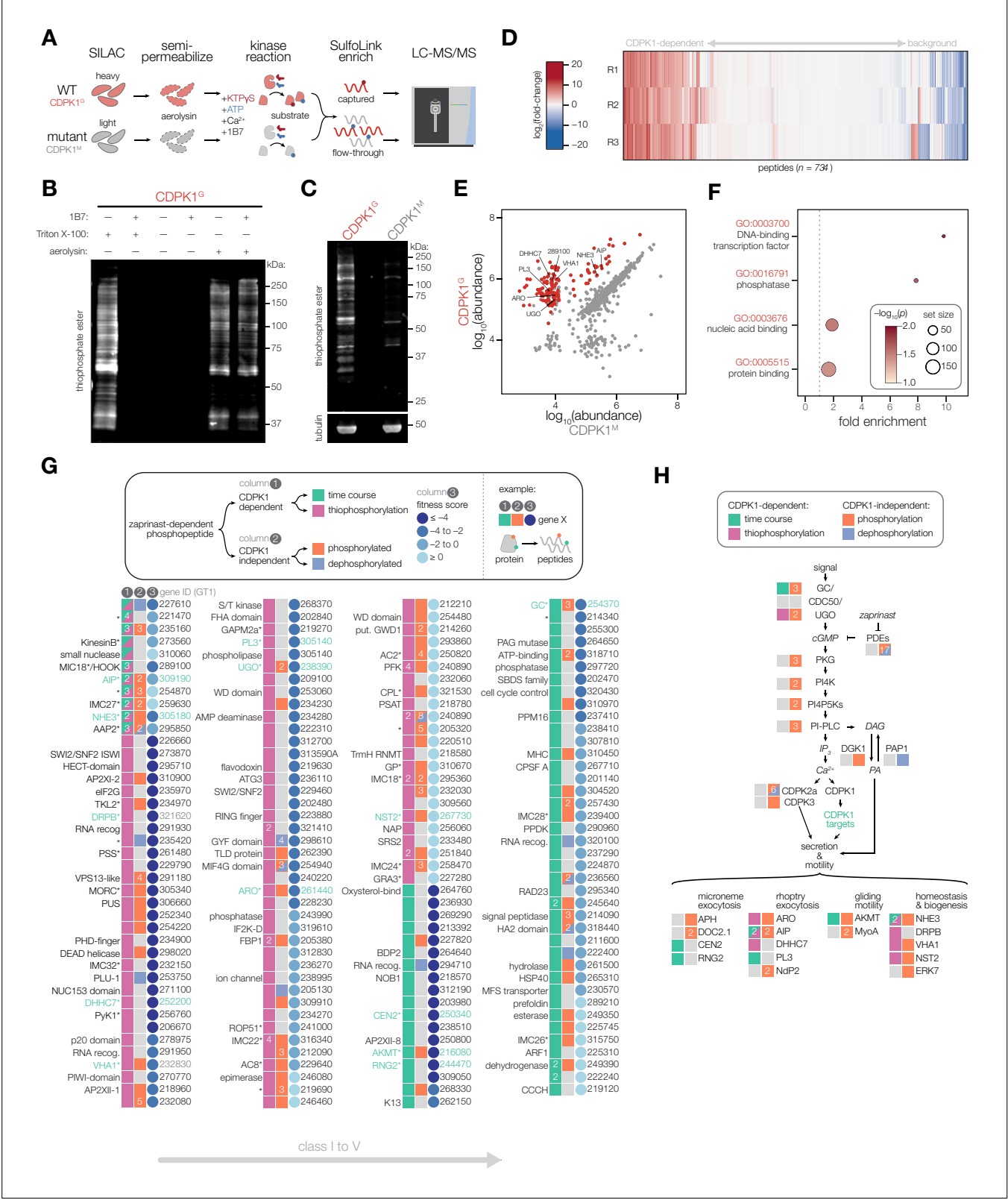

**Figure 3.** Identifying the direct substrates of CDPK1. (**A**) Schematic describing a strategy to identify direct substrates of CDPK1. WT (CDPK1$^G$) and mutant (G128M; CDPK1$^M$) parasites were grown in stable isotope labeling of amino acids in cell culture (SILAC) media for multiplexed quantitation. Extracellular parasites were semi-permeabilized with aerolysin, enabling diffusion of small molecules but not proteins. CDPK1 substrate labeling was initiated by treating semi-permeabilized parasites with Ca²⁺, KTPγS, ATP, and 1B7. While CDPK1 in both WT and mutant parasites can utilize ATP to

*Figure 3 continued on next page*

*Figure 3 continued*

phosphorylate substrates, only WT parasites can use KTPγS to thiophosphorylate substrates. Thiophosphorylated peptides were specifically enriched and the remaining flow-through was saved for whole-proteome analysis. Enriched and whole-proteome samples were analyzed by LC-MS/MS. (**B**) 1B7 nanobody treatment inhibits non-specific extracellular kinase activity of CDPK1. Thiophosphorylated substrates were detected in lysates using an anti-thiophosphate ester antibody immunoblot. Extracellular CDPK1 activity (lane 1) was blocked by 1B7 (lane 2). Aerolysin treatment resulted in intracellular labeling (lane 5) that was unaffected by 1B7 (lane 6). (**C**) Thiophosphorylation performed in aerolysin-treated parasites comparing WT (CDPK1$^G$) and mutant (CDPK1$^M$) strains. Detection was performed as in (**B**). Tubulin was used as a loading control. (**D**) Heatmap quantification of peptides using LC-MS/MS. Fold-change of peptide abundance shown as a ratio of WT (CDPK1$^G$) to mutant (CDPK1$^M$) abundances. Experiment was performed in n = 3 biological replicates. (**E**) Abundances of unique peptides after thiophosphorylation in CDPK1$^G$ and CDPK1$^M$ parasites across n = 3 biological replicates. Significantly enriched phosphorylated peptides are colored in red (-log$_{10}$(*p*)*fold-change >4), one-tailed *t*-test. (**F**) GO terms enriched among significant phosphopeptides from (**E**). Significance was determined using a hypergeometric test. (**G**) Putative targets of CDPK1 determined by sub-minute phosphoproteomics and thiophosphorylation of direct substrates. For a given CDPK1 target gene, the presence of a unique peptide phosphorylated in a CDPK1-dependent manner (column 1) is indicated if identified in the time course (green) and/or thiophosphorylation (magenta). The presence of additional unique phosphorylated peptides exhibiting zaprinast-dependent effects (column 2) is indicated if the peptide was phosphorylated (orange) or dephosphorylated (blue). Numbered boxes indicate multiple unique peptides. Fitness scores (column 3) obtained from genome-wide KO screen data (blues). Lower scores indicate gene is required for lytic stages of the parasite. Gene names (left), TGGT1 gene IDs (right). Gene names with asterisks (*) are associated with published data. (**H**) Signaling diagram describing parasite motility. Proteins exhibiting CDPK1-dependent phosphorylation by either sub-minute phosphoproteomics or thiophosphorylation are indicated (green). Proteins exhibiting CDPK1-independent phosphorylation (red) or dephosphorylation (blue) are indicated.

The online version of this article includes the following source data and figure supplement(s) for figure 3:

**Source data 1.** This file contains source data that was used to generate the blot in *Figure 3* (thiophosphate ester, LI-COR 700 channel).

**Source data 2.** This file contains source data that was used to generate the blot in *Figure 3* (thiophosphate ester, LI-COR 700 channel).

**Source data 3.** This file contains source data that was used to generate the blot in *Figure 3* (TUB1, LI-COR 800 channel).

**Source data 4.** This file contains the annotated source data that were used to generate blots in *Figure 3*.

**Figure supplement 1.** Factors controlling parasite motility.

Thiophosphorylation experiments were performed in biological triplicate comparing CDPK1$^G$ and CDPK1$^M$ parasites. We assessed thiophosphorylation labeling by immunoblot, identifying an array of proteins specifically labeled in CDPK1$^G$, but not CDPK1$^M$ parasites (*Figure 3C*). MS analyses quantified the abundance of peptides in CDPK1$^G$ relative to CDPK1$^M$ parasites, identifying 734 unique peptides across three biological replicates (*Figure 3D*). Samples from CDPK1$^G$ parasites were significantly enriched in phosphorylated peptides, consistent with CDPK1-mediated thiophosphorylation of targets (*Figure 3E*). In total, 123 peptides across 104 proteins were likely direct substrates of CDPK1. GO enrichment analysis did not reveal any pathways relevant to the function of CDPK1 in exocytosis (*Figure 3F*). While our approach largely maintains kinases in their subcellular context, aerolysin treatment disrupts native ion concentrations and may detach the plasma membrane from the inner membrane complex (IMC) (*Wichroski et al., 2002*). We determined that CDPK1 localization remains unaltered upon aerolysin treatment, indicating that subcellular partitioning of CDPK1 is not affected by permeabilization (*Figure 3—figure supplement 1*). We proceeded to consider the thiophosphorylated substrates in the context of the time-resolved phosphoproteomics.

## CDPK1 targets participate in pathways controlling parasite motility

Considering the thiophosphorylation and time-resolved phosphoproteomics, we arrived at a prioritization scheme to identify 163 proteins phosphorylated in a CDPK1-dependent manner (*Figure 3G*). Proteins were divided into five classes based on the overlap of phosphorylated sites between both approaches. Class I contains five proteins for which the same phosphorylated site was identified in both the time-course and thiophosphorylation experiments and includes TGGT1_227610, TGGT1_221470, TGGT1_235160, TGGT1_273560 (KinesinB), and TGGT1_310060. Class II contains four proteins for which phosphorylated sites identified across both approaches were within 50 amino acid residues of one another and includes TGGT1_289100 (MIC18), TGGT1_309190 (AIP), TGGT1_254870, and TGGT1_259630. Class III contains two proteins that were enriched by thiophosphorylation and were also CDPK1-dependent in the time course, but the identified sites were more than 50 residues apart. Class IV contains 93 proteins that were exclusively enriched by thiophosphorylation. Lastly, class V contains 59 proteins that were CDPK1-dependent exclusively in the time course and are likely indirect targets of the kinase. Proteins within each class were further stratified by fitness scores reported from

genome-wide knockout screens, with lower scores representing genes more essential for parasite fitness in cell culture (*Sidik et al., 2016b*). Of the 163 targets of CDPK1, 72 proteins across all classes also displayed changes in zaprinast-dependent phosphorylation at distinct sites that were independent of CDPK1. This overlap suggests that some proteins modified by CDPK1 are also regulated by additional kinases and phosphatases.

We hypothesized that regulators of exocytosis would be found among the targets of CDPK1. Of the 163 protein targets, 38 have been previously localized and/or functionally characterized (indicated with asterisks; *Figure 3G*). Of these 38 proteins, 13 have been implicated in regulating motile stages of the parasite (*Figure 3H*, *Figure 3—figure supplement 1*). Among the candidates associated with microneme exocytosis, centrin 2 (CEN2) and RNG2 have previously been localized to tubulin-based structures in the apical complex of the parasite, and their knockdown is sufficient to inhibit secretion of microneme proteins and block host cell invasion (*Katris et al., 2014*; *Lentini et al., 2019*; *Leung et al., 2019*). Candidates associated with rhoptry exocytosis include the armadillo repeats only protein (ARO), ARO-interacting protein (AIP), palmitoyl acyltransferase DHHC7, and patatin-like phospholipase (PL3) (*Beck et al., 2013*; *Mueller et al., 2016*; *Mueller et al., 2013*; *Wilson et al., 2020*). ARO and DHHC7 both localize to the cytosolic face of rhoptries, where they influence the recruitment of AIP and ACβ. Knockdown of either DHHC7 or ARO disrupts the localization of mature rhoptries, inhibiting the secretion of rhoptry proteins required for invasion. Another potential CDPK1 target, the apical complex lysine methyltransferase (AKMT), rapidly relocalizes from the apical complex to the parasite body during $Ca^{2+}$-regulated motility, and its knockdown impairs the gliding motility of parasites required for invasion and egress (*Heaslip et al., 2011*). Other putative CDPK1 targets participate in homeostasis and biogenesis of secretory organelles, such as the vacuolar type $Na^+/H^+$ exchanger (NHE3), dynamin-related protein B (DrpB), V-ATPase a1 (VHA1), and GDP-fucose transporter (NST2). NHE3-knockout parasites exhibit sensitivity to osmotic shock and dysregulated cytosolic $Ca^{2+}$, resulting in reduced secretion of microneme proteins and an inhibition of invasion (*Francia et al., 2011*). DrpB was previously identified as a CDPK1 target, and its depletion results in severe defects in the biogenesis of micronemes and rhoptries (*Breinich et al., 2009*; *Lourido et al., 2013a*). VHA1 and NST2 participate in the maturation of microneme and rhoptry proteins (*Bandini et al., 2019*; *Stasic et al., 2019*). Lastly, signaling proteins regulating intracellular cGMP levels, such as the guanylate cyclase (GC) and the unique GC organizer (UGO), were also phosphorylated in a CDPK1-dependent manner (*Bisio et al., 2019*; *Brown et al., 2018a*; *Brown and Sibley, 2018b*; *Yang et al., 2019*). CDPK1 acts downstream of GC-mediated production of cGMP, which may suggest regulatory feedback on $Ca^{2+}$ release, as has been suggested for CDPK3 (*Nofal et al., 2022*). These data demonstrate that identification of CDPK1 targets can uncover proteins involved in $Ca^{2+}$-regulated exocytosis.

To identify new factors involved in $Ca^{2+}$-regulated secretion, we prioritized class I and II candidates. Seven candidates in this category lack functional annotation or have been associated with apical structures: TGGT1_227610, TGGT1_221470, TGGT1_235160, TGGT1_254870, KinesinB (TGGT1_273560), a small nuclease (TGGT1_310060), and TGGT1_289100 (MIC18). TGGT1_221470 was previously identified as a CDPK1 substrate after a pulldown from parasite lysates but remained functionally uncharacterized (*Lourido et al., 2013a*). KinesinB localizes to cortical microtubules, TGGT1_254870 localizes to the apical complex, and TGGT1_289100 co-localizes with micronemes, each of which are parasite structures relevant to exocytosis (*Butler et al., 2014*; *Leung et al., 2017*; *Long et al., 2017*). Of these candidates, only TGGT1_227610 and TGGT1_289100 appear to be required for parasite fitness (*Long et al., 2017*; *Sidik et al., 2016b*). Despite prior annotation as a microneme protein, TGGT1_289100 lacks a signal peptide and is localized to the cytosol by spatial proteomics (*Barylyuk et al., 2020*; *Butler et al., 2014*). Instead, TGGT1_289100 is predicted to have an N-terminal Hook domain and extensive coiled-coil domains (*Simm et al., 2015*; *Soding et al., 2005*) with homology to activating adaptors that bind to endosomes and activate super-processive dynein-mediated trafficking towards the minus end of microtubules (*Figure 4A*; *Bielska et al., 2014*; *Guo et al., 2016*; *Zhang et al., 2014*). The predicted polarity of cortical microtubules and their association with micronemes support a model where the vesicles are trafficked by dyneins towards the apical end of the parasite (*Chen et al., 2015*; *Leung et al., 2017*; *Wang et al., 2021*). We observed extensive CDPK1-dependent phosphorylation between the Hook and coiled-coil domains of TGGT1_289100 (S167, S177, and S189–191), consistent with potential regulation of the putative adaptor. These data motivated functional characterization of this factor—henceforth referred to as HOOK.

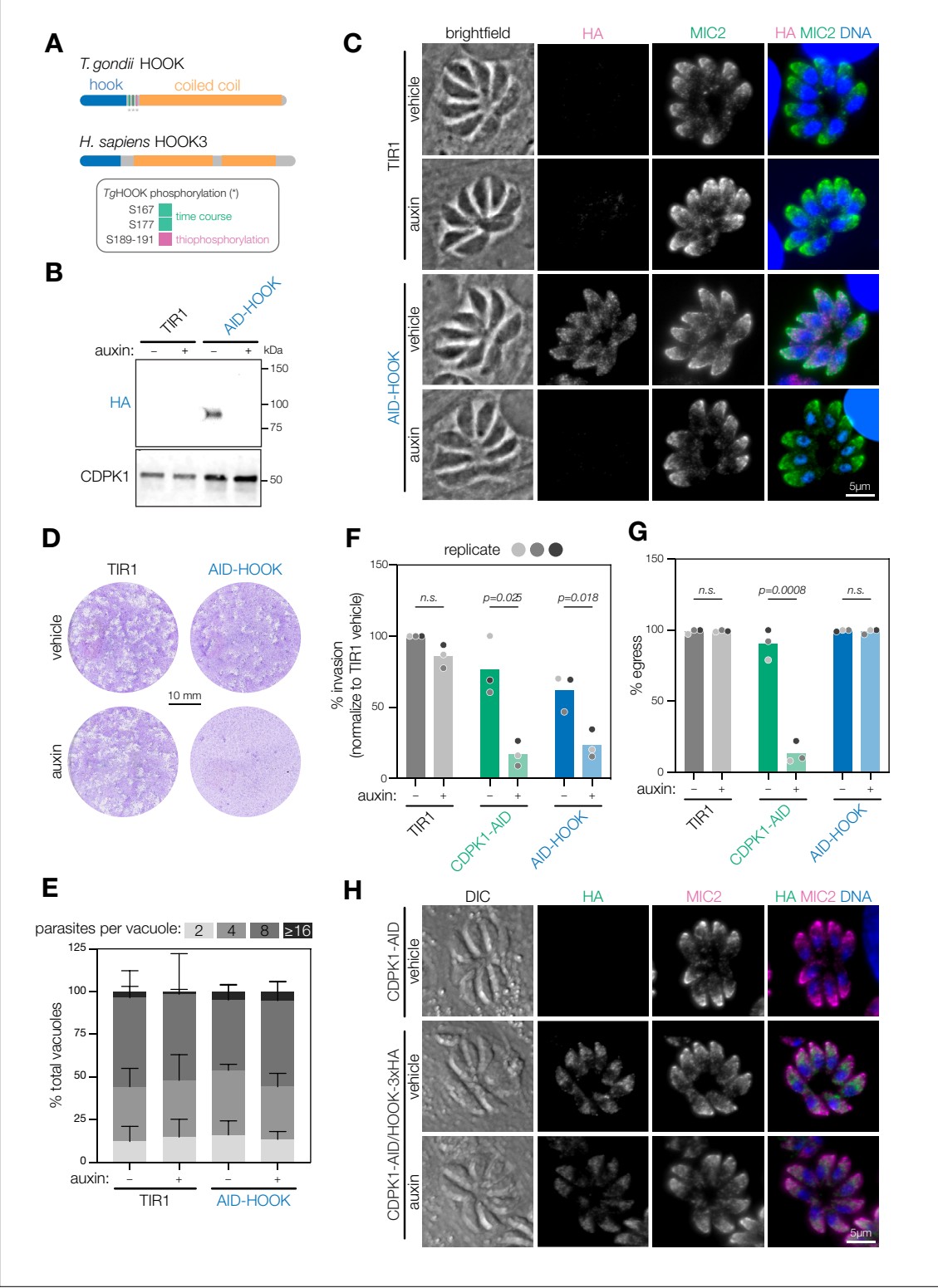

**Figure 4.** HOOK is required for host cell invasion, but dispensable for egress. (**A**) Schematic of *T. gondii* and *H. sapiens* HOOK protein domains. HOOK domain (blue), coiled-coil domain (yellow), sites phosphorylated by CDPK1 (red). (**B**) Immunoblot of HOOK conditional knockdown parasites (AID-HOOK) after auxin treatment for 40 hr compared to untagged TIR1 parasites. CDPK1 was used as a loading control. (**C**) AID-HOOK is visualized in fixed intracellular parasites by immunofluorescence after auxin treatment for 24 hr. Hoechst and MIC2 are used as counterstains. (**D**) Plaque assays of host cells infected with TIR1 or AID-HOOK parasites for 8 d in auxin. Host cells are stained with crystal violet. (**E**) Replication assays of host cells infected with

*Figure 4 continued on next page*

*Figure 4 continued*

TIR1 or AID-HOOK parasites in auxin for 24 hr. Parasites per vacuole were quantified from immunofluorescence on fixed intracellular parasites. p>0.9. Two-way ANOVA. (**F**) Invasion assays of untagged TIR1, CDPK1-AID, and AID-HOOK parasites treated auxin for 40 hr. Medians are plotted for n = 3 biological replicates (different shades of gray); n.s., p>0.05, Welch's *t*-test. (**G**) Parasite egress stimulated with zaprinast following treatment with auxin for 24 hr. Egress was monitored by live microscopy. Percent egress plotted for n = 3 biological replicates, n.s., p>0.05, Welch's *t*-test. (**H**) HOOK tagged with a C-terminal 3xHA in CDPK1 cKD parasites (CDPK1-AID) visualized in fixed intracellular parasites by immunofluorescence as in (**D**).

The online version of this article includes the following source data and figure supplement(s) for figure 4:

**Source data 1.** This file contains source data that was used to generate the blot in *Figure 4* (HA-AID-HOOK, chemiluminescence).

**Source data 2.** This file contains source data that was used to generate the blot in *Figure 4* (HA-AID-HOOK, colorimetric, ladder).

**Source data 3.** This file contains source data that was used to generate the blot in *Figure 4* ( CDPK1, LI-COR 700 channel).

**Source data 4.** This file contains the annotated source data that were used to generate blots in *Figure 4*.

**Source data 5.** This file contains source data that was used to make the graph presented in *Figure 4*.

**Source data 6.** This file contains source data that was used to make the graph presented in *Figure 4F* (GraphPad Prism data).

**Source data 7.** This file contains source data that was used to make the graph presented in *Figure 4* and *Figure 5*.

**Figure supplement 1.** Extended analysis of HOOK knockdown.

**Figure supplement 1—source data 1.** This file contains source data that was used to generate the DNA gel in *Figure 4—figure supplement 1* (AID-HOOK integration).

**Figure supplement 1—source data 2.** This file contains the annotated source data that were used to generate blots in *Figure 4—figure supplement 1*.

## A Hook domain protein phosphorylated by CDPK1 regulates parasite invasion

To study the role of HOOK during the acute stages of the parasite, we generated a conditional knockdown by fusing an HA-mAID tag to its N terminus (AID-HOOK; *Figure 4—figure supplement 1*). HOOK was depleted from parasites following 24–40 hr of auxin treatment, as determined by immunoblotting or immunofluorescence microscopy (*Figure 4B and C*). In contrast to previous observations that localized HOOK to micronemes, we observed only partial co-localization with micronemes with a majority of HOOK localized to the cytosol (*Figure 4C*). Plaque formation was impaired when AID-HOOK parasites were grown in the presence of auxin, consistent with the strong effect on parasite fitness reported from genome-wide knockout screens (*Figure 4D*; *Sidik et al., 2016b*). These results indicate that HOOK is required during the acute stages of *T. gondii*. Parasite replication was not affected following 24 hr of auxin treatment (*Figure 4E*). Microneme and rhoptry biogenesis were also unaffected following 24 hr of auxin treatment by immunofluorescence analysis (*Figure 4C*, *Figure 4—figure supplement 1*). These data suggest HOOK functions during stages associated with parasite motility, such as parasite invasion and egress. Parasites depleted of HOOK displayed reduced invasion efficiency, consistent with that of CDPK1-depleted parasites (*Figure 4F*). We noted that even in the absence of auxin treatment AID-HOOK parasites required longer incubations to reach comparable levels of invasion to wildtype, suggesting that manipulation of the N terminus partially affects HOOK function.

HOOK depletion only partially recapitulates the effects of CDPK1 loss. Parasites depleted of CDPK1 were unable to egress—as documented previously (*Lourido et al., 2012*; *Lourido et al., 2010*)—whereas depletion of HOOK had no effect on parasite egress (*Figure 4G*). Finally, we also examined whether the stability of CDPK1 affected HOOK expression by tagging the C terminus of HOOK with a 3xHA tag in CDPK1 cKD parasites (HOOK-3xHA). HOOK localization and abundance was unaffected by depleting parasites of CDPK1 for 24 hr as determined by immunofluorescence (*Figure 4H*). Taken together, our results indicate that HOOK is required for invasion of host cells, but is dispensable for egress.

## HOOK forms a complex that regulates parasite invasion

Opisthokont HOOK proteins function in complexes to activate dynein-mediated trafficking of vesicular cargo along microtubules (*Bielska et al., 2014*; *Christensen et al., 2021*; *Gillingham et al., 2014*; *Guo et al., 2016*; *Xu et al., 2008*; *Yao et al., 2014*). In *Drosophila melanogaster* and mammals, HOOK proteins form dimers and bind Fused Toes (FTS) and FTS and HOOK-interacting protein (FHIP) via a C-terminal region that interacts with vesicular cargo (*Christensen et al., 2021*; *Krämer and*

*Phistry, 1996*; *Lee et al., 2018*; *Xu et al., 2008*). To identify analogous binding partners of HOOK in *T. gondii*, we performed IP-MS on lysates from HOOK-3xHA parasites and an untagged control (*Figure 5—figure supplement 1*). Three proteins showed comparable enrichment to HOOK-3xHA: TGGT1_264050, TGGT1_316650, and TGGT1_306920. Like FTS, TGGT1_264050 bears homology to E2 ubiquitin-conjugating enzymes, but lacks the catalytic cysteine required for enzymatic activity at position 162. Because of these shared features and their homology, we designated TGGT1_264050 as FTS.

To confirm that FTS forms a complex with HOOK, we investigated its localization and performed reciprocal IPs. We fused a 3xHA tag to the C terminus of FTS and performed IP-MS on the FTS-3xHA parasite lysates and compared protein enrichment to the HOOK-3xHA IP (*Figure 5A*). Four proteins were significantly enriched in the reciprocal IP experiments—HOOK, FTS, TGGT1_316650, and TGGT1_306920—confirming the four-member complex. We also observed a similar localization to HOOK by immunofluorescence: primarily cytosolic with partial overlap with micronemes (*Figure 5B*). To further confirm the interaction, we fused a 3xHA tag to the C terminus of TGGT1_306920, performed IP-MS and compared protein enrichment to the HOOK-3xHA IP (*Figure 5C*). HOOK, FTS, and TGGT1_306920 were significantly enriched across both IP-MS experiments, whereas TGGT1_316650 is only significantly enriched in HOOK and FTS pulldowns. This suggests the presence of multiple HOOK complexes composed of the core HOOK and FTS proteins that bind with either TGGT1_316650 or TGGT1_306920.

To determine whether FTS functions in the same pathway as HOOK, we generated a conditional knockdown by fusing an AID-HA tag to the C terminus of FTS (FTS-AID; *Figure 5—figure supplement 1*). FTS was readily depleted from parasites following 40 hr of auxin treatment (*Figure 5D*). FTS depletion completely blocked plaque formation (*Figure 5E*). Similar to HOOK, depletion of FTS had no observable effect on parasite replication or microneme biogenesis (*Figure 5—figure supplement 1*, *Figure 5F*). During the motile stages of the parasite, FTS depletion resulted in a block in invasion efficiency but had no effect on egress, phenocopying the effects of HOOK depletion (*Figure 5G and H*). FTS appeared to tolerate tagging better than HOOK since the FTS-AID parasites behaved normally in the absence of auxin. These data suggest that HOOK and FTS form a functional complex required for parasite invasion.

We pursued enzyme-catalyzed proximity labeling to complement IP-MS studies and capture transient protein interactions (*Branon et al., 2018*). In this approach, cells express a protein of interest tagged with the promiscuous biotin ligase TurboID. Addition of a biotin substrate to live cells results in biotinylation within a few nanometers of the TurboID-tagged protein in a matter of minutes. Biotinylated proteins can be enriched from cellular lysates using streptavidin-affinity purification and identified using MS. Proximity labeling was recently used to characterize the cargo diversity of distinct human FHF complexes in human cells (*Christensen et al., 2021*). To identify additional proteins that interact with the *T. gondii* HOOK complex, we performed proximity labeling after fusing TurboID to the C terminus of FTS, which was more amenable to tagging compared to HOOK (*Figure 5I*, *Figure 5—figure supplement 1*). As a control, we used parasites expressing a cytosolic mNeonGreen-TurboID that would broadly label cytosolic proteins and identify non-specific interactions (*Figure 5—figure supplement 1*).

Of the 14 proteins significantly enriched in FTS-TurboID parasites, members of the HOOK complex (HOOK, FTS, TGGT1_306920, and TGGT1_316650) were the top four most enriched proteins. In humans, proximity labeling studies suggest FTS and FHIP bind at the C terminus of the HOOK dimer to mediate FHIP binding to RAB5 endosomes (*Christensen et al., 2021*). Our FTS proximity labeling results suggest that TGGT1_306920 and TGGT1_316650 may also bind in this manner. Among the other significantly enriched proteins, PC and ACC1 encode carboxylases known to be covalently modified by biotin and likely represent non-specific enrichment. DHFR enrichment is likely due to its use as a selectable marker when generating the FTS-TurboID strain. TGGT1_294610 and TGGT1_280770 are a putative histone lysine methyltransferase and regulator of chromosome condensation (RCC1)-repeat containing protein, respectively, and have no characterized functions. Of the remaining enriched proteins that lack annotations, TGGT1_221180 was previously localized to micronemes by spatial proteomics and contains a transmembrane domain, suggesting a molecular link between the HOOK complex and micronemes (*Barylyuk et al., 2020*).

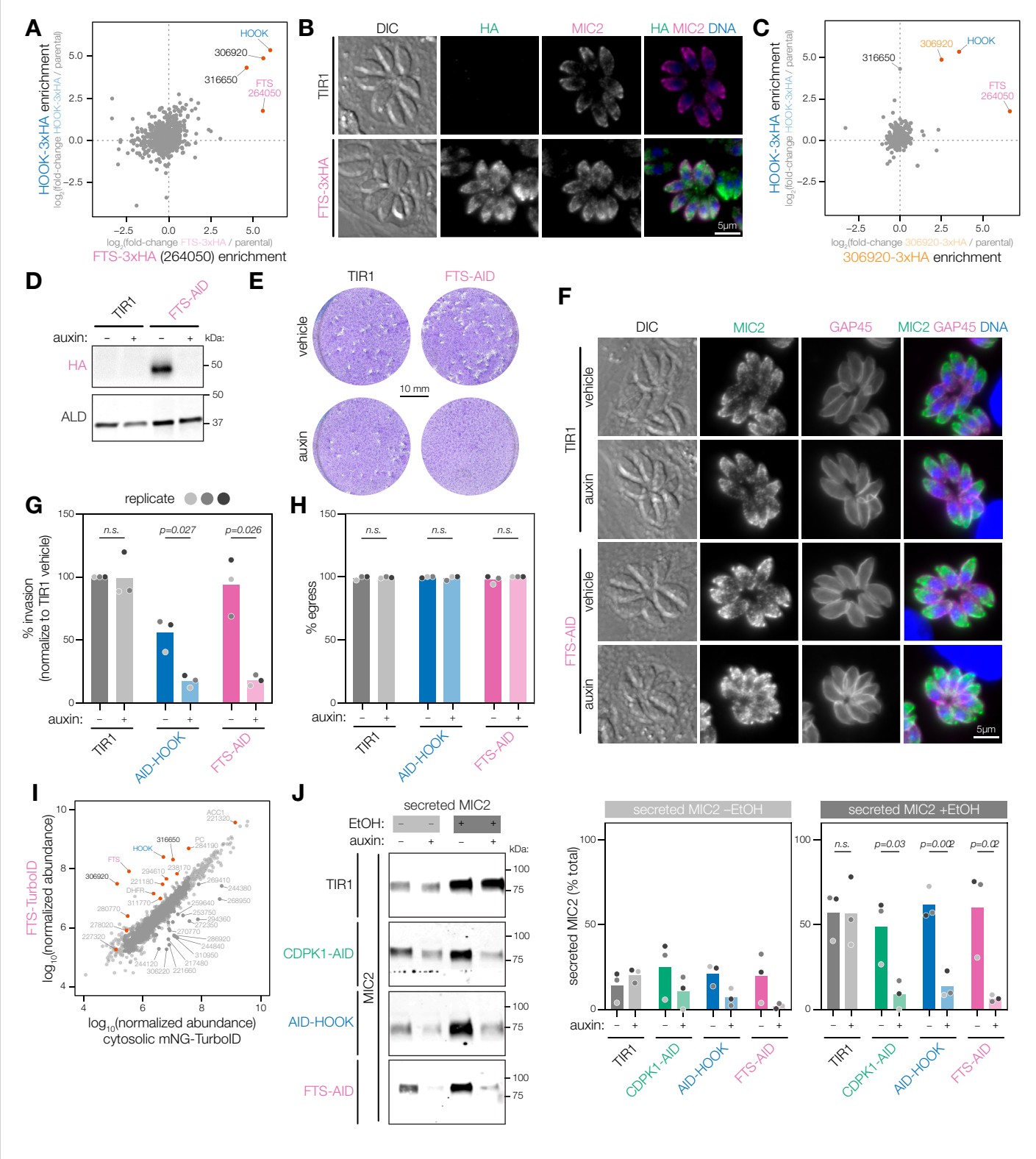

**Figure 5.** The HOOK complex is required for microneme exocytosis. (**A**) Reciprocal IP-MS of HOOK-3xHA and FTS-3xHA. FTS is tagged with a C-terminal 3xHA epitope at the endogenous locus (FTS-3xHA). IP enrichment is shown as the fold-change of protein abundances in tagged versus untagged strains determined by LC-MS/MS across n = 3 biological replicates. Significantly enriched proteins with more than 3 unique peptides highlighted (red); p$_{HOOK}$<0.05, and p$_{FTS}$<0.05; ANOVA was Benjamini–Hochberg corrected. (**B**) FTS-3xHA visualized in fixed intracellular parasites by

*Figure 5 continued on next page*

*Figure 5 continued*

immunofluorescence after treatment with auxin for 24 hr. Hoechst and MIC2 are used as counterstains. (**C**) Reciprocal IP-MS of HOOK-3xHA and 306920-3xHA. *TGGT1_306920* is tagged with a C-terminal 3xHA epitope at the endogenous locus (306920-3xHA). IP enrichment is shown as a fold-change of protein abundances in tagged versus untagged strains determined by LC-MS/MS across n = 3 and n = 2 biological replicates for the HOOK-3xHA and 306920-3xHA IP, respectively. Significantly enriched proteins with more than hree unique peptides highlighted (red); $p_{HOOK}<0.05$ and $p_{306920}<0.05$; ANOVA was Benjamini–Hochberg corrected. (**D**) Immunoblot of FTS cKD parasites. FTS is tagged with an C-terminal mAID-HA at its endogenous locus (FTS-AID) and treated with auxin for 40 hr. ALD is used as a loading control. (**E**) Plaque assays of host cells infected with TIR1 or FTS-AID parasites for 8 d in auxin. Host cells are stained with crystal violet. (**F**) Micronemes are visualized in fixed intracellular FTS-AID and TIR1 parasites by immunofluorescence after treatment auxin for 24 hr. Hoechst and GAP45 are used as counterstains. (**G**) Invasion assays of untagged TIR1, AID-HOOK, and FTS-AID parasites treated auxin for 40 hr. Medians are plotted for n = 3 biological replicates (different shades of gray), n.s., p>0.05, Welch's *t*-test. (**H**) Parasite egress stimulated zaprinast following auxin treatment for 24 hr. Egress was monitored by live microscopy. Percent egress plotted for n = 3 biological replicates, n.s., p>0.05, Welch's *t*-test. (**I**) Proximity labeling MS of FTS using TurboID (FTS-TurboID) compared to a cytosolic TurboID control (cytosolic mNeonGreen-TurboID). Protein abundances determined by LC-MS/MS are shown for n = 3 biological replicates. Significantly enriched proteins in FTS-TurboID are colored in red (red and blue), unique peptides >3, ratio > 1, p<0.05, ANOVA was Benjamini–Hochberg corrected. (**J**) Microneme protein secretion assays of parasites treated with auxin for 40 hr. Extracellular parasites are stimulated with 1% ethanol (EtOH) and 3% IFS for 1.5 hr. Percent MIC2 secreted is plotted for n = 3 biological replicates, n.s., p>0.05, Welch's *t*-test.

The online version of this article includes the following source data and figure supplement(s) for figure 5:

**Source data 1.** This file contains source data that was used to generate the blot in *Figure 5* (FTS-AID-HA, chemiluminescence).

**Source data 2.** This file contains source data that was used to generate the blot in *Figure 5* (FTS-AID-HA, colorimetric, ladder).

**Source data 3.** This file contains source data that was used to generate the blot in *Figure 5* (ALD, LI-COR 700 channel).

**Source data 4.** This file contains source data that was used to generate the blot in *Figure 5* and *Figure 5—figure supplement 1* (MIC2, LI-COR 800 channel [TIR1]).

**Source data 5.** This file contains source data that was used to generate the blot in *Figure 5* and *Figure 5—figure supplement 1* (MIC2, LI-COR 800 channel [CDPK1-AID]).

**Source data 6.** This file contains source data that was used to generate the blot in *Figure 5* and *Figure 5—figure supplement 1* (MIC2, LI-COR 800 channel [AID-HOOK]).

**Source data 7.** This file contains source data that was used to generate the blot in *Figure 5* and *Figure 5—figure supplement 1* (MIC2, LI-COR 800 channel [FTS-AID]).

**Source data 8.** This file contains the annotated source data that were used to generate blots in *Figure 5*.

**Source data 9.** This file contains source data that was used to make the graph presented in *Figure 5G* (GraphPad Prism data).

**Source data 10.** This file contains source data that was used to make the graphs presented in *Figure 5J* (GraphPad Prism data).

**Figure supplement 1.** Extended analysis of FTS knockdown, proximity labeling, and microneme protein secretion.

**Figure supplement 1—source data 1.** This file contains source data that was used to generate the DNA gel in *Figure 5—figure supplement 1* (FTS-AID integration).

**Figure supplement 1—source data 2.** This file contains source data that was used to generate the DNA gel in *Figure 5—figure supplement 1* (FTS-TurboID tag).

**Figure supplement 1—source data 3.** This file contains source data that was used to generate the blot in *Figure 5—figure supplement 1* (Biotin, LI-COR 700 channel).

**Figure supplement 1—source data 4.** This file contains source data that was used to generate the blot in *Figure 5—figure supplement 1* (CDPK1, LI-COR 800 channel).

**Figure supplement 1—source data 5.** This file contains source data that was used to generate the blot in *Figure 5—figure supplement 1* (CDPK1, LI-COR 700 channel [TIR1 standard]).

**Figure supplement 1—source data 6.** This file contains source data that was used to generate the blot in *Figure 5—figure supplement 1* (GAP45, LI-COR 700 channel [CDPK1-AID standard]).

**Figure supplement 1—source data 7.** This file contains source data that was used to generate the blot in *Figure 5—figure supplement 1* (CDPK1, LI-COR 700 channel [AID-HOOK standard]).

**Figure supplement 1—source data 8.** This file contains source data that was used to generate the blot in *Figure 5—figure supplement 1* (CDPK1, LI-COR 700 channel [FTS-AID standard]).

**Figure supplement 1—source data 9.** This file contains the annotated source data that were used to generate blots in *Figure 5—figure supplement 1* (CDPK1, LI-COR 700 channel [FTS-AID standard]).

**Figure supplement 1—source data 10.** This file contains source data that was used to make the graph presented in *Figure 5—figure supplement 1C* (GraphPad Prism data).

## The HOOK complex promotes invasion by regulating microneme exocytosis

To determine whether the HOOK complex is required for microneme exocytosis, we directly measured secretion of the microneme protein MIC2. Microneme exocytosis exposes integral membrane proteins like MIC2 that function as adhesins during gliding motility. These adhesins subsequently undergo proteolytic cleavage and are released into the supernatant, which can be analyzed by immunoblot (*Carruthers et al., 2000*). Quantification of the proportion of secreted MIC2 was performed by generating a standard curve from unstimulated lysates. Strikingly, parasites depleted of either member of the HOOK complex were severely impaired in their ability to secrete microneme proteins (*Figure 5J*). Inhibition of the complex led to a similar degree of secretion impairment as CDPK1 depletion. Together, these data suggest that HOOK and FTS form a functional complex that is required for microneme exocytosis.

## Microneme trafficking depends on CDPK1 activity and HOOK

If the HOOK complex is required for sustained microneme exocytosis, then depletion of HOOK would inhibit trafficking of micronemes during parasite motility. Dynamic relocalization of micronemes can be visualized using live microscopy of parasites expressing the integral microneme protein CLAMP fused to mNeonGreen (CLAMP-mNG) (*Figure 6A*; *Sidik et al., 2016b*). To trigger microneme exocytosis, parasites are treated with zaprinast to stimulate a rise in intracellular $Ca^{2+}$ and activate CDPK1. Dynamic relocalization of micronemes is defined as the CLAMP-mNG signal that concentrates at the apical end of the parasite over time. To quantify microneme relocalization, mNG signal is measured, over time, along the apical-basal axis of individual parasites within a vacuole. We scored microneme relocalization by calculating the difference in maximum apical intensity comparing the instant of zaprinast addition to the time point just prior to parasite egress of the uninhibited zaprinast-stimulated controls (*Figure 6A*).

We first assessed whether CDPK1 kinase activity is required for microneme relocalization using the specific inhibitor 3-MB-PP1 (*Lourido et al., 2010*). We incubated parasites with 3-MB-PP1 for 30 min, imaged for 1 min to establish a baseline, and then stimulated with zaprinast. Parasites pre-treated with DMSO relocalized micronemes as expected; however, microneme relocalization was blocked when CDPK1 was inhibited by 3-MB-PP1 (*Figure 6B–D*) (*Videos 1–3*). These results indicate microneme relocalization depends on CDPK1 kinase activity.

We next assessed whether HOOK is also required for microneme relocalization. We fused an mNG reporter to the C terminus of CLAMP in AID-HOOK parasites and TIR1 parasites as a control (*Figure 6—figure supplement 1*). We treated parasites with auxin for 24 hr prior to stimulation with zaprinast. Microneme relocalization was unaffected in TIR1/CLAMP-mNG parasites treated with auxin (*Figure 6E and F*) (*Videos 4–6*). By contrast, microneme relocalization did not occur in intracellular AID-HOOK parasites regardless of auxin treatment (*Figure 6G and H*) (*Videos 7–9*). We attributed the lack of relocalization in vehicle-treated parasites to the hypomorphism resulting from N-terminal tagging, which is consistent with the delayed invasion and reduced plaque size documented for this strain. To determine whether vehicle-treated AID-HOOK parasites relocalized micronemes over longer periods, we examined CLAMP-mNG localization in extracellular parasites after egress. While vehicle-treated AID-HOOK parasites were able to relocalize micronemes, parasites depleted of HOOK exhibited micronemes dispersed throughout the cytosol (*Figure 6I and J*). As expected, TIR1/CLAMP-mNG parasites maintained the apical relocalization of micronemes regardless of auxin treatment in extracellular parasites. We used an alternative metric for microneme relocalization to account for the disorganized localization of micronemes in parasites depleted of HOOK. We quantified the percent of the total CLAMP-mNG signal localized to the apical end versus the remaining body of the parasite (*Figure 6K*). The apical region cutoffs were experimentally derived from vehicle-treated TIR1/CLAMP-mNG parasites and defined as the apical 12.5% of the parasite. While a majority of CLAMP-mNG signal was localized to the apical region of TIR1/CLAMP-mNG parasites, most of the CLAMP-mNG signal observed in parasites depleted of HOOK was found in the parasite body. Micronemes rapidly adopt an aberrant localization in the absence of HOOK during parasite motility. These results indicate that CDPK1 kinase activity and HOOK are required for microneme trafficking during parasite motility.

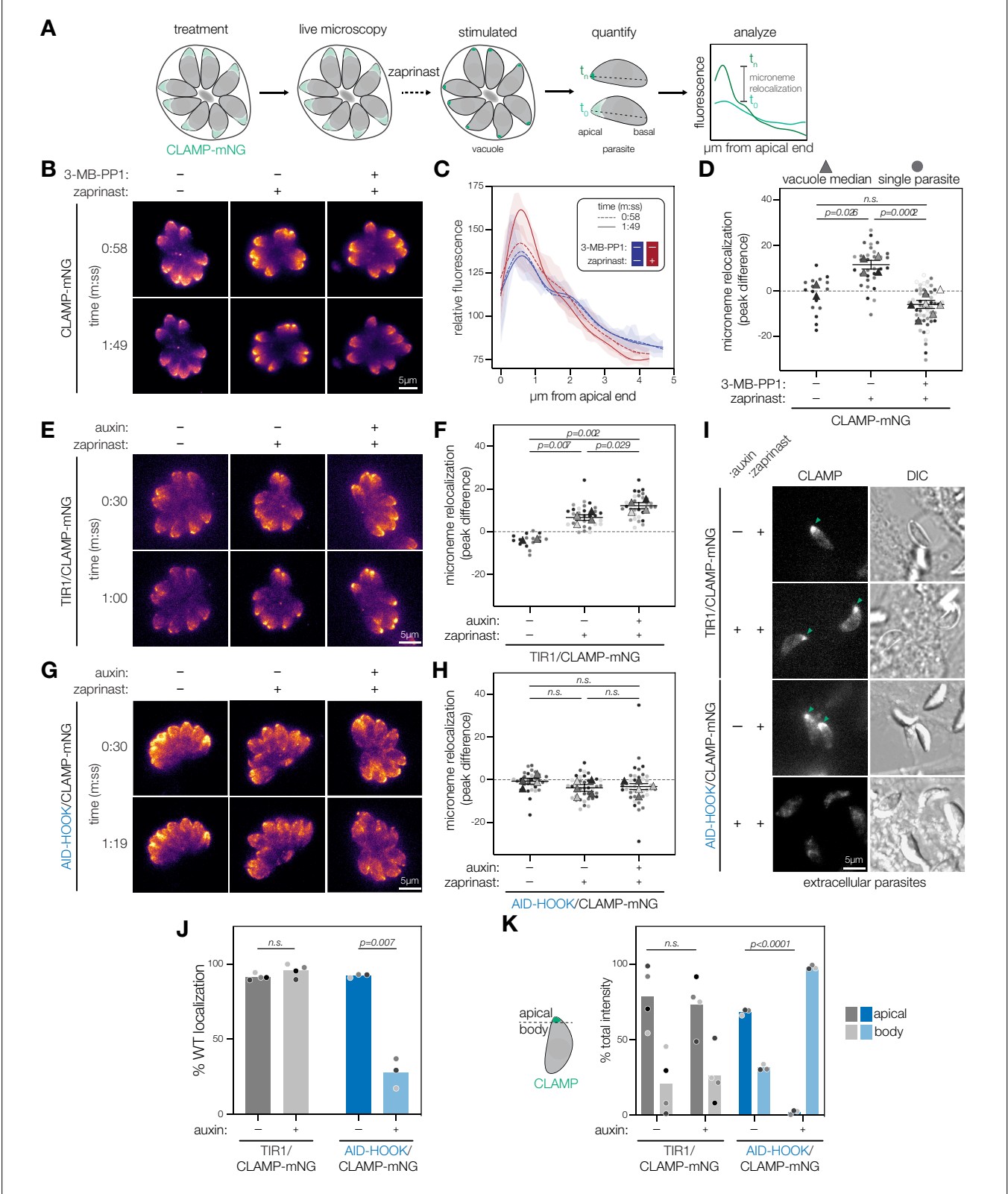

**Figure 6.** CDPK1 activity and HOOK are required for microneme trafficking during parasite motility stages. (**A**) Schematic to analyze microneme trafficking during parasite motile stages. Intracellular parasites expressing microneme protein CLAMP endogenously tagged with mNeonGreen (CLAMP-mNG). Parasites are treated either with 3-MB-PP1 (inhibit CDPK1) or auxin (for conditional knockdown). Live microscopy was performed to detect CLAMP-mNG signal over time. Zaprinast was added at 1 min or 30 s to stimulate microneme relocalization to the apical end of the parasite.

*Figure 6 continued on next page*

*Figure 6 continued*

Fluorescence intensities across the apical-basal axis of each individual parasite within a vacuole was measured across time. Microneme relocalization was quantified by calculating the difference of maximum CLAMP intensity between time points preceding drug addition and egress. (**B**) Maximum intensity projections at single time points of CLAMP-mNG parasites treated with 3 µM 3-MB-PP1 or vehicle and zaprinast. (**C**) Relative fluorescence intensity of CLAMP-mNG signal across the apical-basal axis of parasites in (**B**). Zaprinast (red) or vehicle (blue). Splines mean intensity for all parasites in each vacuole are shown with SD shaded. (**D**) Microneme relocalization. SuperPlots showing vacuole median peak differences are displayed as triangles. Individual parasites are displayed as circles. Replicates are differentially shaded, n.s., p>0.05, unpaired *t*-test. (**E**) Maximum intensity projections at single time points of TIR1/CLAMP-mNG parasites treated with auxin and stimulated with zaprinast. (**F**) Microneme relocalization was quantified for TIR1/CLAMP-mNG parasites as in (**D**). (**G**) Maximum intensity projections at single time points of AID-HOOK/CLAMP-mNG parasites treated auxin and stimulated with zaprinast. (**H**) Microneme relocalization was quantified for AID-HOOK/CLAMP-mNG parasites as in (**D**). (**I**) Maximum intensity projections of extracellular TIR1/CLAMP-mNG and AID-HOOK/CLAMP-mNG parasites. (**J**) Percent of extracellular parasites in (**I**) with WT CLAMP-mNG localization, n.s., p>0.05, Welch's *t*-test. (**K**) Percent total CLAMP-mNG signal intensity in the apical versus body of extracellular parasites, n.s., p>0.05, Welch's *t*-test.

The online version of this article includes the following source data and figure supplement(s) for figure 6:

**Source data 1.** This file contains source data that was used to make the graph presented in *Figure 6C* (GraphPad Prism data).

**Source data 2.** This file contains source data that was used to make the graph presented in *Figure 6D* (GraphPad Prism data).

**Source data 3.** This file contains source data that was used to make the graph presented in *Figure 6F* (GraphPad Prism data).

**Source data 4.** This file contains source data that was used to make the graph presented in *Figure 6H* (GraphPad Prism data).

**Source data 5.** This file contains source data that was used to make the graph presented in *Figure 6J* (GraphPad Prism data).

**Source data 6.** This file contains source data that was used to make the graph presented in *Figure 6K* (GraphPad Prism data).

**Figure supplement 1.** Extended analysis of FTS knockdown, proximity labeling, and microneme protein secretion.

**Figure supplement 1—source data 1.** This file contains source data that was used to generate the DNA gel in *Figure 6—figure supplement 1* (CLAMP-mNeonGreen integration [TIR1/CLAMP-mNG]).

**Figure supplement 1—source data 2.** This file contains source data that was used to generate the DNA gel in *Figure 6—figure supplement 1* (CLAMP-mNeonGreen integration [AID-HOOK/CLAMP-mNG]).

**Figure supplement 1—source data 3.** This file contains the annotated source data that were used to generate blots in *Figure 6—figure supplement 1*.

## Ultrastructure expansion microscopy reveals HOOK is required for apical microneme positioning

We performed expansion microscopy to determine whether micronemes remain associated with cortical microtubules or detach in the absence of HOOK (*Dos Santos Pacheco and Soldati-Favre, 2021*; *Gambarotto et al., 2021*; *Gambarotto et al., 2019*; *Pavlou et al., 2020*; *Tosetti et al., 2020*). We

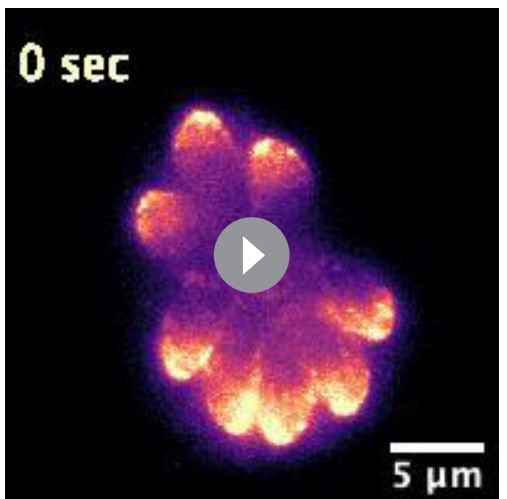

**Video 1.** Representative image series of parasites expressing endogenously tagged CLAMP-mNG following pre-treatment with DMSO and stimulation with DMSO.

https://elifesciences.org/articles/85654/figures#video1

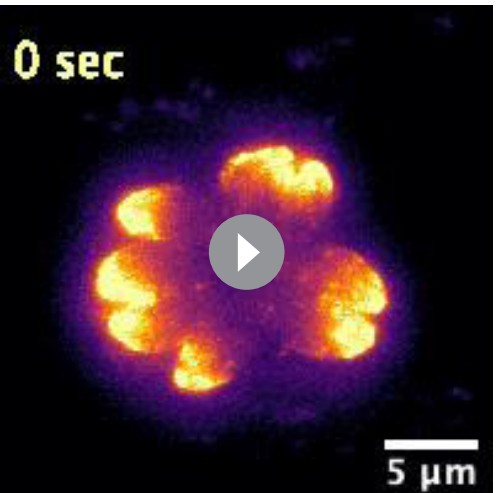

**Video 2.** Representative image series of parasites expressing endogenously tagged CLAMP-mNG following pre-treatment with DMSO and stimulation with 500 µM zaprinast.

https://elifesciences.org/articles/85654/figures#video2

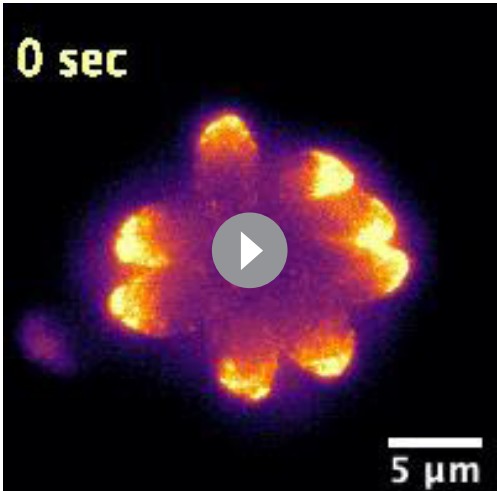

**Video 3.** Representative image series of parasites expressing endogenously tagged CLAMP-mNG following pre-treatment with 3 µM 3-MB-PP1 and stimulation with 500 µM zaprinast.

https://elifesciences.org/articles/85654/figures#video3

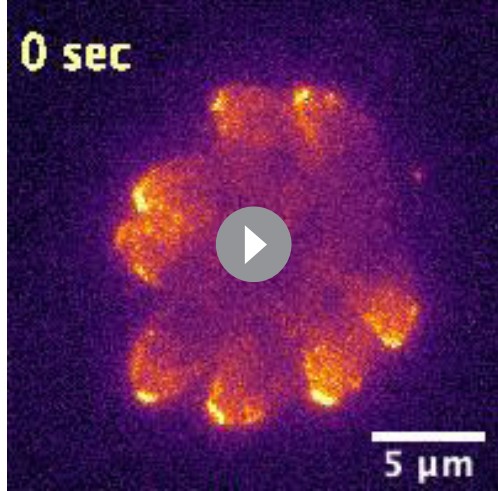

**Video 4.** Representative image series of parasites expressing endogenously tagged TIR1/CLAMP-mNG following pre-treatment with vehicle and stimulation with DMSO.

https://elifesciences.org/articles/85654/figures#video4

examined extracellular parasites after natural egress. Approximately 60% of parasites displayed extruded conoids, indicating active motility, and this proportion was unaffected by HOOK knockdown (*Figure 7—figure supplement 1*). We obtained maximum intensity projections of parasites with extruded conoids using confocal microscopy (*Figure 7A*). Three-dimensional reconstructions of the cortical microtubule filaments and microneme organelles were generated using acetylated tubulin and MIC2 intensities, respectively (*Figure 7B*). HOOK-knockdown parasites contained significantly more micronemes, compared to the parental strain, consistent with HOOK's function in microneme exocytosis (*Figure 7—figure supplement 1*). We calculated distances from individual micronemes to either the apical end of the parasite or the nearest microtubule filament (*Figure 7—figure supplement 1*). HOOK knockdown resulted in a significant depletion of micronemes in the apical region of parasites (*Figure 7C*). However, the distance between micronemes and the nearest microtubule remained unaffected by HOOK knockdown (*Figure 7D*). These results are consistent with

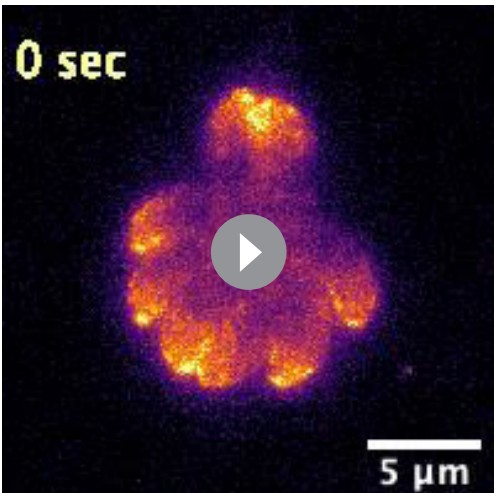

**Video 5.** Representative image series of parasites expressing endogenously tagged TIR1/CLAMP-mNG following pre-treatment with vehicle and stimulation with 500 µM zaprinast.

https://elifesciences.org/articles/85654/figures#video5

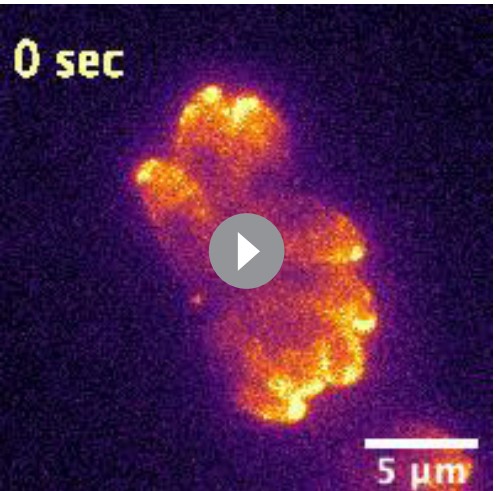

**Video 6.** Representative image series of parasites expressing endogenously tagged TIR1/CLAMP-mNG following pre-treatment with auxin and stimulation with 500 µM zaprinast.

https://elifesciences.org/articles/85654/figures#video6

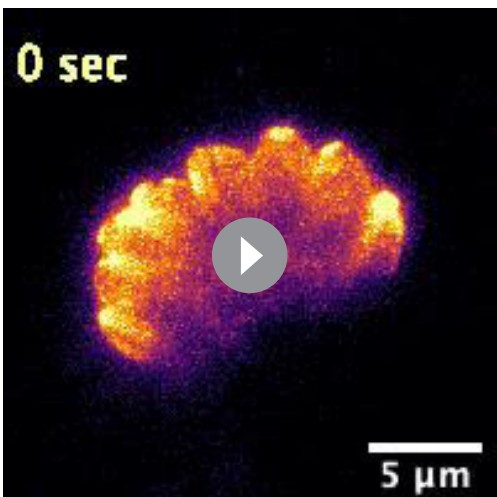

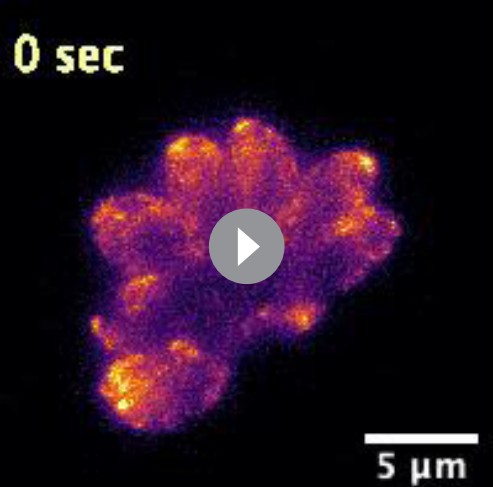

**Video 7.** Representative image series of parasites expressing endogenously tagged AID-HOOK/CLAMP-mNG following pre-treatment with vehicle and stimulation with DMSO.

https://elifesciences.org/articles/85654/figures#video7

**Video 8.** Representative image series of parasites expressing endogenously tagged AID-HOOK/CLAMP-mNG following pre-treatment with vehicle and stimulation with 500 µM zaprinast.

https://elifesciences.org/articles/85654/figures#video8

the requirement of HOOK for apical trafficking of micronemes during exocytosis, but suggest mislocalized micronemes remain associated to the cortical microtubules by an unknown factor in the absence of HOOK.

## Discussion

In this study, we sought to identify new regulators of $Ca^{2+}$-mediated exocytosis in *T. gondii* by studying the targets of a key regulator, the kinase CDPK1. We identified 163 proteins phosphorylated in a CDPK1-dependent manner using sub-minute resolution phosphoproteomics and thiophosphorylation for direct substrate capture. We determined that myristoylation of CDPK1 contributes to the kinase's function during the lytic cycle. Thirteen of the identified CDPK1 targets have previously described functions in parasite motility, revealing possible points of regulation within relevant signaling pathways. Furthermore, we characterized a new regulator of microneme exocytosis called HOOK, which forms a stable complex with FTS and two other proteins. Homologs of HOOK and FTS participate in dynein-mediated vesicular trafficking in other organisms. In *T. gondii*, knockdown of HOOK or FTS blocked invasion of host cells and altered rapid microneme trafficking during $Ca^{2+}$-regulated motility. Overall, we show how studying parasite signaling pathways can illuminate the cellular adaptations that support parasitism.

Over the past decade, several efforts have sought to characterize phosphorylation within apicomplexans. Improvements in phosphopeptide enrichment and mass spectrometry have enabled global characterization of the phosphoproteomes from *T. gondii* tachyzoites and *Plasmodium falciparum* schizonts, trophozoites, and ring-stages (*Lasonder et al., 2012*; *Pease et al., 2013*; *Solyakov et al., 2011*; *Treeck*

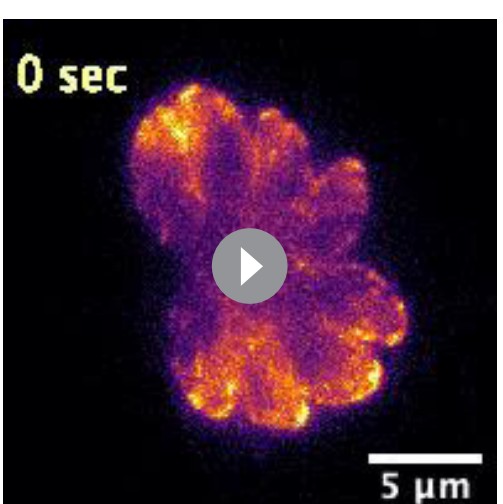

**Video 9.** Representative image series of parasites expressing endogenously tagged AID-HOOK/CLAMP-mNG following pre-treatment with auxin and stimulation with 500 µM zaprinast.

https://elifesciences.org/articles/85654/figures#video9

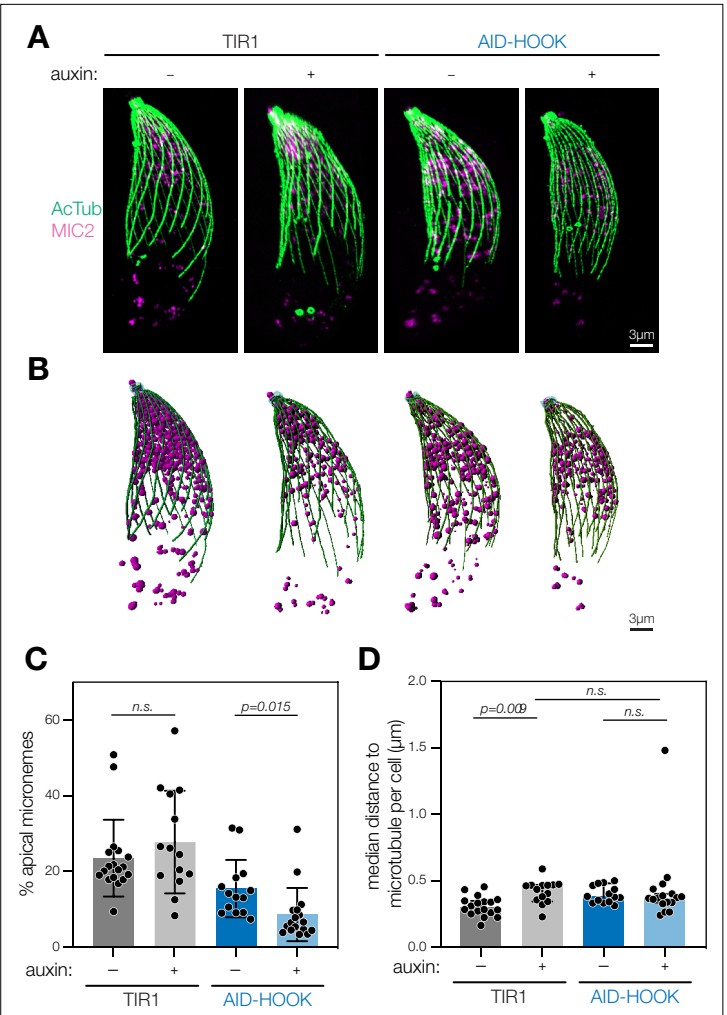

**Figure 7.** Ultrastructure expansion microscopy reveals HOOK is required for apical microneme positioning. (**A**) Maximum intensity projection of fixed extracellular parasites subjected to expansion microscopy. Parasites were pre-treated with auxin. Acetylated tubulin (Lys40) is used to visualize cortical microtubules (green) and MIC2 to visualize micronemes (magenta). (**B**) 3D reconstruction of maximum intensity projections. Filaments are constructed for cortical microtubules (green) and globular organelles constructed for micronemes (magenta). (**C**) Percent of all micronemes localized to the apical region (4 μm from parasite apex) in each parasite. Mean ± SD plotted for n = 14–18 parasites, n.s., p>0.05, Welch's *t*-test. (**D**) Median shortest distance between individual micronemes and closest cortical microtubule per parasite. Median with 95% confidence interval plotted for n = 14–18 parasites, n.s., p>0.05, Welch's *t*-test. Actual distances in expanded samples were used. The expansion factor was 4×.

The online version of this article includes the following source data and figure supplement(s) for figure 7:

**Source data 1.** This file contains source data that was used to make the graph presented in *Figure 7C* (GraphPad Prism data).

**Source data 2.** This file contains source data that was used to make the graph presented in *Figure 7D* (GraphPad Prism data).

**Figure supplement 1.** Extended analysis of ultrastructure expansion microscopy of HOOK knockdown parasites.

**Figure supplement 1—source data 1.** This file contains source data that was used to make the graph presented in *Figure 7—figure supplement 1B* (GraphPad Prism data).

**Figure supplement 1—source data 2.** This file contains source data that was used to make the graph presented in *Figure 7—figure supplement 1C* (GraphPad Prism data).

**Figure supplement 1—source data 3.** This file contains source data that was used to make the graph presented in *Figure 7—figure supplement 1D* (GraphPad Prism data).

*et al., 2011*). Coupling such approaches with genetic and pharmacological tools has enabled the deconvolution of kinase-specific effects from the vast global phosphorylation program (*Bhappu et al., 2020*; *Fang et al., 2017*; *Invergo et al., 2017*; *Nofal et al., 2022*; *Pease et al., 2018*; *Treeck et al., 2014*), revealing the activation of MyoA-mediated gliding by *Tg*CDPK3, the roles of *Pb*CDPK4 and *Pb*SRPK1 during male gametogenesis, and the erythrocytic-stage effects of *Pf*CDPK1, *Pf*PK7, and *Pf*CDPK5 (*Bhappu et al., 2020*; *Fang et al., 2017*; *Gaji et al., 2015*; *Invergo et al., 2017*; *Pease et al., 2018*; *Tang et al., 2014*). Attributing individual phosphorylation events to specific kinases can be complicated by cumulative changes or adaptation of the signaling networks in knockouts. Chemical-genetic approaches can more precisely inhibit the activity of a given kinase at the time of the assay, while controlling for off-target effects, and have been used to characterize *Pf*PKG-dependent phosphorylation (*Alam et al., 2015*; *Brochet et al., 2014*). Kinetically resolved phosphoproteomics has enabled a more nuanced understanding of signaling cascades during parasite motility (*Herneisen et al., 2022*; *Invergo et al., 2017*; *Nofal et al., 2022*). While such technical advances have led to more in-depth phosphoproteome mapping, identifying the direct targets of a given kinase remains a key challenge to structuring the observed changes into signaling pathways.

To resolve direct and indirect effects on the phosphoproteome for *T. gondii* CDPK1, we combined two complementary approaches: temporally resolved phosphoproteomics and bio-orthogonal labeling of direct substrates. We implemented several methodological advances. Conditional knock-down using the auxin-inducible degron system (*Brown et al., 2018a*) resulted in rapid and specific depletion of the kinase of interest for the phosphoproteomic studies (*Brown et al., 2018a*). Additionally, quantification and coverage by mass spectrometry were improved through TMTpro multiplexing and ion mobility spectrometry (FAIMS) (*Bekker-Jensen et al., 2020*; *Li et al., 2020*). We also improved on the low coverage observed in previous thiophosphorylation experiments with parasite lysates (*Fang et al., 2017*; *Lourido et al., 2013a*) by maintaining a more native signaling environment using aerolysin semi-permeabilization, SILAC-based peptide quantification, and shortened labeling times (*Rothenberg et al., 2016*). These approaches characterize proteome-wide phosphorylation kinetics in live parasites and reveal direct kinase–substrate relationships for CDPK1.

We observed that myristoylation of CDPK1 contributes to its function. Contrary to previous reports describing the kinase as cytosolic or nuclear (*Ojo et al., 2010*; *Pomel et al., 2008*), we demonstrate that myristoylated CDPK1 is at least partially associated with structures that can be fractionated from the cytoplasm by differential centrifugation. Loss of myristoylation led to CDPK1 release from the insoluble fraction. The ortholog of CDPK1 in *Plasmodium berghei*, *Pb*CDPK4, also displays myristoylation-specific functions during male gametogenesis: myristoylated *Pb*CDPK4 is critical for the first genome replication, whereas the non-myristoylated *Pb*CDPK4 is important for the completion of gametogenesis (*Fang et al., 2017*). Myristoylation could impact CDPK1's ability to access certain targets efficiently. However, such interactions appear to be too weak or transient to be captured by immunoprecipitation. Mutating the myristoylation site of CDPK1 only modestly affected ionophore-induced egress. However, minor effects in these key transitions may be magnified over repeated lytic cycles, which could explain the more substantial impact of losing CDPK1 myristoylation on plaque formation. Alternatively, other kinases may compensate for decreased CDPK1 activity, particularly under hyperactivated conditions like stimulated egress. A plausible candidate for compensation is CDPK2A, which also appears to be myristoylated and was recently shown to display epistatic interactions with CDPK1 (*Broncel et al., 2020*; *Shortt et al., 2022*).

CDPK1 plays a critical role in the transition from the replicative intracellular stages to motile extracellular stages (*Lourido et al., 2010*; *Shortt et al., 2022*). During this transition, parasites must execute rapid cellular changes that involve the exocytosis of micronemes and rhoptries, reorganization of the cytoskeleton, gliding motility, and maintenance of ion homeostasis (*Blader et al., 2016*). CDPK1 has also been suggested to control the actomyosin system and extrusion of the conoid (*Tosetti et al., 2019*). Our detailed target analysis suggests CDPK1 regulates—and possibly helps coordinate—multiple pathways. We revealed over a hundred CDPK1 targets with diverse predicted functions, including many relevant to phenotypes dependent on CDPK1. In addition to the effects on microneme trafficking, our results point to a direct link between CDPK1 and rhoptry exocytosis. $Ca^{2+}$ has been implicated in rhoptry discharge, as the rhoptry-localized $Ca^{2+}$-binding FER2 is required for rhoptry exocytosis (*Coleman et al., 2018*). We observed a preponderance of CDPK1-dependent phosphorylation on proteins regulating rhoptry exocytosis, including DHHC7, ARO, and AIP. Phosphorylation

on ARO was observed near the N-terminal acylation sites required for rhoptry targeting and may regulate the bundling and positioning of mature rhoptries during motile stages (*Mueller et al., 2016*). However, formally demonstrating the relationship between Ca²⁺/CDPK1 and rhoptry discharge is complicated by the dependency of the latter on the secretion of certain microneme proteins (*Ben Chaabene et al., 2021*; *Carruthers and Sibley, 1997*). *T. gondii* tachyzoites have several rhoptries, yet only two are docked for exocytosis at a given time (*Aquilini et al., 2021*; *Mageswaran et al., 2021*; *Segev-Zarko et al., 2022*). Regulating the activity of ARO during motile stages may influence the ability to mobilize and re-dock rhoptries in preparation for invasion. Considering how different CDPK1 substrates function, we expect the various phenotypes associated with CDPK1 will depend on distinct sets of substrates.

Although CDPK1 is known to mediate microneme exocytosis (*Lourido et al., 2010*), the precise molecular events controlling this process have remained elusive. In this study, we determined that CDPK1 activity is required for the Ca²⁺-stimulated trafficking of micronemes to the apical end. This phenotype depends on HOOK, one of the CDPK1 targets uncovered by both global phosphoproteomics and thiophosphorylation studies. Despite prior annotation as a microneme protein (*Butler et al., 2014*), HOOK likely functions as an activating adaptor in *T. gondii* based on homology to other Hook proteins and the functional characterization we have performed. Phosphorylation has been reported to regulate the function of activating adaptors. In HeLa cells, phosphorylation of BICD2 facilitates recruitment of dynein and dynactin (Gallisà-Suñé et al. 2023). Analogously, phosphorylation of JIP1 mediates the switch between kinesin and dynein motility of autophagosomes in murine neurons (*Fu et al., 2014*). We therefore speculate that phosphorylation of HOOK by CDPK1 may activate the adaptor by promoting its interaction with dynein and dynactin to initiate trafficking of micronemes.

Immunoprecipitation of HOOK identified three interacting proteins: FTS, TGGT_316650, and TGGT1_306920. In *Homo sapiens*, *Aspergillus nidulans*, and *D. melanogaster*, Hook proteins complex with FTS and FHIP (the FHF complex), to link cargo to the dynein machinery for trafficking along microtubules (*Bielska et al., 2014*; *Gillingham et al., 2014*; *Guo et al., 2016*; *Xu et al., 2008*; *Yao et al., 2014*). FHF complexes have been shown to recognize early endosomes via FHIP binding to RAB5 (*Bielska et al., 2014*; *Guo et al., 2016*; *Zhang et al., 2014*). Homology to FHIP was lacking in TGGT1_316650 and TGGT1_306920. However, genome-wide knockout screens indicate TGGT1_306920 is critical for parasite fitness, leading us to speculate that it may mediate cargo binding in a manner structurally distinct from known adaptors. Additional work will be needed to elucidate the function of the HOOK complex and to understand how it mediates trafficking of apicomplexan-specific organelles.

Our results place the HOOK complex downstream of CDPK1, regulating the stimulated relocalization of micronemes to the apical end of parasites. However, the HOOK complex only partially explains CDPK1's role during egress. Whereas CDPK1 depletion blocks egress entirely, parasites depleted of HOOK or FTS exit host cells, despite subsequent defects in microneme secretion and invasion. Microneme discharge is required for egress, supplying adhesins for gliding motility and the perforin PLP1 to rupture the parasitophorous vacuole membrane (*Kafsack et al., 2009*). It therefore appears that the initial round of microneme discharge during egress depends on CDPK1, and only subsequent rounds require the HOOK complex. Indeed, a fraction of micronemes are already found docked at the apical complex prior to the transition from the replicative to the motile stages, and may constitute the first round of microneme exocytosis (*Mageswaran et al., 2021*; *Sun et al., 2022*). Ferlin 1 (FER1) was recently shown to be involved in microneme positioning and overexpression of FER1 was sufficient to initiate an initial round of microneme exocytosis and induce egress (*Tagoe et al., 2020*). Dynein-mediated trafficking might become important immediately following egress when continuous microneme discharge is necessary, as is the case during gliding motility and invasion. Consistent with this hypothesis, depletion of the dynein light chain 8a (DLC8a) yielded similar phenotypes to HOOK and FTS knockdown: intact egress but dysfunctional microneme protein secretion, gliding motility, and invasion (*Lentini et al., 2019*). Loss of DLC8a also led to defects in rhoptry positioning, which were not observed in HOOK knockdowns (*Lentini et al., 2019*), so other activating adaptors may be involved in rhoptry trafficking. Our results suggest that the HOOK complex has been specifically adapted for microneme trafficking in *T. gondii*.

Several aspects of microneme trafficking remain to be determined. The precise nature of the dynein complex is poorly understood. Eukaryotes typically express a single cytoplasmic dynein heavy chain. In *T. gondii*, TGGT1_294550 is the top candidate for the cytoplasmic dynein heavy chain (DHC)

as it contains the necessary domains, is conserved among apicomplexans, and is required for parasite fitness (*Lentini et al., 2019*; *Sidik et al., 2016b*). Other putative DHCs in the *T. gondii* genome are likely axonemal dyneins required for flagellar function in sexual-stage parasites. Two putative plus-end motors have been characterized in *T. gondii*. KinesinA localizes to the apical polar ring where cortical microtubules emanate from and is required for ring stability and parasite motility (*Leung et al., 2017*). In contrast, KinesinB localizes to the distal two-thirds of the cortical microtubules but is dispensable for the lytic cycle (*Leung et al., 2017*). Motile parasites depleted of HOOK mislocalized micronemes to the distal two-thirds of the cortical microtubules. We hypothesize that KinesinB, which is also a target of CDPK1, may maintain the association between micronemes and microtubules in the absence of HOOK. It also remains unknown how the HOOK complex binds to micronemes. In *H. sapiens* and *D. melanogaster*, RAB5 on vesicles interacts with FHIP in the HOOK complex (*Bielska et al., 2014*; *Gillingham et al., 2014*; *Guo et al., 2016*; *Xu et al., 2008*; *Yao et al., 2014*). We speculate that TGGT1_306920 may serve the role of FHIP within the HOOK complex as it is fitness conferring, whereas TGGT1_316650 appears dispensable but the complex's binding partner on micronemes remains unknown. RAB5A and RAB5C have been implicated in the biogenesis of micronemes, but their roles during exocytosis have not been explored (*Kremer et al., 2013*). Understanding how micronemes are recognized may elucidate how cargo specificity is achieved and regulated.

$Ca^{2+}$ signaling has been tuned to support apicomplexan-specific cellular processes but identifying and integrating effectors within the pathway has remained a major bottleneck. Signaling effectors conserved amongst eukaryotes are often integrated into apicomplexan cellular pathways in novel or unusual ways. Here, we identified candidate downstream effectors of the $Ca^{2+}$-regulated kinase CDPK1 by monitoring proteome-wide protein phosphorylation. In addition to creating a catalog of candidates for future characterization, we identified, characterized, and integrated the HOOK complex downstream of CDPK1 within the $Ca^{2+}$ signaling network controlling microneme exocytosis. Lastly, our results are the first to implicate activating adaptors as critical factors for the pathogenesis of an apicomplexan organism by promoting exocytic trafficking.

# Materials and methods

## Cell culture

*T. gondii* parasites were grown in human foreskin fibroblasts (HFFs, ATCC SRC-1041) maintained in DMEM (GIBCO 11965118) supplemented with 3% inactivated fetal calf serum and 10 μg/mL gentamicin (Thermo Fisher Scientific), referred to as media. When noted, DMEM was supplemented with 10% inactivated fetal bovine serum (IFS) and 10 μg/mL gentamicin, referred to as 10% IFS media. Parasites and HFFs were grown at 37°C/5% $CO_2$ unless indicated otherwise. HFFs and parasites were tested routinely for mycoplasma using the ATCC Universal Mycoplasma Detection Kit (30-1012K).

## Parasite transfection and strain construction

### Genetic background of parasite strains

*T. gondii* RH strains were used as genetic backgrounds for this study. All strains contain the *Δku80Δhxgprt* mutations to facilitate homologous recombination (*Huynh and Carruthers, 2009*). TIR1 expresses the TIR1-FLAG ubiquitin ligase and the CAT enzyme conferring chloramphenicol resistance (*Brown et al., 2017*).

### Transfection

Parasites were pelleted at 1000 × *g* for 5–10 min and resuspended with Cytomix (10 mM $KPO_4$, 120 mM KCl, 0.15 mM $CaCl_2$, 5 mM $MgCl_2$, 25 mM HEPES, 2 mM EDTA, 2 mM ATP, and 5 mM glutathione) and combined with DNA to a final volume of 400 μL. Parasites were electroporated using an ECM 830 Square Wave electroporator (BTX) in 4 mm cuvettes with the following setting: 1.7 kV, 2 pulses, 176 μs pulse length, and 100 ms interval.

### Endogenous tagging of CDPK1 (CDPK1-AID)

CDPK1-AID was generated in the study (*Shortt et al., 2022*). Briefly, *V5-mNG-mAID-Ty* was PCR amplified from pBM050 (V5-TEV-mNG-mAID-Ty; GenBank: OM640006) to attach homology arms to *TGGT1_301440* (*CDPK1*). DNA was co-transfected with a Cas9 expression plasmid targeting *CDPK1*

in TIR1 parasites. mNG-expressing parasites were isolated using FACS and subcloned by limiting dilution. Tagging was confirmed by PCR, flow cytometry, and immunoblotting.

## Endogenous tagging of CDPK1 (iKD) and complementation (cWT and cMut)

Generating the inducible CDPK1 knock-down strain (iKD CDPK1). The pTUB1_YFP_mAID_3HA vector was amplified by inverse PCR using primer pair P1/P2 to substitute the 3xHA tag sequence for a Myc tag encoding sequence (*Brown et al., 2017*). A Cas9 expression plasmid targeting the 3′UTR of *CDPK1* was generated by inverse PCR on pSag1_Cas9-U6_sgUPRT using primers P3/P4 (*Shen et al., 2014*). The sequence encoding *mAID-Myc-HXGPRT* was PCR amplified using primer pair P5/P6 and co-transfected into TIR1 parasites with the Cas9 expression plasmid. Recombinant parasites were selected 24 hr post transfection by addition of mycophenolic acid (MPA; 25 µg/mL) and xanthine (XAN; 50 µg/mL) to culture medium. Lines were cloned, and successful 5′ and 3′ integration of the *mAID-Myc-HXGPRT* cassette was confirmed using primer pairs P30/P31 and P32/P33. Absence of WT was confirmed using primers P34/P35.

## Generating the cWT and cMut CDPK1 complementation strain

To generate the complementation construct, pUPRT_CDPK1_ HA_T2A_GFP (GenBank: pending), the *CDPK1* 5′UTR was amplified from genomic DNA using primer pair P7/P8. Recodonized *CDPK1* cDNA-HA sequence was synthesized (GeneArt strings, Life Technologies) and amplified with appropriate overhangs using primers P9/P10. Sequence encoding T2A-GFP was amplified from an in-house unpublished plasmid using primer pair P11/P12. The three resulting fragments were Gibson cloned into the PacI-linearized pUPRT_HA vector (*Reese et al., 2011*). To generate the complementation construct, pUPRT_CDPK1(G2A)_HA_T2A_mCherry (GenBank: pending), the CDPK1 5′UTR was amplified from genomic DNA using primer pair P13/P8. Recodonized *CDPK1-HA* was amplified from pUPRT_CDPK1_ HA_T2A_GFP with appropriate overhangs using primers P14/P15. Primers P13/P15 were used to introduce a G2A point mutation within *CDPK1*. Sequence encoding *T2A-mCherry* was amplified from an in-house unpublished plasmid using primer pair P11/P16. The three resulting PCR amplicons were Gibson cloned into the PacI-linearized pUPRT_HA vector. Complementation plasmids were linearized with AclI and individually co-transfected with the Cas9 expression plasmid targeting the *UPRT* locus. Transgenic parasites were subjected to 5′-fluo-2′-deoxyuridine (FUDR) selection (5 µM) 24 hr post transfection. Resistant parasites were cloned, and successful 5′ and 3′ integration was confirmed using primer pairs P36/P37 and P38/P39, respectively. Disruption of the endogenous *UPRT* locus was confirmed using primer pair P40/P41.

## Endogenous tagging of genes

Genes were endogenously tagged using the previously described high-throughput tagging (HiT) strategy (*Smith et al., 2022*). Cutting units specific to each gene were purchased as gene fragments (IDT gBlocks; P22, P23, and P50) and integrated with the following empty HiT vector backbones via Gibson assembly: pALH086 (V5-mAID-HA; GenBank: ON312869), pALH047 (V5-3HA; GenBank: ON312868), and pALH173 (TurboID-Ty; GenBank: pending).

Between 30 and 50 µg of each vector was linearized with BsaI and co-transfected with 50 µg of the pSS014 Cas9 expression plasmid (GenBank: OM640002). Vectors targeting *TGGT1_289100* (*HOOK*) were transfected into TIR1 or CDPK1-AID. Vectors targeting *TGGT1_264050* (*FTS*) and *TGGT1_306920* were transfected into TIR1. Parasites underwent drug selection for approximately 1 wk in 10% IFS media with 3 µM pyrimethamine or 25 µg/mL mycophenolic acid and 50 µg/mL xanthine, followed by subcloning into 96-well plates. Single clones were screened for tag expression by PCR, immunofluorescence, or immunoblot. PCR validation was performed using the primers P44/P45 (FTS-AID) and P46/P47 (FTS-TurboID).

## Endogenous tagging of HOOK N terminus

AID-HOOK N-terminal tagging was generated by PCR amplifying the gene fragment encoding the *HA-mAID* with homology arms to *HOOK* (IDT gBlock; P17) using the primers P18/19. A sgRNA targeting *HOOK* was assembled into the pSS013 Cas9 expression plasmid (GenBank: OM640003) using the primers P20/P21. Also, 10 µg of the PCR product was transfected with 50 µg of the Cas9

expression plasmid. Parasites were subcloned into 96-well plates. Single clones were screened using PCR primers P42/P43 and validated by immunofluorescence and immunoblot.

## TIR1/pMIC2-mNeonGreen-TurboID-Ty

*pMIC2-mNG-TurboID-Ty* was amplified with primers P24/P25 from pALH184 (pMIC2-mNG-TurboID-Ty-3'DHFR; GenBank: pending) with homology arms to the 5′ and 3′ ends of a defined, neutral genomic locus (*Markus et al., 2019*). Amplified DNA was co-transfected with the Cas9 expression plasmid targeting the neutral locus (pBM041; GenBank: MN019116). mNeonGreen-expressing parasites were isolated by FACS and subcloned in 96-well plates. Expression was confirmed by fluorescence.

## Endogenous tagging of CLAMP

*mNeonGreen* DNA was amplified using PCR from pGL015 (V5-mNG-mAID-Ty; GenBank: OM640005) with homology arms targeting the C terminus of *TGGT1_265790* (*CLAMP*) using the primers P26/P27. A sgRNA targeting *CLAMP* was assembled into the pSS013 Cas9 expression plasmid (GenBank: OM640003) using the primers P28/P29. Between 5 and 10 µg of *mNeonGreen* DNA was co-transfected with 50 µg of the Cas9 expression plasmid targeting *CLAMP* into TIR1 and AID-HOOK. Parasites were grown in 10% IFS media until lysing the HFF monolayer, followed by FACS-mediated isolation of mNeonGreen expressing parasites and subcloning into 96-well plates. Single clones were screened for mNeonGreen expression by PCR using primers P48/P49 and live microscopy.

## Sub-minute phosphoproteomics

### Parasite harvest and treatment

*T. gondii* tachyzoites from the RH strain CDPK1-AID were used to infect nine 15 cm dishes. $3.75 \times 10^7$ parasites were used to infect each dish in 20 mL of media. Approximately 24 hr later, the media of eight dishes was replaced with 15 mL of media containing 1 µM of compound 1 to block egress and synchronize parasites by inhibiting PKG (*Donald et al., 2006*; *Hopp et al., 2012*; *Taylor et al., 2010*). The media of the ninth dish was replaced with media containing DMSO. On day 2, parasites were harvested when the monolayer of HFFs in the DMSO dish were approximately 80% lysed. The eight dishes treated with compound 1 were washed once with 10 mL of warm 1× PBS, once with 30 mL of warm 1× PBS, and incubated with 10 mL of warm FluoroBrite (FluoroBrite DMEM A1896701, 4 mM glutamine, 10 µg/mL gentamicin) for 10 min at 37°C. Infected HFFs were scraped and passed through a 27G needle to mechanically liberate parasites and passed through a 5 µm filter. Parasites were pelleted at $1000 \times g$ at 4°C for 10 min, resuspended in 10 mL of FluoroBrite and a 1:250 dilution was used to count parasites (approximately $2.25 \times 10^9$ total parasites). Parasites were pelleted at $1000 \times g$ at 4°C for 7 min and resuspended in 2 mL of FluoroBrite. Then, 1 mL of parasites were diluted into a total volume of 40 mL of FluoroBrite containing either 500 µM of auxin or vehicle of 1× PBS and incubated at 37°C for 3.5 hr. Then, 50 µL of auxin- or vehicle-treated parasites and untagged TIR1 parasites were analyzed by flow cytometry (Miltenyi MACSQuant VYB) to detect mNeonGreen fluorescence. After confirming depletion of CDPK1, parasites were pelleted at $1000 \times g$ for 10 min at room temperature and resuspended in 210 µL of 500 µM auxin or vehicle of PBS diluted in FluoroBrite. To obtain a time course, the 0 s time point was first collected by mixing 16 µL of parasites with 4 µL of 5× DMSO (0.5% DMSO in FluoroBrite) and immediately lysed with 20 µL of 2× Lysis Buffer (10% SDS, 100 mM TEAB pH 7.5, 2 mM MgCl$_2$, and 2× HALT protease and phosphatase inhibitors). To obtain the 9, 30, and 300 s time points, 80 µL of parasites in 1.5 mL tubes were incubated in a ThermoMixer (Eppendorf) set to 37°C. The parasites were stimulated by adding 20 µL of warmed 5× zaprinast (500 µM zaprinast in FluoroBrite) or a vehicle of 5× DMSO. At each time point, 20 µL of stimulated parasites were transferred directly into 20 µL of 2× Lysis Buffer to quench the reaction. Complete time courses were collected sequentially in the following order: auxin-treated stimulated with zaprinast, vehicle-treated stimulated with zaprinast, auxin-treated stimulated with DMSO, and vehicle-treated stimulated with DMSO. Lysates were treated with benzonase at a final concentration of 5 units/µL to remove DNA and were immediately subjected to protein cleanup and digestion.

## Protein cleanup and digestion

Proteins were prepared for mass spectrometry using a modified version of the S-trap protocol (Protifi). Proteins in lysates were reduced with 5 mM TCEP for 10 min at 55°C and alkylated with 15 mM MMTS for 10 min at room temperature. The lysates were acidified to a final concentration of 1.2% v/v phosphoric acid. A 6× volume of S-trap binding buffer (90% methanol, 100 mM TEAB, pH 7.55) was mixed to each sample to precipitate proteins. The solution was loaded onto S-trap micro columns (Protifi) and spun at 4000 × $g$ for 1 min until all the solution had passed through the column. The columns were washed four times with 150 µL of S-trap binding buffer and centrifuged at 4000 × $g$ for 1 min between each wash. Proteins were digested on-column with 0.75 µg of trypsin (Promega) in 50 mM TEAB pH 8.5 overnight at 37°C in a humidified incubator. Digested peptides were eluted in three steps at 4000 × $g$ for 1 min: 40 µL of 50 mM TEAB, 40 µL of 0.2% formic acid, and 35 µL of 50% acetonitrile/0.2% formic acid. The peptide concentrations of eluted peptides were quantified using the Pierce Fluorometric Peptide Assay (Thermo Fisher Scientific) according to the manufacturer's instructions. The remaining samples were frozen in liquid nitrogen and lyophilized.

## TMTpro labeling

Lyophilized peptides were resuspended in 100 mM TEAB pH 8.5 to peptide concentrations of 1.6 µg/µL. TMTpro reagents (Thermo Fisher Scientific; A44522 LOT# VI306829) were resuspended in acetonitrile to 25 µg/µL. Then, 80 µg of peptides in 50 µL were combined with 250 µg of TMTpro reagent in 10 µL to achieve approximately a 3:1 label:peptide weight/weight ratio (*Zecha et al., 2019*). The TMTpro labels were assigned to minimize reporter ion interference and inter batch variability in the following scheme: replicate 1 auxin: 0, 9, 30, 300 s (126, 128C 130C, 132C); replicate 1 vehicle: 0, 9, 30, 300 s (127N, 129N, 131N, 133N); replicate 2 auxin: 0, 9, 30, 300 s (127C, 129C, 131C, 133C); replicate 2 vehicle: 0, 9, 30, 300 s (128N, 130N, 132N, 134) (*Brenes et al., 2019*). The zaprinast and DMSO samples were incubated for 1 hr at room temperature shaking at 400 rpm. Unreacted TMTpro reagent was quenched with hydroxylamine at a final concentration of 0.2%. The samples were pooled, acidified to 3% with formic acid, and were processed using the EasyPep Maxi Sample Prep column (Thermo Fisher Scientific) according to the manufacturer's instructions. Five percent of the eluate volume was reserved as the unenriched proteome sample. The remaining eluted peptides were frozen in liquid nitrogen and lyophilized.

## Phosphopeptide enrichment

Phosphopeptides were enriched using the Sequential enrichment from Metal Oxide Affinity Chromatography (SMOAC) protocol according to manufacturer instructions (*Tsai et al., 2014*). First, the High-Select TiO$_2$ Phosphopeptide Enrichment Kit (Thermo Fisher Scientific) was used to enrich phosphopeptides from lyophilized TMTpro-labeled samples. The flow-through and contents of the first wash were pooled, frozen in liquid nitrogen, and lyophilized, along with the eluate. Second, the High-Select Fe-NTA Phosphopeptide Enrichment Kit (Thermo Fisher Scientific) was used to enrich phosphopeptides from the pooled flow-through and first wash from the previous enrichment. The eluted phosphopeptides were frozen in liquid nitrogen and lyophilized.

## Fractionation

The enriched and unenriched samples were fractionated with the Pierce High pH Reversed-Phase Peptide Fractionation Kit (Thermo Fisher Scientific) according to the manufacturer's instructions for TMT-labeled peptides. The acetonitrile wash was omitted for enriched samples to prevent loss of phosphopeptides. The eluted peptides from the High-Select TiO$_2$ Phosphopeptide Enrichment and High-Select Fe-NTA Phosphopeptide Enrichment were pooled prior to fractionation. Then, 100 µg of unenriched samples were fractionated. Eight fractions were collected for each TMTpro set: zaprinast phosphoproteome (enriched) [1], zaprinast proteome (unenriched) [2], DMSO phosphoproteome (enriched) [3], and DMSO proteome (unenriched) [4]. The fractions were frozen in liquid nitrogen and lyophilized.

## MS data acquisition

Lyophilized peptides were resuspended in approximately 15 µL (enriched) or 50 µL (unenriched) of 0.1% formic acid and were analyzed on an Exploris 480 Orbitrap mass spectrometer equipped with a FAIMS Pro source (*Bekker-Jensen et al., 2020*) connected to an EASY-nLC 1200 chromatography system using 0.1% formic acid as Buffer A and 80% acetonitrile/0.1% formic acid as Buffer B. Peptides were loaded onto a heated analytical column (ES900, Thermo, PepMap RSLC C18 3 µm, 100 Å, 75 µm × 15 cm, 40°C) via trap column (164946, Thermo, Acclaim PepMap C18 3 µm, 100 Å, 75 µm × 20 mm nanoViper). Peptides were separated at 300 nL/min. Enriched samples were separated on a gradient of 5–20% B for 110 min, 20–28% B for 10 min, 28–95% B for 10 min, 95% B for 10 min, 95–2% B for 2 min, 2% B for 2 min, 2–98% B for 2 min, 98% B for 2 min, 98–2% B for 2 min, and 2% B for 2 min. Unenriched samples were separated on a gradient of 5–25% B for 110 min, 25–40% B for 10 min, 40–95% B for 10 min, 95% B for 10 min, 95–2% B for 2 min, 2% B for 2 min, 2–98% B for 2 min, 98% B for 2 min, 98–2% B for 2 min, and 2% B for 2 min. The orbitrap and FAIMS were operated in positive ion mode with a positive ion voltage of 1800 V; with an ion transfer tube temperature of 270°C; using a standard FAIMS resolution and compensation voltage of –50 and –65 V, an inner electrode temperature of 100°C, and outer electrode temperature 80°C with 4.5 mL/min carrier gas. DDA analysis was performed with a cycle time of 1.5 s. Full-scan spectra were acquired in profile mode at a resolution of 60,000, with a scan range of 400–1400 m/z, 300% AGC target, maximum injection time of 50 ms, intensity threshold of $5 \times 10^4$, 2–5 charge state, dynamic exclusion of 30 s, mass tolerance of 10 ppm, purity threshold of 70%, and purity window of 0.7. MS2 spectra were generated with a HCD collision energy of 32 at a resolution of 45,000 using TurboTMT settings with a first mass at 110 m/z, an isolation window of 0.7 m/z, 200% AGC target, and maximum injection time of 120 ms.

## Phosphoproteomic time-course analysis

Raw files were analyzed in Proteome Discoverer 2.4 (Thermo Fisher Scientific) to generate peak lists and protein and peptide IDs using Sequest HT (Thermo Fisher Scientific) and the ToxoDB release 49 GT1 protein database. The maximum missed cleavage sites for trypsin was limited to 2. Precursor and fragment mass tolerances were 10 ppm and 0.02 Da, respectively. The following modifications were included in the search: dynamic oxidation (+15.995 Da; M), dynamic phosphorylation (+79.966 Da; S,T,Y), dynamic acetylation (+42.011 Da; N-terminus), static TMTpro (+304.207 Da; any N-terminus), static TMTpro (+304.207 Da; K), and static methylthio (+45.988 Da; C). TMTpro 16plex isotope correction values were accounted for (Thermo Fisher Scientific; A44522 LOT# VI306829). Peptides identified in each sample were filtered by Percolator to achieve a maximum FDR of 0.01 (*Käll et al., 2008a*; *Käll et al., 2008bKäll et al., 2007*). Site localization scores were generated using ptmRS, with phosphoRS and use of diagnostic ions set to true (*Taus et al., 2011*). Reporter ion quantification used an integration tolerance of 20 ppm on the most confident centroid. For reporter ion quantification, unique peptides were quantified using a co-isolation threshold of 50 and average reporter signal-to-noise ratio of 10. Abundances were normalized on the total protein amount. Protein level and peptide level ratios were generated for each time point relative to 0 s vehicle-treated parasites stimulated with DMSO. The mass spectrometry proteomics data have been deposited to the ProteomeXchange Consortium via the PRIDE partner repository (*Perez-Riverol et al., 2022*) with the dataset identifier PXD039426 (10.6019/PXD039426). Protein and peptide abundances are reported in *Supplementary files 1–4*.

   Exported peptide and protein abundance files from Proteome Discoverer 2.4 were loaded into R (version 4.1.1). To determine CDPK1-dependent phosphorylation, $\log_2$ ratios for peptide abundances in DMSO and zaprinast-treated samples derived from Proteome Discoverer were used. Only phosphorylated peptides quantified across all time points were used for analysis. Area under the curve (AUC) values were calculated for individual peptides undergoing vehicle ($AUC_{veh}$) and auxin ($AUC_{auxin}$) treatment using trapezoidal integration. AUC differences ($AUC_{diff}$) were calculated by taking the difference between $AUC_{veh}$ and $AUC_{auxin}$ values. The distribution of zaprinast $AUC_{diff}$ values were tested against a null distribution derived from the DMSO $AUC_{diff}$ values to calculate z-scores and p-values using a two-tailed t-test. Replicates 1 and 2 were analyzed independently and phosphopeptides with p-values<0.05 across both replicates were determined to be CDPK1-dependent (Group A). Peptides exhibiting phosphorylation independent of CDPK1 (Groups B–D) were determined similarly, but compared the distribution of zaprinast $AUC_{vehicle}$ values to a null distribution of DMSO $AUC_{vehicle}$ values

and excluded phosphopeptides already determined to be CDPK1-dependent. CDPK1-independent phosphopeptides were clustered into Groups B–D using projection-based clustering (*Thrun and Ultsch, 2021*).

## Gene ontology enrichment

Gene ontology terms were obtained for all genes present in the enriched zaprinast phosphoproteome from ToxoDB.org (Computed GO function and GO function IDs). Gene ontology IDs within each group of zaprinast-dependent genes (Groups A–D) were tested for enrichment against the entire enriched zaprinast phosphoproteome. Enrichment p-values were generated using a hypergeometric test. Enrichment ratios were calculated by dividing the gene ratio (overlap/signatures) by the relative frequency of gene sets (gene sets/background). Only GO IDs with significant enrichment and an overlap of 2 were plotted. Thiophosphorylation enrichment was performed similarly.

## Metabolic tagging, click reaction, pull down, and western blotting

### Metabolic tagging and cell lysis

Upon infection of HFF monolayers the medium was removed and replaced by fresh culture media supplemented with 25 µM YnMyr (Iris Biotech) or Myr (Tokyo Chemical Industry). The parasites were then incubated for 16 hr, washed with PBS (2×), and lysed on ice using a lysis buffer (PBS, 0.1% SDS, 1% Triton X-100, EDTA-free complete protease inhibitor [Roche Diagnostics]). Lysates were kept on ice for 20 min and centrifuged at 17,000 × $g$ for 20 min to remove insoluble material. Supernatants were collected and protein concentration was determined using a BCA protein assay kit (Pierce).

### Click reaction and pulldown

Lysates were thawed on ice. Proteins (100–300 µg) were taken and diluted to 1 mg/mL using the lysis buffer. A click mixture was prepared by adding reagents in the following order and vortexing between the addition of each reagent: a capture reagent (stock solution 10 mM in water, final concentration 0.1 mM), CuSO$_4$ (stock solution 50 mM in water, final concentration 1 mM), TCEP (stock solution 50 mM in water, final concentration 1 mM), and TBTA (stock solution 10 mM in DMSO, final concentration 0.1 mM) (*Heal et al., 2011*). Capture reagent used herein was the Trypsin cleavable reagent (*Broncel et al., 2020*). Following the addition of the click mixture, the samples were vortexed (room temperature, 1 hr), and the reaction was stopped by addition of EDTA (final concentration 10 mM). Subsequently, proteins were precipitated (chloroform/methanol, 0.25:1, relative to the sample volume), the precipitates isolated by centrifugation (17,000 × $g$, 10 min), washed with methanol (1 × 400 µL), and air-dried (10 min). The pellets were then resuspended (final concentration 1 mg/mL, PBS, 0.4% SDS) and the precipitation step was repeated to remove excess of the capture reagent. Next, samples were added to 15 µL of pre-washed (0.2% SDS in PBS [3 × 500 µL]) Dynabeads MyOneTM Streptavidin C1 (Invitrogen) and gently vortexed for 90 min. The supernatant was removed, and the beads were washed with 0.2% SDS in PBS (3 × 500 µL).

### SDS-PAGE and western blotting

Beads were supplemented with 2% SDS in PBS (20 µL) and 4× SLB (Invitrogen), boiled (95°C, 10 min), centrifuged (1000 × $g$, 2 min), and loaded on 10% or 4–20% SDS-PAGE gel (Bio-Rad). Following electrophoresis (60 min, 160 V), gels were briefly washed with water and proteins were transferred (25 V, 1.3 A, 7 min) onto nitrocellulose membranes (Bio-Rad) using Bio-Rad Trans Blot Turbo Transfer system. After brief wash with PBS-T (PBS, 0.1% Tween-20), membranes were blocked (5% milk, TBS-T, 1 hr) and incubated with primary antibodies (5% milk, TBS-T, overnight, 4°C) at the following dilutions: rat anti-HA (1:1000; Roche Diagnostics), mouse anti-Myc (1:1000; Millipore), rabbit anti-Gra29 (1:1000; Moritz Treeck Lab), rabbit anti-SFP1 (1:1000; Moritz Treeck Lab), mouse anti-*T. gondii* [TP3] (1:1000; Abcam), mouse anti-CDPK1 (1:3000; John Boothroyd Lab), rabbit anti-SAG1 (1:10,000; John Boothroyd Lab), rabbit anti-GAP45 (1:1000; Peter Bradley Lab), mouse anti-GFP (1:1000, Roche Diagnostics), and rabbit anti-mCherry (1:1000, Abcam). Following washing (TBS-T, 3×), membranes were incubated with IR dye-conjugated secondary antibodies (1:10,000, 5% milk, TBS-T, 1 hr) and after a final washing step imaged on a LI-COR Odyssey imaging system (LI-COR Biosciences).

## MS detection of myristoylated CDPK1

Mass spectrometry proteomics methods and data for myristoylated CDPK1 are available from the ProteomeXchange Consortium via the PRIDE partner repository (ID PXD019677) and the associated publication (*Broncel et al., 2020*).

## Depletion of mAID tagged CDPK1 (iKD)

Parasites were treated with 500 µM auxin or equivalent volume of vehicle (ethanol) for at least 2 hr prior to western blot analysis.

## Subcellular fractionation

RH Δ*ku80*Δ*hxgprt* YFP expressing parasites were metabolically tagged with 25 µM Myr or YnMyr for 16 hr. Following a PBS wash, the parasites were syringe lysed in Endo buffer (44.7 mM $K_2SO_4$, 10 mM $MgSO_4$, 106 mM sucrose, 5 mM glucose, 20 mM Tris–$H_2SO_4$, 3.5 mg/mL BSA, pH 8.2) and collected by centrifugation (512 × *g*, 10 min). The parasites were then lysed in 300 µL of cold hypotonic buffer (10 mM HEPES, pH 7.5) supplemented with protease inhibitors (Roche), passed through 25G needle (5×) and left on ice for 40 min. Next, lysates were pelleted by centrifugation (16,000 × *g*, 20 min, 4°C) and the resulting supernatant was subjected to an additional high-speed (100,000 × *g*, 1 hr, 4°C) centrifugation step. To avoid the loss of the high-speed pellet, only half of the supernatant was removed at this point. Each fraction was then taken up in 0.4% (final) SDS HEPES, clicked to a capture reagent, and pulled down as described above. Myristoylation-dependent partitioning was revealed by SDS-PAGE and western blotting.

Myristoylation-dependent fractionation for CDPK1 complemented WT and Mut lines: parasites were seeded 24 hr prior experiment. Following a PBS wash, the parasites were syringe lysed in Endo buffer (44.7 mM $K_2SO_4$, 10 mM $MgSO_4$, 106 mM sucrose, 5 mM glucose, 20 mM Tris–$H_2SO_4$, 3.5 mg/mL BSA, pH 8.2) and collected by centrifugation (512 × *g*, 10 min). The parasites were then lysed in 300 µL of cold hypotonic buffer (10 mM HEPES, pH 7.5) supplemented with protease inhibitors (Roche), passed through 25G needle (5×) and left on ice for 40 min. Next, lysates were pelleted by centrifugation (100,000 × *g*, 1 hr, 4°C), the supernatant was removed, and cytosolic proteins precipitated with methanol/chloroform. Proteins from the pellet and supernatant fractions were dissolved in 2% (final) SDS PBS and myristoylation-dependent partitioning was revealed by SDS-PAGE and western blotting.

## Immunoprecipitation of cWT and cMut CDPK1

### Parasite harvest

cWT, cMut, and TIR1 (untagged) parasites were infected onto confluent HFFs in 15 cm dishes. At 1 day post-infection (dpi), 50 µM auxin or vehicle was added to deplete endogenous mAID-tagged CDPK1 and 1 µM compound 1 was added to block egress until parasites were ready to harvest. At 2 dpi, infected HFFs were washed twice with PBS to wash out drugs. Parasites were mechanically released in Endo buffer with a 27G needle, passed through a 5 µm filter, and spun at 1000 × *g* for 10 min. Parasite pellets were resuspended in a cold hypotonic buffer with 1× HALT protease inhibitors to parasite concentrations of $1.1 \times 10^9$ tg/mL, passed through a 27G needle five times, and incubated on ice for 1 hr to complete hypotonic lysis. The samples were spun at 1000 × *g* for 5 min to pellet unlysed parasites and the supernatant was saved. NaCl was added to a final concentration of 150 mM, and this was used as the immunoprecipitation input.

### Immunoprecipitation

25 µL of anti-HA magnetic beads (Thermo) were used per condition. Beads were washed twice with a wash buffer (10 mM HEPES, 150 mM NaCl, pH 7.5). To begin pulldown, parasite lysate was used to resuspend washed beads and incubated for 1 hr rotating at room temperature. Beads were washed four times with a wash buffer. Proteins were eluted by resuspending beads in 20 µL of 1× S-trap sample buffer (5% SDS, 50 mM TEAB, pH 7.5) and incubated at 70°C for 10 min. The eluate was collected for MS sample processing and analysis. Results are representative of two independent experiments.

## Protein cleanup and digestion

Proteins were prepared for mass spectrometry as described above in 'Sub-minute phosphoproteomics—protein cleanup and digestion.' Eluted peptides were frozen in liquid nitrogen, lyophilized, and stored at –80°C until MS analysis.

## MS data acquisition

Lyophilized peptides were resuspended in 20 µL of 0.1% formic acid and were analyzed on an Exploris 480 Orbitrap mass spectrometer equipped with a FAIMS Pro source (*Bekker-Jensen et al., 2020*) connected to an EASY-nLC chromatography system using 0.1% formic acid as Buffer A and 80% acetonitrile/0.1% formic acid as Buffer B. Peptides were separated at 300 nL/min on a gradient of 1–6% B for 1 min, 6–21% B for 41 min and 30 s, 21–36% B for 20 min and 45 s, 36–50% B for 10 min and 15 s, 100% B for 14 min and 30 s, 100–2% B for 3 min, 2% B for 3 min, 2–98% B for 3 min, and 98% B for 3 min. The orbitrap and FAIMS were operated in positive ion mode with a positive ion voltage of 1800 V; with an ion transfer tube temperature of 270°C; using a standard FAIMS resolution and an inner and outer electrode temperature of 100°C with 4.5 mL/min carrier gas. Samples were analyzed twice in DIA mode with a compensation mode of –50 and –65 V. Full-scan spectra were acquired in profile mode at a resolution of 120,000, with a scan range of 400–1000 m/z, 300% AGC target, and auto mode for maximum injection time. MS2 spectra for the DIA scan were generated with a isolation window of 20 m/z with a 0 m/z window overlap, 30 scan events, an HCD collision energy of 30 at a resolution of 30,000, first mass at 200 m/z, precursor mass range of 400–1000 m/z, and a standard AGC target and automatically determined maximum injection time.

## Immunoprecipitation data analysis

DIA-MS samples were analyzed using Scaffold DIA (2.0.0). DIA-MS data files were converted to mzML format using ProteoWizard (3.0.19254) (*Chambers et al., 2012*). Analytic samples were aligned based on retention times and individually searched against dku80_FAIMS_DIA_90min_autoIT.blib with a peptide mass tolerance of 10 ppm and a fragment mass tolerance of 0.02 Da. The ToxoDB release 46 GT1 protein database was used for protein identification. The following modifications were included in the search: dynamic oxidation (+15.995 Da; M), dynamic phosphorylation (+79.966 Da; S,T,Y), and static methylthio (+45.988 Da; C). The digestion enzyme was trypsin with a maximum of two missed cleavage sites allowed. Only peptides with charges in the range of 2–3 and length in the range 6–30 were considered. Peptides identified in each sample were filtered by Percolator to achieve a maximum FDR of 0.01 (*Käll et al., 2008a*; *Käll et al., 2007*). Individual search results were combined, and peptide identifications were assigned posterior error probabilities and filtered to an FDR threshold of 0.01 by Percolator. Peptide quantification was performed by Encyclopedia (0.9.2). For each peptide, the five highest quality fragment ions were selected for quantification. Proteins that contained similar peptides and could not be differentiated based on MS/MS analysis were grouped to satisfy the principles of parsimony. Protein groups were thresholded to achieve a protein FDR < 1%. Significance values were derived from *t*-tests across two replicates and adjusted with Benjamini–Hochberg correction with an FDR of 0.05. Exported protein abundance files from Scaffold DIA were loaded into R (version 4.1.1). The mass spectrometry proteomics data have been deposited to the ProteomeXchange Consortium via the PRIDE partner repository (*Perez-Riverol et al., 2022*) with the dataset identifier PXD04408. Protein abundances are reported in *Supplementary file 5*.

## Thiophosphorylation of CDPK1 substrates

### Parasite harvest and treatment

*T. gondii* tachyzoites from the RH strain (CDPK1[G] and CDPK1[M]) were passaged twice across 4 d in SILAC media in T12.5 flasks. CDPK1[G] parasites were grown in 'heavy' SILAC media (DMEM 88364, 10% dialyzed FBS, 0.1 mg/mL $^{13}C^{15}N$ L-arginine, 0.1 mg/mL $^{13}C^{15}N$ L-lysine) and CDPK1[M] parasites were grown in 'light' SILAC media (DMEM 88364, 10% dialyzed FBS, 0.1 mg/mL L-arginine, 0.1 mg/mL L-lysine). On day 4, confluent HFFs in 15 cm dishes were infected with CDPK1[G] and CDPK1[M] parasites in SILAC media. On day 6, extracellular parasites were harvested by filtering through a 5 µm filter and pelleted at 1000 × *g* for 7 min at 4°C. Parasites were washed once in 1× intracellular buffer (ICB) (137 mM KCl, 5 mM NaCl, 20 mM HEPES, 10 mM MgCl₂, pH 7.5 KOH) and then resuspended in

400 µL of 1× ICB. Parasites were semi-permeabilized after the addition of 400 µL of 6 µg/mL aerolysin diluted in 1× ICB and incubated at 37°C for 10 min. After semi-permeabilization, 400 µL of 4× Ca$^{2+}$ solution (16 mM CaEGTA, 100 ng/mL 1B7 nanobody in 1× ICB) followed by 400 µL of 4× ATP solution (4 mM GTP, 0.4 mM ATP, 0.2 mM KTPγS, 1× HALT protease and phosphatase inhibitor in 1× ICB). The kinase reaction was initiated by incubating parasites at 30°C for 5 min. Parasites were pelleted at 1000 × $g$ at 4°C for 10 min. Parasite pellets were resuspended in 250 µL of 1× lysis buffer (10% TritonX-100, 1× HALT protease and phosphatase inhibitor, 10 mM K$_2$EGTA in 1× ICB).

## Protein quantification and thiophosphorylation immunoblotting

Proteins in lysates were quantified using DC assay (Bio-Rad) utilizing BSA as a protein standard and a diluent (150 mM NaCl, 20 mM Tris pH 7.6) to prepare standard curves and dilution series. Thiophosphorylation was verified using immunoblot by first incubating 3 µL of sample with p-nitrobenzyl mesylate diluted in 1× ICB at a final concentration at 2 mM for 2 hr at room temperature. A 5× Laemmli sample buffer was added (see 'Immunoblotting' for recipe) and samples were boiled for 10 min. Samples were resolved on a home-made polyacrylamide gel (5% stacking, 15% resolving, 15-well, 0.75 mm) and transferred overnight at 4°C. Nitrocellulose membranes were blocked with 5% milk in TBS-T for 1 hr. Primary antibody incubations were performed with an anti-thiophosphate ester antibody (rabbit 51-8; 1:5000) and anti-tubulin (mouse 12G10; 1:2000) for 1 hr at room temperature. Secondary incubations were performed with LI-COR antibodies (rabbit 680, mouse 800; 1:10,000) for 1 hr at room temperature. Imaging was performed using a LI-COR Odyssey.

## Protein cleanup and digestion

Proteins were precipitated using methanol chloroform extraction. For 250 µL of lysate, 800 µL methanol and 200 µL of chloroform were added, followed by vortexing. After adding 600 µL of water, the sample was vortexed and centrifuged at max speed for 5 min at 4°C. After removing the supernatant without disrupting the precipitate, 375 µL of methanol was added and vortexed. Samples were centrifuged at max speed for 15 min at 4°C. The protein pellet was allowed to air dry after removing the supernatant. Dried protein pellets were resuspended in 200 µL of 8 M urea. Then, 1 mg of protein (determined from protein quantification; see above) from CDPK1$^G$ and CDPK1$^M$ parasites were pooled for a total of 2 mg of protein. To digest protein, 5× volume of 1× Trypsin Digest Buffer (100 mM ammonium acetate pH 8.9, 1 mM CaCl$_2$, 2 mM TCEP) was added to the sample followed by 40 µg of sequencing grade trypsin (Promega). Proteins were digested overnight rotating at room temperature. To prepare samples for desalting, glacial acetic acid was added to 10% (v/v) and debris was briefly spun down. A C18 Sep-Pak Plus cartridge (Waters) was prepared using a syringe pump and 10 mL syringe with three 10 mL washes at a flow rate of 2 mL/min: 0.1% acetic acid, 90% acetonitrile/0.1% acetic acid, and 0.1% acetic acid. The sample was loaded in a 5 mL syringe at a flow rate of 0.5 mL/min. The syringe and cartridge were washed with 5 mL of 0.1% acetic acid at a flow rate of 0.5 mL/min. A final wash was performed using 10 mL of 0.1% acetic acid at a flow rate of 2 mL/min. Peptides were eluted with 4.5 mL of 40% acetonitrile/0.1% acetic acid at a flow rate of 0.5 mL/min. Samples were spun in a speed vac for 3 hr until samples could be pooled into a single 2 mL tube. Samples were frozen in liquid nitrogen and lyophilized overnight. Lyophilized peptides were stored at –80°C.

## Thiophosphate enrichment

To enrich for thiophosphorylated peptides, 400 µL of SulfoLink Coupling Resin slurry was used (Thermo). Incubations were performed in the dark using aluminum foil due to the light sensitivity of the resin. The resin was washed twice with 1 mL of binding buffer (25 mM HEPES, 50% acetonitrile, pH 7.0 NaOH + HCl) rotating for 5 min followed by a 1000 rpm spin for 10 s. The supernatant was removed without disturbing the pelleted resin. The beads were blocked with 1 mL of blocking buffer (binding buffer with 25 µg/mL β-casein and 2 mM TCEP) and rotated for 5 min at room temperature. After pelleting the resin and removing the supernatant, the resin was resuspended with lyophilized peptides dissolved in 500 µL of the blocking buffer. Samples were incubated overnight at room temperature. Samples were spun at 1000 rpm for 10 s and the supernatant was spun again, lyophilized, and stored at –80°C prior to MS analysis. The beads received the following series of 1 mL washes for 5 min rotating and spun at 1000 rpm for 10 s: twice with binding buffer with 2 mM TCEP, once with quenching buffer (25 mM HEPES, 50% acetonitrile, 5 mM DTT, pH 8.5 NaOH), once with

binding buffer with 2 mM TCEP, once with 5% formic acid (no rotation), once with binding buffer with 2 mM TCEP, and three times with 0.1% acetic acid. The resin was no longer light sensitive after incubation with the quenching buffer. To elute captured peptides, resin was resuspended in 500 μL OXONE (2 mg/mL potassium monopersulfate) and rotated for 5 min at room temperature. The supernatant containing eluted peptides was collected after spinning at 1000 rpm for 10 s. OXONE was removed from the sample using C18 spin columns (Pierce) according to the manufacturer's instructions. A total of four washes with the equilibrium/wash buffer was performed. Peptides were spun in a speed vac until dry and stored at –80°C until MS analysis.

## MS data acquisition

The samples were resuspended in 10–20 μL of 0.1% formic acid for MS analysis and were analyzed on a Q-Exactive HF-X Orbitrap mass spectrometer connected to an EASY-nLC 1200 chromatography system using 0.1% formic acid as Buffer A and 80% acetonitrile/0.1% formic acid as Buffer B. Peptides were loaded onto a analytical column (column: PF360-75-15-N-5, New Objective, 360 μm OD, 75 μm ID, 15 μm Tip PicoFrit Emitter; resin: 04A-4506, Phenomenex Aeris Peptide, C18 1.7 μm) via trapping column (column: 360 um OD, 100 um ID; resin: AA12S11, YMC Gel ODS-A, C18 10 μm). Peptides were separated at 300 nL/min on a gradient of 2% B for 5 min, 2–25% B for 100 min, 25–40% B for 20 min, 40–100% B for 1 min, and 100% B for 12 min. The orbitrap was operated in positive ion mode with a positive ion voltage of 2700 V with an ion transfer tube temperature of 300°C. Full-scan spectra were acquired in profile mode at a resolution of 60,000, with a scan range of 375–1600 m/z, $1 \times 10^6$ AGC target, maximum injection time of 50 ms, intensity threshold of $4 \times 10^5$, dynamic exclusion of 13 s, and 20 data-dependent scans (DDA Top 20). MS2 spectra were generated with an HCD collision energy of 27 at a resolution of 15,000, first mass at 100 m/z, isolation window of 1.5 m/z, an AGC target of $1 \times 10^5$ with a maximum injection time of 20 msec, and scan range of 200–2000 m/z.

## Thiophosphorylation analysis

Raw files were analyzed in Proteome Discoverer 2.2 (Thermo Fisher Scientific) to generate peak lists and protein and peptides identifications using Sequest HT (Thermo Fisher Scientific) and the ToxoDB release 34 GT1 protein database. The maximum missed cleavage sites for trypsin was limited to 2. The following modifications were included in the search: dynamic oxidation (+15.995 Da; M), dynamic phosphorylation (+79.966 Da; S,T,Y), dynamic $^{13}C_6{}^{15}N_4$ (+10.008 Da; R), dynamic $^{13}C_6{}^{15}N_2$ (+8.014 Da; K), and dynamic acetylation (+42.011 Da; N-terminus). Site localization scores were generated using ptmRS, with phosphoRS and use of diagnostic ions set to true. SILAC 2plex (Arg10, Lys8) method was used for relative quantification of protein and unique peptide abundances. For peptide level analysis, ratios of unique peptides comparing CDPK1$^G$ and CDPK1$^M$ were generated and low abundance resampling (5%) was used to impute missing values. For protein level analysis, ratios of proteins were determined from the summed abundance of unique peptides comparing CDPK1$^G$ and CDPK1$^M$ strains. The mass spectrometry proteomics data have been deposited to the ProteomeXchange Consortium via the PRIDE partner repository (*Perez-Riverol et al., 2022*) with the dataset identifier PXD039431 (10.6019/PXD039431). Protein and peptide abundances are reported in *Supplementary files 6 and 7*.

Peptide and protein level data were exported from Proteome Discoverer for analysis in R. For enriched peptides, abundance ratios were calculated within each replicate for high-confidence peptides (CDPK1$^G$/CDPK1$^M$) and normalized to the median abundance ratio of the whole-proteome peptides derived from the flow-through samples. Normalized abundance ratios in the flow-through proteome samples were calculated by dividing abundance ratios by the median abundance ratios within each replicate. For enriched peptides, an average $\log_2$ normalized abundance ratio was calculated from three replicates. Significantly enriched peptides were calculated using a one-tailed *t*-test using the normalized abundance ratios for three replicates. A nonlinear significance threshold was calculated using the function ($y = |4/x|$), where significantly enriched peptides had product value of $-\log_{10}(p) *$ (mean $\log_2$ abundance ratio) > 4.

## Immunoblotting

Samples were combined with 5× Laemmli sample buffer (10% SDS, 50% glycerol, 300 mM Tris HCl pH 6.8, 0.05% bromophenol blue, 5% beta-mercaptoethanol) and were incubated at 95°C for 10 min. The samples were run on precast 4–15% or 7.5% SDS gels (Bio-Rad) and were transferred overnight

onto a nitrocellulose membrane in transfer buffer (25 mM Tris–HCl, 192 mM glycine, 0.1% SDS, 20% methanol) at 4°C. Blocking was performed with 5% milk in PBS for 1 hr rocking at room temperature. Antibody incubations were performed with 5% milk in TBS-T for 1 hr rocking at room temperature. Three 5 min TBS-T washes were performed before and after secondary antibody incubations rocking at room temperature. After a final PBS wash, imaging was performed using a LI-COR Odyssey.

For immunoblot detection of the HA tag in AID-HOOK and FTS-AID after auxin-mediated depletion, secondary antibody incubations were performed with anti-rabbit HRP antibodies (Jackson ImmunoResearch) and detected with chemiluminescence (Azure) for increased sensitivity. Imaging was performed using a Bio-Rad Gel Doc XR.

## Immunofluorescence analysis

HFFs were seeded onto coverslips and grown until confluence. Confluent HFFs were infected with parasites. Approximately 2 hr later, media was exchanged with media containing either 50 µM auxin or vehicle solution of PBS. At 24 hr post-infection, the media was aspirated, and the coverslips were washed with PBS three times before fixation with 4% formaldehyde in PBS for 20 min. Following three washes in PBS, the fixed cells were permeabilized with 0.25% Triton X-100 for 15 min. After three washes in PBS, the coverslips were incubated in a blocking solution (5% IFS/5% NGS in PBS) for 10 min at room temperature. Coverslips were incubated for 1–2 hr with primary antibody diluted in blocking solution. Anti-GAP45 or anti-MIC2 was used as a parasite counterstain. After three washes in PBS, the coverslips were incubated in blocking solution for 5 min, followed by secondary antibody diluted in blocking solution containing Hoechst 33342 for 1 hr. The coverslips were washed three times in PBS, once in water, and finally mounted with Prolong Diamond overnight at room temperature. Microscope images were acquired with the Nikon Ti Eclipse and NIS Elements software package.

### Immunofluorescence analysis (*Figure 2*)

Parasite-infected HFF monolayers grown on glass coverslips were fixed with 3% formaldehyde for 15 min prior to washing with PBS. Fixed cells were then permeabilized (PBS, 0.1% Triton X-100, 10 min), blocked (3% BSA in PBS, 1 hr), and labeled with anti-HA (1:1000, 1 hr; Roche). HA-tagged CDPK1 in the cWT and cMut lines was visualized with secondary goat antibodies (1:2000, 1 hr; Life Technologies) conjugated to Alexa Fluor 594 and 488, respectively. Cytosolic GFP (cWT) and mCherry (cMut) were used as parasite counterstains. Nuclei were visualized with the DNA stain (DAPI; Sigma) added at 5 µg/mL with the secondary antibody. Stained coverslips were mounted on glass slides with Slowfade (Life Technologies) and imaged on a Nikon Eclipse Ti-U inverted fluorescent microscope. Images were analyzed using Nikon NIS Elements imaging software.

## Plaque assays

A total of 600 and 1200 parasites were used to infect 6-well plates of HFFs in 10% IFS media. At 1 dpi, media was exchanged with 10% IFS media containing either 50 µM auxin or vehicle solution of PBS. Parasites were allowed to grow undisturbed for 8 d total. Plates were washed with PBS and fixed for 10 min with 100% ethanol at room temperature. After removing ethanol, plates were allowed to dry prior to staining with crystal violet solution for 30 min to 1 hr. Plates were washed three times with PBS, once with water, and allowed to dry before scanning.

### Plaque assays (*Figure 2*)

Parasites were harvested by syringe lysis, counted, and 200 parasites were seeded on confluent HFF monolayers grown in 24-well plates (Falcon). Wells were treated with 500 µM auxin or vehicle (ethanol) and plaques were allowed to form for 5 d. Plaque formation was assessed by inspecting the methanol fixed and 0.1% crystal violet stained HFF monolayers.

## Invasion assays

Confluent HFFs seeded in T12.5 flasks were infected with parasites. Approximately 2 hr later, the media was exchanged for 10% IFS media containing either 50 µM auxin or vehicle solution of PBS. Parasites were harvested at 2 dpi and the media was exchanged for 1% IFS in invasion media (DMEM Sigma D2902, 20 mM HEPES, pH 7.4) with auxin or vehicle PBS added. Confluent HFFs seeded in clear-bottomed 96-well plates were infected with $2 \times 10^5$ extracellular parasites. After centrifuging

the plate at 290 × *g* for 5 min to synchronize invasion, the infected 96-well plate was incubated by floating on a water bath at 37°C/5% $CO_2$ for 90 min. Wells were washed once with PBS before fixing infected HFFs with 4% formaldehyde in PBS for 20 min at room temperature. Wells were washed three times with a wash buffer (1% NGS in PBS) and then incubated in a blocking buffer (5% IFS and 1% NGS in PBS) overnight at 4°C. To stain extracellular parasites, wells were incubated with anti-SAG1 antibody for 30 min at room temperature. After washing wells three times with a wash buffer, fixed HFFs were permeabilized in 0.25% TritonX-100 in blocking buffer for 8 min room temperature. After washing the wells three times with a wash buffer, the wells were incubated in a blocking buffer for 10 min at room temperature, followed by anti-GAP45 antibody for 30 min at room temperature to label all parasites. After three washes in a wash buffer, wells were incubated with a secondary antibody solution containing Hoechst for 30 min. After three washes in PBS, wells were imaged using a Biotek Cytation 3.

## Egress assays

Confluent HFFs seeded in glass-bottomed 35 mm dishes (Ibidi or Mattek) were infected with approximately $2 \times 10^5$ parasites. Approximately 2 hr later, the media was exchanged for 10% IFS media containing either 50 µM auxin or vehicle solution of PBS. At 24 hr post-infection, the media was replaced with 1% IFS in Ringer's (155 mM NaCl, 2 mM $CaCl_2$, 3 mM KCl, 1 mM $MgCl_2$, 3 mM $NaH_2PO_4$, 10 mM HEPES, 10 mM glucose, pH 7.4). Zaprinast (final concentration 500 µM) was added into the dishes after 15 s of imaging and imaging of infected HFFs continued for a total of 10 min. Images were acquired using a Nikon Ti Eclipse with an enclosure maintained at 37°C. The number of intracellular vacuoles was quantified at 0 min and 10 min. Results are the sum of four fields of view per condition and are representative of three independent experiments.

### Egress assays (*Figure 2*)

Parasites were added to HFF monolayer and grown for 24 hr in a 96-well plate. The wells were then treated with 500 µM auxin or an equivalent volume of vehicle (ethanol) for 2 hr and then washed with PBS (2×). The media was exchanged for 100 µL Ringer's solution (155 mM NaCl, 3 mM KCl, 2 mM $CaCl_2$, 1 mM $MgCl_2$, 3 mM $NaH_2PO_4$, 10 mM HEPES, 10 mM glucose) and the plate was placed on a heating block to maintain the temperature at 37°C. To artificially induce egress, 50 µL of Ringer's solution containing 24 µM ionophore (8 µM final, Thermo) was added to each well. At specified time points, the cells were fixed by adding 33 µL 16% formaldehyde (3% final) for 15 min. Cells were washed in PBS (3×) and stained with rabbit anti-TgCAP 1:2000 (*Hunt et al., 2019*) followed by goat anti-rabbit Alexa Fluor 488 (1:2000) and DAPI (5 µg/mL). Automated image acquisition of 25 fields per well was performed on a Cellomics Array Scan VTI HCS reader (Thermo Scientific) using a 20× objective. Image analysis was performed using the Compartmental Analysis BioApplication on HCS Studio (Thermo Scientific). Egress levels were determined in triplicate for three independent assays. Vacuole counts were normalized to *t* = 0 to determine how many intact vacuoles had remained after egress. The results were statistically tested using one-way ANOVA with Tukey's multiple-comparison test in GraphPad Prism 7. The data are presented as mean ± SD.

## Replication assays

Confluent HFFs seeded on coverslips were infected with parasites. Parasites were centrifuged at 290 × *g* for 5 min to synchronize invasion of host cells. Approximately 2 hr later, the media was exchanged for 10% IFS media containing either 50 µM auxin or vehicle solution of PBS. At 24 hr post-infection, the media was aspirated and fixed as described in 'Immunofluorescence analysis.' Coverslips were incubated in primary and secondary antibodies for 30 min. Anti-GAP45 antibody was used to visualize parasites. Microscope images were acquired with the Nikon Ti Eclipse and NIS Elements software package. Tiled images were acquired in a four-by-four manner using a 40× objective. Parasites per vacuole were quantified for at least 100 vacuoles for three biological replicates.

## Immunoprecipitation of HOOK, FTS, and TGGT1_306920 complex

### Parasite harvest

HOOK-3xHA, CDPK1-AID (parental), FTS-3xHA, 306920-3xHA, and TIR1 (parental) parasites were infected onto confluent HFFs in 15 cm dishes. After the parasites lysed the HFF monolayer

(approximately 40 hr post-infection), extracellular parasites were passed through 5 µm filters and washed twice with DMEM by pelleting at 1000 × $g$ for 10 min. Parasite pellets were resuspended in 1× Dynein Lysis Buffer (DLB) (30 mM HEPES, 50 mM KOAc1, 2 mM MgOAc, 10% glycerol, 1 mM EGTA pH 8.0, 1 mM DTT, 0.5 mM ATP, 125 units/mL benzonase, 1× HALT protease and phosphatase inhibitor, and 1% NP-40 IGEPAL CA 630) to achieve a concentration of 6.67 × $10^8$ parasites/mL (*Redwine et al., 2017*). Parasites were lysed on ice for 10 min and vortexed periodically to facilitate lysis. Lysates were centrifuged at 1000 × $g$ for 5 min at room temperature to pellet unlysed parasites and the lysate supernatant was collected and used as the immunoprecipitation input.

## Immunoprecipitation
25 µL of anti-HA magnetic beads (Thermo Scientific 88836) was used for 200 µL of parasite lysate. Beads were aliquoted and washed three times with a wash buffer (30 mM HEPES, 50 mM KOAc1, 2 mM MgOAc, 10% glycerol, 1 mM EGTA pH 8.0, 1 mM DTT, 0.5 mM ATP, and 0.01% NP-40 IGEPAL CA 630) using a magnetic rack. To begin the pulldown, 200 µL of parasite lysate was used to resuspend washed beads. Tubes were rotated at 4°C for 1 hr (FTS-AID and TIR1) and 3 hr (HOOK-3xHA and CDPK1-AID). Beads were resuspended in 250 µL of wash buffer, transferred to a new tube, and received two additional washes with the same volume. Proteins were eluted by resuspending beads in 22 µL of 1× S-trap sample buffer (5% SDS, 50 mM TEAB, pH 7.5) and incubated at 70°C for 10 min. The eluate was collected for MS sample processing and analysis. Results are representative of three independent experiments.

## Protein cleanup and digestion
Proteins were prepared for mass spectrometry as described above in 'Sub-minute phosphoproteomics—protein cleanup and digestion.' Eluted peptides were frozen in liquid nitrogen, lyophilized, and stored at –80°C until MS analysis.

## MS data acquisition
Lyophilized peptides were resuspended in 25 µL of 0.1% formic acid and were analyzed on an Exploris 480 Orbitrap mass spectrometer equipped with a FAIMS Pro source (*Bekker-Jensen et al., 2020*) connected to an EASY-nLC chromatography system using 0.1% formic acid as Buffer A and 80% acetonitrile/0.1% formic acid as Buffer B. Peptides were separated at 300 nL/min on a gradient of 1–6% B for 1 min, 6–21% B for 41 min and 30 s, 21–36% B for 20 min and 45 s, 36–50% B for 10 min and 15 s, 100% B for 14 min and 30 s, 100–2% B for 3 min, 2% B for 3 min, 2–98% B for 3 min, and 98% B for 3 min. The orbitrap and FAIMS were operated in positive ion mode with a positive ion voltage of 1800 V; with an ion transfer tube temperature of 270°C; using a standard FAIMS resolution and compensation voltage of –50 and –65 V, an inner and outer electrode temperature of 100°C with 4.5 mL/min carrier gas. Full-scan spectra were acquired in profile mode at a resolution of 60,000, with a scan range of 350–1400 m/z, 300% AGC target, 25 ms maximum injection time, intensity threshold of 5 × $10^3$, 2–6 charge state, dynamic exclusion of 20 s, 15 data-dependent scans (DDA Top 15), and mass tolerance of 10 ppm. MS2 spectra were generated an HCD collision energy of 30 at a resolution of 15,000, first mass at 110 m/z, with an isolation window of 1.3 m/z, and an automatically determined AGC target and maximum injection time in standard and auto mode.

## Immunoprecipitation MS analysis
Raw files were analyzed in Proteome Discoverer 2.4 (Thermo Fisher Scientific) to generate peak lists and protein and peptides identifications using Sequest HT (Thermo Fisher Scientific) and the ToxoDB release 49 GT1 protein database. The maximum missed cleavage sites for trypsin was limited to 2. The following modifications were included in the search: dynamic oxidation (+15.995 Da; M), dynamic phosphorylation (+79.966 Da; S,T,Y), dynamic acetylation (+42.011 Da; N-terminus), and static methylthio (+45.988 Da; C). Label-free quantification of proteins was performed using summed abundances from unique peptides. Abundances were normalized on the total peptide amount. Pairwise ratios were calculated for protein abundances comparing strains and conditions. Significance values were derived from *t*-tests across three replicates and adjusted with Benjamini–Hochberg correction. Exported protein abundance files from Proteome Discoverer 2.4 were loaded into R (version 4.1.1).

The mass spectrometry proteomics data have been deposited to the ProteomeXchange Consortium via the PRIDE partner repository (*Perez-Riverol et al., 2022*) with the dataset identifier PXD039432 (10.6019/PXD039432) for HOOK and FTS pulldowns and PXD044080 (10.6019/PXD044080) for TGGT1_306920 pulldowns. Protein abundances are reported in *Supplementary file 8*.

## Proximity labeling FTS-TurboID

### Parasite harvest and treatment

FTS-TurboID and mNG-TurboID (cytosolic control) parasites were infected onto confluent HFFs in 15 cm dishes. After the parasites lysed the HFF monolayer (approximately 40 hr post-infection), extra-cellular parasites were passed through 5 µm filters and washed twice with DMEM by pelleting at 1000 × $g$ for 10 min. At least 1 × $10^8$ parasites were resuspended in 200 µL of DMEM. 200 µL of biotin (final concentration 500 µM) or vehicle DMSO in DMEM was mixed with parasites. Tubes were incubated in a water bath at 37°C for 5 min. Parasites were pelleted at 12,000 × $g$ for 1 min at 4°C for a total of three 1 mL PBS washes. Parasites were resuspended in a RIPA NP-40 lysis buffer (10 mM Tris-HCl pH 7.5, 140 mM NaCl, 1% NP-40 IGEPAL CA 630, 0.1% sodium deoxycholate, 0.1% SDS, 1× HALT protease inhibitor, and 125 units/mL benzonase) at a parasite concentration of 5 × $10^8$ parasites/mL. Parasites were lysed at room temperature for 15 min and stored at –20°C. Results are representative of three independent experiments.

### Biotinylated protein enrichment

Streptavidin magnetic beads were washed three times with a RIPA NP-40 lysis buffer. Thawed lysates were spun at 16,000 × $g$ for 5 min at 4°C and the supernatant was used as the pulldown input. Then, 20 µL of streptavidin magnetic beads was used for 200 µL of lysates (1 × $10^8$ parasites). Beads were incubated with lysate rotating for 1 hr. Beads received a series of 1 mL washes: twice with the RIPA NP-40 lysis buffer, once with 1 M KCl, once with 0.1 M $Na_2CO_3$ (pH 11), once with 2 M urea in 10 mM Tris pH 8.0, and twice with RIPA NP-40 lysis buffer. Proteins were eluted by incubating beads with 2.5 mM biotin in S-trap sample buffer (5% SDS, 50 mM TEAB, pH 7.5) for 10 min at 95°C.

### Protein cleanup and digestion

Proteins were prepared for mass spectrometry as described above in 'Sub-minute phosphopro-teomics—protein cleanup and digestion.' Eluted peptides were frozen in liquid nitrogen, lyophilized, and stored at –80°C until MS analysis.

### MS data acquisition

Lyophilized peptides were resuspended in 25 µL of 0.1% formic acid and were analyzed on an Exploris 480 Orbitrap mass spectrometer equipped with a FAIMS Pro source (*Bekker-Jensen et al., 2020*) connected to an EASY-nLC chromatography system using 0.1% formic acid as Buffer A and 80% aceto-nitrile/0.1% formic acid as Buffer B. Peptides were separated at 300 nL/min on a gradient of 2% B for 1 min, 2–25% B for 41 min, 25–40% B for 6 min, 40–100% B for 12 min, 100–2% B for 3 min, 2% B for 3 min, 2–98% B for 3 min, and 98% B for 3 min. The orbitrap and FAIMS were operated in positive ion mode with a positive ion voltage of 1800 V; with an ion transfer tube temperature of 270°C; using a standard FAIMS resolution and compensation voltage of –50 and –65 V, an inner and outer electrode temperature of 100°C with 4.5 mL/min carrier gas. Full-scan spectra were acquired in profile mode at a resolution of 60,000, with a scan range of 350–1400 m/z, 300% AGC target, 25 ms maximum injection time, intensity threshold of 5 × $10^3$, 2–6 charge state, dynamic exclusion of 20 s, 15 data-dependent scans (DDA Top 15), and mass tolerance of 10 ppm. MS2 spectra were generated with a HCD collision energy of 30 at a resolution of 15,000, first mass at 110 m/z, with an isolation window of 1.3 m/z, and a normalized AGC target of 200% with an automatically determined maximum injection time.

### Proximity labeling MS analysis

Raw files were analyzed in Proteome Discoverer 2.4 (Thermo Fisher Scientific) to generate peak lists and protein and peptides identifications using Sequest HT (Thermo Fisher Scientific) and the ToxoDB release 49 GT1 protein database. The maximum missed cleavage sites for trypsin was limited to 2. The following modifications were included in the search: dynamic oxidation (+15.995 Da; M), dynamic

phosphorylation (+79.966 Da; S,T,Y), dynamic biotinylation (+226.078 Da; K, N-terminus), dynamic acetylation (+42.011 Da; N-terminus), and static methylthio (+45.988 Da; C). Label-free quantification of proteins was performed using summed abundances from unique peptides. Abundances were normalized on the total peptide amount. Pairwise ratios were calculated for protein abundances comparing strains and conditions. Significance values were derived from $t$-tests across three replicates and adjusted with Benjamini–Hochberg correction. Exported protein abundance files from Proteome Discoverer 2.4 were loaded into R (version 4.1.1). The mass spectrometry proteomics data have been deposited to the ProteomeXchange Consortium via the PRIDE partner repository (*Perez-Riverol et al., 2022*) with the dataset identifier PXD039434 (10.6019/PXD039434). Protein abundances are reported in *Supplementary file 9*.

## MIC2 secretion assays

### Parasite harvest and treatment

Extracellular parasites were harvested in chilled DMEM and washed twice in DMEM after centrifugation at $1000 \times g$ for 10 min at 4°C. Parasites were resuspended at a concentration of $6 \times 10^8$ parasites/mL in cold DMEM. $3 \times 10^7$ parasites were aliquoted into round-bottom 96-well plates. An additional aliquot of parasites was lysed in 5× Laemmli sample buffer (see 'Immunoblotting' for recipe) to obtain the total parasite lysate used to determine total MIC2 levels. Secretion was stimulated in plates with IFS and ethanol in DMEM (final concentration 3% IFS and 1% ethanol) or a vehicle solution of DMEM. Plates were incubated by floating on a water bath at 37°C/5% $CO_2$ for 90 min. Plates were spun at $1000 \times g$ for 5 min at 4°C to separate parasites from secreted proteins. Supernatants were transferred to a new well and spun again. The final supernatant was mixed with a 5× Laemmli sample buffer, boiled at 95°C for 10 min, and stored at –20°C along with total lysates until immunoblot analysis.

### Immunoblot analysis and quantification

To quantify MIC2 protein levels, a standard curve derived from the total parasite lysate was generated alongside supernatants containing secreted microneme proteins. The standard curve was derived from threefold serial dilutions of the total parasite lysate (undiluted, 1:3, 1:9, and 1:27) in a 1× Laemmli sample buffer. The serial dilutions and supernatants of auxin- and vehicle-treated parasites of a single strain were loaded onto the same precast 4–15% gel (Bio-Rad). Subsequent immunoblotting steps were performed as described in 'Immunoblotting.' Anti-MIC2 was used to detect total and secreted MIC2 proteins. Secreted MIC2 proteins have a lower molecular weight due to parasite-mediated proteolytic cleavage. Anti-CDPK1 was used as a loading control for total parasite lysate and to reveal any parasite lysis or carry over in the supernatant. Immunoblots confirming the depletion of tagged proteins were also collected. Imaging was performed on a LI-COR Odyssey at high resolution for quantification.

Immunoblot quantification was performed in Fiji on unadjusted inverted images. For a single immunoblot, lane profiles of uniform dimensions were generated for MIC2 signal. Background-subtracted signal intensity was measured as an area using the line and wand tool. A standard curve was derived from the quantified MIC2 signal of the dilution series. Standard curves across all conditions and replicates had $R^2$ values >0.90. Secreted MIC2 signals were within the linear range of the standard curve and were used to calculate the percent of total MIC2 secreted. Results are representative of three independent experiments for each parasite strain.

## Microneme relocalization

Parasites expressing the CLAMP-mNG reporter were grown in HFFs in glass-bottomed 35 mm dishes (Ibidi and Mattek) for approximately 20 hr. For 3-MB-PP1 treatment, media was exchanged 30 min prior to live microscopy for 3% IFS in Ringer's buffer (155 mM NaCl, 2 mM $CaCl_2$, 3 mM KCl, 1 mM $MgCl_2$, 3 mM $NaH_2PO_4$, 10 mM HEPES, 10 mM glucose, pH 7.4) containing either 3 μM 3-MB-PP1 or vehicle solution of DMSO. At approximately 1 min after beginning live microscopy, parasites were stimulated with zaprinast (500 μM final concentration) or a vehicle solution of DMSO in corresponding Ringer's buffer. For TIR1/CLAMP-mNG and AID-HOOK/CLAMP-mNG parasite, media was exchanged 2 hr after infection for 10% IFS media containing either 50 μM auxin or vehicle solution of PBS. Media was exchanged for 3% IFS in Ringer's buffer just prior to live microscopy. At approximately 30 s after beginning live microscopy, parasites were stimulated with zaprinast (500 μM final concentration) or

a vehicle solution of DMSO in corresponding Ringer's buffer. Images were recorded every 5–7 s until egress or approximately 5 min using a Nikon Ti Eclipse with an enclosure maintained at 37°C.

Microneme relocalization was quantified using built-in commands from ImageJ (v. 1.53e). To determine the distribution of fluorescent micronemes at a specific time frame, a line was drawn from the parasite's apical end to its basal end. The 'plot profile' command was applied to this line to calculate fluorescence intensity at regularly spaced intervals (0.13 microns) across the parasite on unadjusted images. This process was repeated for each time frame of interest and for each parasite in the analyzed vacuole. Regression analysis was performed on relocalization data from each time frame, resulting in logarithmic regression plots that display how fluorescence intensity correlates with distance from the parasite's apical end. Microneme relocalization is shown as SuperPlots to simultaneously visualize the median relocalization of an entire vacuole and the relocalization of individual parasites within each vacuole (*Lord et al., 2020*).

## Microneme localization in extracellular parasites

Microneme localization in extracellular parasites was quantified using a custom-built, image analysis pipeline. ImageJ's built-in commands were used to measure the length of the parasite's major axis. Based on the patterns of vehicle-treated TIR1/CLAMP-mNG data, 1/8th of the parasite closest to the apical end was deemed the apical region. The remaining region was deemed the body. Using the Interactive Learning and Segmentation Toolkit (ilastik v. 1.3.3) (*Berg et al., 2019*), the apical region and body of parasites were isolated from each other and from the surrounding media. Ilastik's pixel and object quantification tools were then used to determine the total intensity of fluorescence in each of these regions. Comparing these values to each other yielded the percentage of fluorescent micronemes present in each of these sections in relation to the total fluorescence present in the parasite. This process was repeated for several extracellular parasites in each of the conditions tested.

## Ultrastructure expansion microscopy (U-ExM)

U-ExM of extracellular *T. gondii* parasites was performed as described previously (*Dos Santos Pacheco and Soldati-Favre, 2021*). In short, TIR1 and TIR1/AID-HOOK strains were either treated with 50 µM auxin or a vehicle of 1× PBS for 40 hr. Naturally egressed parasites diluted to a concentration of $1 \times 10^8$ parasites/mL were let settle for 10 min on poly-D-lysine-coated glass coverslips (Life Technologies A3890401). Then, a molecular 'anchor' was added to cellular proteins by incubating the coverslips in a solution containing 1.4% formaldehyde (Sigma F8775) and 2% acrylamide (Sigma A4058) in 1× PBS for 5 hr at 37°C. Next, samples were infused with a gel (19% Sodium Acrylate [AK Scientific R624], 10% acrylamide [Sigma A4058], 0.1% N,N'-methylenebisacrylamide [BIS, Sigma M1533], 0.5% tetramethylethylenediamine [TEMED, Sigma T9281], and 0.5% ammonium persulfate [A3678 Sigma] in PBS) for 1 hr at 37°C. The gel mesh is crosslinked to the 'anchors' from the previous step. To enable proper cell expansion, the samples were denatured for 1.5 hr at 95°C in a denaturation buffer containing 200 mM SDS (Invitrogen AM9820), 200 mM NaCl, and 50 mM TRIS-Base at pH 9. The gels are then expanded in water, shrunk in PBS, incubated with antibodies diluted in 2% BSA in PBS, and then expanded again in water. Primary antibodies were anti-Acetyl Tubulin (Lys40) (1:1000, 3 hr; Sigma ABT24) and anti-MIC2 (1:1000, 3 hr; 6D10). Secondary antibodies were Alexa Fluor 594 (1:750, 2 hr; Invitrogen A11008) and Alexa Fluor 488 (1:750, 2 hr; Invitrogen A11005). Antibodies were incubated with agitation at 37°C. Finally, the gels were put on a 35 mm microscopy dish (MatTek Life Sciences), parasites facing down. Images were acquired by Zeiss SLM 980 microscope using the Airyscan super-resolution mode.

## Image analysis of U-ExM

Between 14 and 18 microscopy images per condition were analyzed using Imaris software (release 9.9). Images, each containing one cell, were converted to Imaris files using the standalone ImarisFile-Converter software. To automatically detect the extruded conoid and tubulin filaments of each cell, the FilamentTracer Wizard was trained on a few images to identify the conoid as 'Soma' and tubulin filaments as 'Dendrites.' Micronemes were automatically detected using the 'Spots' Wizard. To apply the same mask parameters to all images, the Batch Wizard was used. As some images contained cropped parts of neighboring cells, manual adjustments to the mask were applied to include or exclude specific conoid, tubulin, or micronemes that did not belong to the analyzed cell. To export

statistics that include the 'spots shortest distance to surface' parameter, both the conoid and tubulin masks were converted to regular surfaces (i.e., not 'soma' and 'dendrites') by creating new channels based on their masks and applying new surface masks on these new channels. Overall, each image contained the following mask objects: micronemes as Spots, tubulin as Surface 1, and the conoid as Surface 2. Then, statistics parameters were exported, including Spots shortest distance from Surface 1, Spots shortest distance from Surface 2, Spots number per cell, and Spots center intensity.

## Materials availability statement

All mass spectrometry proteomics data have been deposited to the ProteomeXchange Consortium via the PRIDE partner repository. Accession numbers are listed in the appropriate 'Materials and methods' sections and here: Sub-minute resolution phosphoproteomics (PXD039426), Immunoprecipitation of cWT and cMut (PXD044081), Thiophosphorylation of CDPK1 substrates (PXD039431), Immunoprecipitation of HOOK and FTS (PXD039432), Immunoprecipitation of TGGT1_306920 (PXD044080), and Proximity labeling FTS-TurboID (PXD039434). Most sequences of cloning vectors generated for this study have been deposited in GenBank, as listed in *Supplementary file 10*. Custom analysis scripts in the R computing language are available in *Supplementary file 11* (Sub-minute resolution phosphoproteomics) and *Supplementary file 12* (Thiophosphorylation of CDPK1 substrates). Strains and plasmids generated for this study can be obtained by emailing the corresponding author.

## Acknowledgements

We thank all the members of the Lourido laboratory for helpful discussions and support, especially Tyler Smith for aiding with the tagging vector design and Christopher Giuliano for insights during data analysis. We thank Forest M White for generous support in developing the mass spectrometry methodologies used in this study and for helpful discussions. For data in *Figure 2*, Caia Dominicus designed the cloning strategy for all the parasite lines (iKD CDPK1, cWT, cMut), and was overseeing the whole cloning process. Alex Hunt performed the IFAs for the cWT and cMut lines. We thank Matthew Child and Matt Bogyo for the CDPK1 antibody, John Boothroyd for the SAG1 antibody. We thank Marc-Jan Gubbels for the alpha-tubulin antibody, L David Sibley for the MIC2, SAG1, and ALD antibody, Peter Bradley for the ROP1 antibody, Drew Etheridge for the GAP45 antibody, and Dominique Soldati-Favre for the GAP45 antibody. This work relied on VEuPathDB.org and we thank all contributors to this resource. This research was supported by funds from National Institutes of Health grants to SL (R01AI144369) and MT (R01AI123457), a National Science Foundation Graduate Research Fellowships to AWC and ALH (174530). MT received funding from the Francis Crick Institute which receives its core funding from Cancer Research UK (CC2132), the UK Medical Research Council (CC2132), and the Wellcome Trust (CC2132). The Science Technology Proteomics Platform at the Francis Crick Institute received funding from Cancer Research UK (CC0199), the UK Medical Research Council (CC0199), and the Wellcome Trust (CC0199). EY is supported by the European Molecular Biology Organisation Postdoctoral Fellowship (ALTF 73-2022) and by the Israeli Council for Higher Education. For the purpose of Open Access, the authors have applied a CC BY public copyright license to any Author Accepted Manuscript version arising from this submission.

## Additional information

### Competing interests

Sebastian Lourido: Reviewing editor, *eLife*. The other authors declare that no competing interests exist.

### Funding

| Funder | Grant reference number | Author |
| --- | --- | --- |
| National Institutes of Health | R01AI144369 | Sebastian Lourido |

| Funder | Grant reference number | Author |
|---|---|---|
| National Institutes of Health | R01AI123457 | Moritz Treeck |
| National Science Foundation | 174530 | Alex W Chan<br>Alice L Herneisen |
| Francis Crick Institute | | Moritz Treeck |
| European Molecular Biology Organization | ALTF 73-2022 | Eden Yifrach |
| Israeli Council for Higher Education | | Eden Yifrach |

The funders had no role in study design, data collection and interpretation, or the decision to submit the work for publication.

## Author contributions

Alex W Chan, Conceptualization, Formal analysis, Validation, Investigation, Methodology, Writing - original draft, Writing - review and editing; Malgorzata Broncel, Formal analysis, Validation, Investigation, Methodology; Eden Yifrach, Elena Andree, Investigation; Nicole R Haseley, Formal analysis, Methodology; Sundeep Chakladar, Formal analysis, Investigation, Methodology; Alice L Herneisen, Resources, Methodology; Emily Shortt, Resources; Moritz Treeck, Resources, Supervision, Funding acquisition, Writing - review and editing; Sebastian Lourido, Conceptualization, Resources, Supervision, Funding acquisition, Methodology, Writing - review and editing

## Author ORCIDs

Alex W Chan http://orcid.org/0000-0002-5444-5756
Malgorzata Broncel http://orcid.org/0000-0003-2991-3500
Eden Yifrach http://orcid.org/0000-0001-5074-0048
Nicole R Haseley http://orcid.org/0000-0002-9346-2161
Alice L Herneisen http://orcid.org/0000-0003-3368-0893
Emily Shortt http://orcid.org/0009-0009-7625-4772
Moritz Treeck http://orcid.org/0000-0002-9727-6657
Sebastian Lourido http://orcid.org/0000-0002-5237-1095

Reviewer #1 (Public Review): https://doi.org/10.7554/eLife.85654.3.sa1
Reviewer #2 (Public Review): https://doi.org/10.7554/eLife.85654.3.sa2
Reviewer #3 (Public Review): https://doi.org/10.7554/eLife.85654.3.sa3
Author Response: https://doi.org/10.7554/eLife.85654.3.sa4

# Additional files

## Supplementary files

• Supplementary file 1. Sub-minute phosphoproteomics time course peptide and abundance assignments from Proteome Discoverer 2.4 and analysis for the zaprinast enriched samples. (A) RAW peptide and abundance assignments. (B) Replicate 1 analysis with $AUC_{difference}$ and p-values. (C) Replicate 2 analysis with $AUC_{difference}$ and p-values. (D) Replicate 1 and 2 analysis with $AUC_{vehicle}$ values for CDPK1-independent phosphorylation. (E) Group assignments for zaprinast-dependent phosphopeptides.

• Supplementary file 2. Sub-minute phosphoproteomics time course protein and abundance assignments from Proteome Discoverer 2.4 for the zaprinast proteome.

• Supplementary file 3. Sub-minute phosphoproteomics time course peptide and abundance assignments from Proteome Discoverer 2.4 and analysis for the DMSO enriched samples. (A) RAW peptide and abundance assignments. (B) Replicate 1 analysis with $AUC_{difference}$ values. (C) Replicate 2 analysis with $AUC_{difference}$ values. (D) Replicate 1 and 2 analysis with $AUC_{vehicle}$ values for CDPK1-independent phosphorylation.

• Supplementary file 4. Sub-minute phosphoproteomics time course protein and abundance assignments from Proteome Discoverer 2.4 for the DMSO proteome.

• Supplementary file 5. CDPK1 immunoprecipitation protein and abundance assignments from Scaffold DIA.

• Supplementary file 6. Thiophosphorylation of CDPK1 substrates. Peptide and abundance assignments from Proteome Discoverer 2.4 and analysis. (A) RAW peptide and abundance assignments. (B) Analyzed peptides with enrichment significance values. (C) Analyzed flow-through peptides used for normalization.

• Supplementary file 7. Thiophosphorylation of CDPK1 substrates. Protein and abundance assignments from Proteome Discoverer 2.4 and analysis.

• Supplementary file 8. HOOK, FTS, and TGGT1_306920 immunoprecipitation protein and abundance assignments from Proteome Discoverer 2.4.

• Supplementary file 9. FTS proximity labeling protein and abundance assignments from Proteome Discoverer 2.4.

• Supplementary file 10. Sequences and accessions of oligonucleotides and plasmids used in this study.

• Supplementary file 11. Zip file containing R scripts and CSVs for sub-minute phosphoproteomic time course analysis.

• Supplementary file 12. Zip file containing R scripts and CSVs for thiophosphorylation analysis.

• MDAR checklist

### Data availability

All mass spectrometry proteomics data have been deposited to the ProteomeXchange Consortium via the PRIDE partner repository. Accession numbers are listed in the appropriate 'Materials and methods' sections and here: Sub-minute resolution phosphoproteomics (PXD039426), Immunoprecipitation of cWT and cMut (PXD044081), Thiophosphorylation of CDPK1 substrates (PXD039431), Immunoprecipitation of HOOK and FTS (PXD039432), Immunoprecipitation of TGGT1_306920 (PXD044080), and Proximity labeling FTS-TurboID (PXD039434). Most sequences of cloning vectors generated for this study have been deposited in GenBank, as listed in *Supplementary file 10*. Custom analysis scripts in the R computing language are available in *Supplementary file 11* (Sub-minute resolution phosphoproteomics) and *Supplementary file 12* (Thiophosphorylation of CDPK1 substrates). Strains and plasmids generated for this study can be obtained by emailing the corresponding author.

The following datasets were generated:

| Author(s) | Year | Dataset title | Dataset URL | Database and Identifier |
|---|---|---|---|---|
| Chan AW | 2023 | Analysis of CDPK1 targets identifies a trafficking adaptor complex that regulates microneme exocytosis in Toxoplasma | https://www.ebi.ac.uk/pride/archive/projects/PXD039426 | PRIDE, PXD039426 |
| Chan AW | 2023 | Analysis of CDPK1 targets identifies a trafficking adaptor complex that regulates microneme exocytosis in Toxoplasma - cWT and cMut IPs | https://www.ebi.ac.uk/pride/archive/projects/PXD044081 | PRIDE, PXD044081 |
| Chan AW | 2023 | Analysis of CDPK1 targets identifies a trafficking adaptor complex that regulates microneme exocytosis in Toxoplasma - Thiophosphorylation | https://www.ebi.ac.uk/pride/archive/projects/PXD039431 | PRIDE, PXD039431 |

*Continued on next page*

*Continued*

| Author(s) | Year | Dataset title | Dataset URL | Database and Identifier |
|---|---|---|---|---|
| Chan AW | 2023 | Analysis of CDPK1 targets identifies a trafficking adaptor complex that regulates microneme exocytosis in Toxoplasma - HOOK and FTS IPs | https://www.ebi.ac.uk/pride/archive/projects/PXD039432 | PRIDE, PXD039432 |
| Chan AW | 2023 | Analysis of CDPK1 targets identifies a trafficking adaptor complex that regulates microneme exocytosis in Toxoplasma - TGGT1_306920 IPs | https://www.ebi.ac.uk/pride/archive/projects/PXD044080 | PRIDE, PXD044080 |
| Chan AW | 2023 | Analysis of CDPK1 targets identifies a trafficking adaptor complex that regulates microneme exocytosis in Toxoplasma - FTS TurboID | https://www.ebi.ac.uk/pride/archive/projects/PXD039434 | PRIDE, PXD039434 |

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

# Appendix 1

## Appendix 1—key resources table

| Reagent type (species) or resource | Designation | Source or reference | Identifiers | Additional information |
|---|---|---|---|---|
| Strain, strain background RH (*Toxoplasma gondii*) | TIR1 | PMID:28465425 | | RH/TIR1/ΔKU80/ΔHXGPRT. Mycoplasma negative. |
| Strain, strain background RH (*T. gondii*) | CDPK1-AID | doi:https://doi.org/10.1101/2022.07.19.500742 | | RH/TIR1/ΔKU80/ΔHXGPRT/CDPK1-V5-mNeonGreen-mAID-Ty. Mycoplasma negative. |
| Strain, strain background RH (*T. gondii*) | iKD CDPK1 | This paper | TGGT1_301440 | RH/TIR1/ΔKU80/ΔHXGPRT/CDPK1-mAID-Myc/HXGPRT. Mycoplasma negative. |
| Strain, strain background RH (*T. gondii*) | cWT CDPK1 | This paper | TGGT1_301440 | RH/TIR1/ΔKU80/ΔHXGPRT/CDPK1-mAID-Myc/HXGPRT/CDPK1-HA-T2A-GFP. Mycoplasma negative. |
| Strain, strain background RH (*T. gondii*) | cMut CDPK1 | This paper | TGGT1_301440 | RH/TIR1/ΔKU80/ΔHXGPRT/CDPK1-mAID-Myc/HXGPRT/CDPK1(G2A)-HA-T2A-mCherry. Mycoplasma negative. |
| Strain, strain background RH (*T. gondii*) | Δku80Δhxgprt YFP | PMID:32618271 | | RH/ΔKU80/ΔHXGPRT/YFP. Mycoplasma negative. |
| Strain, strain background RH (*T. gondii*) | CDPK1G | PMID:19218426 | | RH/ΔKU80/ΔHXGPRT. Mycoplasma negative. |
| Strain, strain background RH (*T. gondii*) | CDPK1M | PMID:23149386 | TGGT1_301440 | RH/ΔKU80/ΔHXGPRT/CDPK1(G128M)-Myc. Mycoplasma negative. |
| Strain, strain background RH (*T. gondii*) | AID-HOOK | This paper | TGGT1_289100 | RH/TIR1/ΔKU80/ΔHXGPRT/HA-mAID-HOOK. Mycoplasma negative. |
| Strain, strain background RH (*T. gondii*) | CDPK1-AID/HOOK-3xHA | This paper | TGGT1_289100 | RH/TIR1/ΔKU80/ΔHXGPRT/CDPK1-V5-mNeonGreen-mAID-Ty/HOOK-V5-3HA/HXGPRT. Mycoplasma negative. |
| Strain, strain background RH (*T. gondii*) | FTS-AID | This paper | TGGT1_264050 | RH/TIR1/ΔKU80/ΔHXGPRT/FTS-V5-mAID-HA/HXGPRT. Mycoplasma negative. |
| Strain, strain background RH (*T. gondii*) | 306920-3xHA | This paper | TGGT1_306920 | RH/TIR1/ΔKU80/ΔHXGPRT/306920-V5-3HA/HXGPRT. Mycoplasma negative. |
| Strain, strain background RH (*T. gondii*) | FTS-3xHA | This paper | TGGT1_264050 | RH/TIR1/ΔKU80/ΔHXGPRT/FTS-V5-3HA/HXGPRT. Mycoplasma negative. |
| Strain, strain background RH (*T. gondii*) | FTS-TurboID | This paper | TGGT1_264050 | RH/TIR1/ΔKU80/ΔHXGPRT/FTS-TurboID-Ty/DHFR. Mycoplasma negative. |
| Strain, strain background RH (*T. gondii*) | Cytosolic mNG-TurboID | This paper | | RH/TIR1/ΔKU80/ΔHXGPRT/pMIC2-mNeonGreen-TurboID-Ty. Mycoplasma negative. |
| Strain, strain background RH (*T. gondii*) | CLAMP-mNG | PMID:27594426 | TGGT1_265790 | RH/ΔKU80/ΔHXGPRT/CLAMP-mNeonGreen. Mycoplasma negative. |
| Strain, strain background RH (*T. gondii*) | TIR1/CLAMP-mNG | This paper | TGGT1_265790 | RH/TIR1/ΔKU80/ΔHXGPRT/CLAMP-mNeonGreen. Mycoplasma negative. |
| Strain, strain background RH (*T. gondii*) | TIR1/CLAMP-mNG/AID-HOOK | This paper | TGGT1_265790 | RH/TIR1/ΔKU80/ΔHXGPRT/HA-mAID-HOOK/CLAMP-mNeonGreen. Mycoplasma negative. |
| Cell line (*Homo sapiens*) | Human Foreskin Fibroblasts (HFFs) | ATCC | SCRC-1041 | Mycoplasma negative. |
| Antibody | Mouse polyclonal anti-CDPK1 | PMID:28246362 | | Provided by Matthew Child and Matt Bogyo. WB (1:3000). Only used in *Figure 2*. |
| Antibody | Mouse monoclonal clone 4A6 anti-Myc | Millipore | Cat# 05-724, RRID:AB_11211891 | WB (1:1000). |

*Appendix 1 Continued on next page*

*Appendix 1 Continued*

| Reagent type (species) or resource | Designation | Source or reference | Identifiers | Additional information |
|---|---|---|---|---|
| Antibody | Rat monoclonal clone 3F10 anti-HA | Roche | Cat# 11867423001, RRID:AB_390919 | WB (1:1000); IFA (1:1000). Only used in *Figure 2*. |
| Antibody | Mouse monoclonal clone TP3 anti-*Toxoplasma* | Abcam | Cat# ab8313, RRID:AB_306466 | WB (1:1000). |
| Antibody | Mouse monoclonal anti-GFP | Roche | Cat# 11814460001, RRID:AB_390913 | WB (1:1000). |
| Antibody | Rabbit polyclonal anti-mCherry | Abcam | Cat# ab167453, RRID:AB_2571870 | WB (1:1000). |
| Antibody | Rabbit monoclonal anti-SAG1 | PMID:3183382 | | Provided by John Boothroyd Lab; WB (1:10,000). Only used in *Figure 2*. |
| Antibody | Rabbit monoclonal clone 51-8 anti-thiophosphate ester | Abcam | Cat# ab92570, RRID:AB_10562142 | WB (1:5000). |
| Antibody | Mouse monoclonal clone 12G10 anti-tubulin | Developmental Studies Hybridoma Bank at the University of Iowa | RRID:AB_1157911 | Provided by Marc-Jan Gubbels Lab. WB (1:2000). |
| Antibody | Guinea pig monoclonal anti-CDPK1 | Covance | Custom antibody | WB (1/50,000). |
| Antibody | Rabbit monoclonal (C29F4) anti-HA | Cell Signaling Technology | Cat# 3724, RRID:AB_1549585 | WB (1:1000); IFA (1:1600). |
| Antibody | Mouse monoclonal clone 6D10 anti-MIC2 | PMID:10799515 | | Provided by L. David Sibley Lab. WB (1:5000); IFA (1:2000). |
| Antibody | Mouse anti-ROP1 | PMID:12467986 | | Provided by Peter Bradley Lab. IFA (1:2000). |
| Antibody | Mouse polyclonal anti-SAG1 | PMID:3183382 | | Provided by L. David Sibley Lab. IFA (1:500). Used for invasion assays. |
| Antibody | Rabbit polyclonal anti-GAP45 | Lampire Biological Laboratory | | Provided by R. Drew Etheridge Lab. IFA (1:1000). Used for invasion assays. |
| Antibody | Mouse monoclonal (16B12) anti-HA | BioLegend | Cat# 901533, RRID:AB_2565005 | WB (1:1000). |
| Antibody | Rabbit polyclonal clone WU1614 anti-ALD | PMID:16923803 | | Provided by L. David Sibley Lab. WB (1:10,000). |
| Antibody | Rabbit polyclonal anti-GAP45 | PMID:18312842 | | Provided by Dominique Soldati-Favre Lab. WB (1:5000); IFA (1:5000). |
| Antibody | Peroxidase-AffiniPure polyclonal Goat Anti-Rabbit IgG (H+L) | Jackson ImmunoResearch Laboratories | Cat# 111-035-003, RRID:AB_2313567 | WB (1/5000). |
| Antibody | Alexa Fluor 594 polyclonal Goat Anti-Rabbit IgG (H+L) | Life Technologies | | IFA (1:1000). |
| Antibody | Alexa Fluor 488 polyclonal Goat Anti-Mouse IgG (H+L) | Life Technologies | | IFA (1:1000). |
| Antibody | Alexa Fluor 488 polyclonal Goat Anti-Rabbit IgG (H+L) | Life Technologies | | IFA (1:1000). |
| Antibody | Alexa Fluor 594 polyclonal Goat Anti-Mouse IgG (H+L) | Life Technologies | | IFA (1:1000). |
| Antibody | IRDye 800CW Donkey anti-Guinea Pig IgG Secondary Antibody | LI-COR | LI-COR: 926-32411 | WB (1:10,000). |
| Antibody | IRDye 680RD Donkey anti-Guinea Pig IgG Secondary Antibody | LI-COR | LI-COR: 926-68077 | WB (1:10,000). |
| Antibody | IRDye 800CW Goat anti-Mouse IgG1-Specific Secondary Antibody | LI-COR | LI-COR: 926-32350 | WB (1:10,000). |
| Antibody | IRDye 680LT Goat anti-Mouse IgG Secondary Antibody | LI-COR | LI-COR: 926-68020 | WB (1:10,000). |
| Antibody | IRDye 800CW Goat anti-Rabbit IgG Secondary Antibody | LI-COR | LI-COR: 926-32211 | WB (1:10,000). |

*Appendix 1 Continued on next page*

*Appendix 1 Continued*

| Reagent type (species) or resource | Designation | Source or reference | Identifiers | Additional information |
|---|---|---|---|---|
| Antibody | IRDye 680LT Goat anti-Rabbit IgG Secondary Antibody | LI-COR | LI-COR: 926-68021 | WB (1:10,000). |
| Antibody | Alpaca monoclonal (1B7) anti-CDPK1 | PMID:26305940 | n/a | |
| Peptide, recombinant protein | IRDye 680RD Streptavidin | LI-COR | LI-COR: 926-68079 | WB (1:3000). |
| Peptide, recombinant protein | Aerolysin | PMID:26584919 | n/a | |
| Peptide, recombinant protein | Sequencing Grade Modified Trypsin | Promega | Promega: V5113 | |
| Chemical compound, drug | 3-Indoleacetic acid (auxin) | Sigma-Aldrich | Sigma-Aldrich: I2886-5G | |
| Chemical compound, drug | Compound 1 | PMID:12455981 | n/a | |
| Chemical compound, drug | Hoechst 33258 | Santa Cruz | Santa Cruz: sc-394039 | Egress assay (1:4000). |
| Chemical compound, drug | Hoechst 33342 | Invitrogen | Invitrogen: H3570 | IFA (1:20,000). |
| Chemical compound, drug | DAPI (4',6-diamidino-2-phenylindole, dihydrochloride) | Invitrogen | Invitrogen: D1306 | |
| Chemical compound, drug | Prolong Diamond | Thermo Fisher | Thermo Fisher: P36965 | |
| Chemical compound, drug | Zaprinast | Calbiochem | Calbiochem: 684500 | |
| Chemical compound, drug | A23187 | Calbiochem | Calbiochem: 100105 | |
| Chemical compound, drug | Myristic acid | Tokyo Chemical Industry | Cat# M0476 | |
| Chemical compound, drug | Alkyne-myristic acid (YnMyr) | Iris Biotech | Cat# RL-2055 | |
| Chemical compound, drug | Trypsin cleavable capture reagent | PMID:25807930; PMID:32618271 | | This reagent was first reported as RTB in PMID:25807930. |
| Chemical compound, drug | L-Proline for SILAC | Thermo Fisher Scientific | Thermo Fisher Scientific: 88211 | |
| Chemical compound, drug | Fetal bovine serum, dialyzed, US origin, One Shot format | Thermo Fisher Scientific | Thermo Fisher Scientific: A3382001 | |
| Chemical compound, drug | DMEM for SILAC | Thermo Fisher Scientific | Thermo Fisher Scientific: 88364 | |
| Chemical compound, drug | L-Arginine-HCl for SILAC | Thermo Fisher Scientific | Thermo Fisher Scientific: 89989 | |
| Chemical compound, drug | L-Lysine-2HCl for SILAC | Thermo Fisher Scientific | Thermo Fisher Scientific: 89987 | |
| Chemical compound, drug | L-Arginine-HCl, 13C6, 15N4 for SILAC | Thermo Fisher Scientific | Thermo Fisher Scientific: 89990 | |
| Chemical compound, drug | L-Lysine-2HCl, 13C6, 15N2 for SILAC | Thermo Fisher Scientific | Thermo Fisher Scientific: 88209 | |
| Chemical compound, drug | p-Nitrobenzyl mesylate (PNBM) | Epitomics | Epitomics # 3700-1 | |
| Chemical compound, drug | OXONE, monopersulfate compound | Sigma-Aldrich | Sigma-Aldrich: 228036-5G | |
| Chemical compound, drug | N6-furfuryladenosine (kinetin)–5'-O-[3-thiotriphosphate] sodium salt (KTPγS) | Axxora | Axxora: BLG-F008-05 | |
| Chemical compound, drug | Adenosine 5'-triphosphate disodium salt hydrate (ATP) | Sigma-Aldrich | Sigma-Aldrich: A6419-1G | |
| Chemical compound, drug | Guanosine 5'-Triphosphate, Disodium Salt (GTP) | Calbiochem | Millipore: 371701 | |
| Chemical compound, drug | β-Casein from bovine milk | Sigma-Aldrich | Sigma-Aldrich: C6905 | |
| Chemical compound, drug | Biotin | Sigma-Aldrich | Sigma-Aldrich: B4501-1G | |

*Appendix 1 Continued on next page*

*Appendix 1 Continued*

| Reagent type (species) or resource | Designation | Source or reference | Identifiers | Additional information |
|---|---|---|---|---|
| Chemical compound, drug | PP1 Analog III, 3-MB-PP1 | Calbiochem | Sigma-Aldrich: 529582 | |
| Chemical compound, drug | Formaldehyde | Sigma-Aldrich | Sigma-Aldrich: F8775 | |
| Chemical compound, drug | Acrylamide | Sigma-Aldrich | Sigma-Aldrich: A4058 | |
| Chemical compound, drug | Sodium acrylate | AK Scientific | AK Scientific: R624 | |
| Chemical compound, drug | N,N'-Methylenebisacrylamide (BIS) | Sigma-Aldrich | Sigma-Aldrich: M1533 | |
| Chemical compound, drug | Tetramethyle thylenediamine (TEMED) | Sigma-Aldrich | Sigma-Aldrich: T9281 | |
| Chemical compound, drug | Ammonium persulfate (APS) | Sigma-Aldrich | Sigma-Aldrich: A3678 | |
| Commercial assay or kit | S-trap micro | Protifi | Protifi: C02-micro-80 | |
| Commercial assay or kit | Pierce Quantitative Fluorometric Peptide Assay | Thermo Fisher Scientific | Thermo Fisher Scientific: 23290 | |
| Commercial assay or kit | TMTpro 16plex Label Reagent Set | Thermo Fisher Scientific | Thermo Fisher Scientific: A44522 | |
| Commercial assay or kit | EasyPep MS Sample Prep Kits - Maxi | Thermo Fisher Scientific | Thermo Fisher Scientific: A45734 | |
| Commercial assay or kit | High-Select TiO$_2$ Phosphopeptide Enrichment Kit | Thermo Fisher Scientific | Thermo Fisher Scientific: A32993 | |
| Commercial assay or kit | High-Select Fe-NTA Phosphopeptide Enrichment Kit | Thermo Fisher Scientific | Thermo Fisher Scientific: A32992 | |
| Commercial assay or kit | Pierce High pH Reversed-Phase Peptide Fractionation Kit | Thermo Fisher Scientific | Thermo Fisher Scientific: 84868 | |
| Commercial assay or kit | Dynabeads MyOne Streptavidin C1 | Invitrogen | Cat# 65001 | |
| Commercial assay or kit | Bio-Rad DC assay | Bio-Rad | Bio-Rad: 5000116 | |
| Commercial assay or kit | Sep-Pak C18 Plus Short Cartridge, 360 mg Sorbent per Cartridge, 55–105 µm | Waters | Waters: WAT020515 | |
| Commercial assay or kit | SulfoLink Coupling Resin | Thermo Fisher Scientific | Thermo Fisher Scientific: 20401 | |
| Commercial assay or kit | Radiance Plus Chemiluminescent Substrate | Azure Biosystems | VWR: 10147-298 | |
| Commercial assay or kit | Pierce Anti-HA Magnetic Beads | Thermo Scientific | Thermo Scientific: 88836 | |
| Commercial assay or kit | Pierce Streptavidin Magnetic Beads | Thermo Scientific | Thermo Scientific: 88817 | |
| Recombinant DNA reagent | All plasmids used in this study are listed in *Supplementary file 10* | This paper | | |
| Sequence-based reagent | All primers and oligonucleotides used in this study are listed in *Supplementary file 10* | This paper | | |
| Software, algorithm | Proteome Discoverer 4.2 | Thermo Fisher | | |
| Software, algorithm | MaxQuant (version 1.5.0.25 and 1.5.2.8) | PMID:19029910 | RRID:SCR_014485 | Free software for searching of mass spectrometry acquisition files. |
| Software, algorithm | Perseus (version 1.5.0.9) | PMID:27348712 | RRID:SCR_015753 | Free software for processing of MaxQuant output files. |
| Software, algorithm | Scaffold DIA | Proteome Software | | |
| Software, algorithm | R version 4.0 | R Foundation for Statistical Computing | | |
| Software, algorithm | Prism 8 | GraphPad | | |
| Software, algorithm | Fiji | PMID:22743772 | | |
| Software, algorithm | ilastik | PMID:31570887 | ilastik.org | |

*Appendix 1 Continued on next page*

*Appendix 1 Continued*

| Reagent type (species) or resource | Designation | Source or reference | Identifiers | Additional information |
|---|---|---|---|---|
| Software, algorithm | HHPRED | PMID:29258817 | | |
| Software, algorithm | SnapGene | Dotmatics | https://www.snapgene.com/ | |
| Software, algorithm | ToxoDB | PMID:18003657 | ToxoDB.org | |
| Software, algorithm | Imaris (release 9.9) | Oxford Instruments | https://imaris.oxinst.com/ | |
| Other | DMEM, high glucose (media) | Life Technologies | Life Technologies: 11965118 | Materials and methods: Cell culture |
| Other | HALT protease inhibitor | Thermo Fisher | Thermo Fisher: 87786 | Materials and methods: Immunoprecipitation, lysis buffer |
| Other | HALT protease and phosphatase inhibitor | Thermo Fisher | Thermo Fisher: PI78440 | Materials and methods: Sub-minute resolution phosphoproteomics, lysis buffer |
| Other | Benzonase | Sigma-Aldrich | Sigma-Aldrich: E1014 | Materials and methods: Sub-minute resolution phosphoproteomics, lysis buffer |

