## [Editor Report · eLife assessment]

This **important** study in the protozoan parasite *Toxoplasma gondii* significantly advances our understanding of calcium signaling mediated by the kinase CDPK1 in this species. The authors' conclusions are supported by **convincing** evidence, with rigorous biochemical experiments and microscopy analysis. The work will be of broad interest to researchers in the fields of signal transduction and protozoan biology.

---

## [Referee Report · Reviewer #1 (Public Review)]

This carefully done research paper presents a fundamental model of techniques that are useful for the elucidation of kinase substrates. The paper utilizes state-of-the-art approaches to define a kinetic phosphoproteome and how to integrate that data with complementary approaches using a chemical probe (in this case KTPyS, a triphosphate) to find these substrates. Using these approaches TgCDPK1 was demonstrated to affect microneme secretion via a direct interaction with a HOOK complex (defined as a HOOK protein TGG1_289100, an FTS TGGT1_264050 and 2 other proteins TGGT1_316650 and 306920).

This work is carefully controlled and the analysis pathways are logical and provide paradigms for how to approach the question of identifying substrates of kinases using proteomic approaches employing genetic and chemical strategies.

The authors succeeded in the identification of candidate substrates for TgCDPK1. Validation of the results was provided by previous studies in the literature that characterized some of these substrates as well as the experiments in this manuscript on the characterization of the HOOK complex that is phosphorylated by CDPK1.

The HOOK complex (defined as a HOOK protein TGG1_289100, an FTS TGGT1_264050, and 2 other proteins TGGT1_316650 and 306920) was clearly demonstrated to be involved in invasion via its role in microneme trafficking.

---

## [Referee Report · Reviewer #2 (Public Review)]

In this study, the authors take a multipronged approach to identify the substrate repertoire of calcium-dependent protein kinase, CDPK1 in Toxoplasma that includes quantitative phosphoproteomics, myristoylation, thiophosphorylation, immunoprecipitation as well as proximity-based labeling. Their finding also reveals that CDPK1 functions in parasite invasion and egress by phosphorylating different protein candidates. More importantly, the authors successfully determine one branch of the CDPK1 signaling pathway that regulates invasion through the phosphorylation of the HOOK protein involved in the translocation and secretion of micronemal proteins.

---

## [Referee Report · Reviewer #3 (Public Review)]

In this manuscript, Chan and collaborators investigate the role of CDPK1 in regulating microneme trafficking and exocytosis in *Toxoplasma gondii*. Micronemes are apicomplexan-specific organelles localized at the apical end of the parasite and depending on cortical microtubules. Micronemes contain proteins that are exocytosed in a Ca²+-dependent manner and are required for *T. gondii* egress, motility, and host-cell invasion. In Apicomplexa, Ca²+ signaling is dependent on Ca²+-dependent protein kinases (CDPKs). CDPK1 has been demonstrated to be essential for Ca²+-stimulated micronemes exocytosis allowing parasite egress, gliding motility, and invasion. It is also known that intracellular calcium storages are mobilized following a cyclic nucleotide-mediated activation of protein kinase G. This step, occurs upstream of CDPK1 functions. However, the exact signaling pathway regulated by CDPK1 remains unknown. In this paper, the authors used phosphoproteomic analysis to identify new proteins phosphorylated by CDPK1. They demonstrated that CDPK1 activity is required for calcium-stimulated trafficking of micronemes to the apical end, depending on a complex of proteins that include HOOK and FTS, which are known to link cargo to the dynein machinery for trafficking along microtubules. Overall, the authors identified evidence for a new protein complex involved in microneme trafficking through the exocytosis process for which circumstantial evidence of its functionality is demonstrated here.

---

## [Author Response]

The following is the authors’ response to the original reviews.

We thank the reviewers for their thoughtful and positive evaluation of our work. Below, we have addressed all of the essential revisions and provide point-by-point responses to all of the reviewer comments. Additionally, we include with this resubmission quantification microneme localization, determined by expansion microscopy, which further validates the central role of HOOK in microneme trafficking.

**Suggested revisions:**
1. Please confirm the interaction between CDPK1 and ROM4 by reciprocal IP.

Prompted by the reviewers suggestions we examined more closely the pulldowns of WT and myristoylation-deficient CDPK1 (cMut). ROM4 had been identified as differentially enriched in the cMut pulldown; however, upon closer examination we realized that the abundance of ROM4 is actually even greater in the untagged control and therefore likely a variable contaminant in the pulldowns. We have re-analyzed the results of those pulldowns to focus on proteins significantly enriched in association with either WT or cMut CDPK1, relative to untagged controls. Among this set of 16 enriched proteins, only three proteins appeared differentially enriched between WT and cMut. None of the proteins associated with CDPK1 inform pathways related to parasite motility and were therefore not pursued further in this study.

2. Please compare the expression of the tagged and complemented (cWT and cMut) CDPK1 with the endogenous expression of the non-tagged and non-complemented gene.

We compared expression levels of CDPK1 using immunoblot with an anti-CDPK1 antibody comparing TIR1, CDPK1-AID, cWT and cMut parasites, which we have included in panel G of Figure 2–figure supplement 1. Endogenous AID tagging of CDPK1 resulted in a decrease in the abundance of CDPK1. cWT and cMut complementation result in similar expression levels to the AID-tagged iKD CDPK1, albeit the cMut complement has marginally higher expression. Since CDPK1 is essential for the lytic cycle, insufficient levels of the cWT expression would have displayed defects in our plaque assays. We have updated our results to reflect this new data:

“Additionally, we compared endogenous CDPK1 expression to mAID-tagged, cWT, and cMut strain (Figure 2–figure supplement 1). Introduction of a mAID tag to CDPK1 led to a reduction in CDPK1 levels, but these levels were equivalent to complementation products in cWT and cMut parasites.”

3. Please attempt to confirm that aerolysin treatment does not impact myristoylation-dependent subcellular partitioning of CDPK1.

The kinase activity in aerolysin-treated parasites was unaffected by the 1B7 inhibitory nanobody, demonstrating that parasites remain impermeable to proteins as small as 15 kDa. Furthermore, we localize CDPK1 by immunofluorescence in aerolysin-treated parasites to show that the localization of CDPK1 is indistinguishable from that of vehicle-treated parasites, suggesting that overall CDPK1 localization is unaffected by aerolysin treatment. We include this data in panel B in Figure 3–figure supplement 1. Nevertheless, in the manuscript we discuss the limitations of the thiophosphorylation experiment:

“While our approach largely maintains kinases in their subcellular context, aerolysin treatment disrupts native ion concentrations and detaches the plasma membrane from the inner membrane complex (IMC) (135).”

Because of these limitations we rely on the overlap of CDPK1-dependent targets between our thiophosphorylation and time course experiments.

4. Please confirm the interaction of TGGT1_306920 and TGGT1_316650 with the HOOK and FTS proteins.

In response to this suggestion, we tagged the C termini of TGGT1_306920 and TGGT1_316650 with 3xHA epitopes. Although immunoprecipitation of TGGT1_316650 was unsuccessful, immunoprecipitation of TGGT1_306920 identified HOOK and FTS as significantly enriched proteins. We include this new data in panel C of Figure 5 and have updated our results:

“To further confirm the interaction, we fused a 3xHA tag to the C terminus of TGGT1_306920, performed IP-MS and compared protein enrichment to the HOOK-3xHA IP (Figure 5C). HOOK, FTS, and TGGT1_306920 were significantly enriched across both IP-MS experiments, whereas TGGT1_316650 is only significantly enriched in HOOK and FTS pulldowns. This suggests the presence of multiple HOOK complexes composed of the core HOOK and FTS proteins that bind with either TGGT1_316650 or TGGT1_306920.”

While further interactions with other members of the complex still need to be validated it is not the standard of the field to validate every member of a protein complex by reciprocal IP. Our HOOK and FTS IP-MS results each identified HOOK, FTS, TGGT1_306920, and TGGT1_316650 and our TGGT1_306920 IP-MS identified all members except TGGT1_316650. These interaction partners were found significantly enriched compared to parental controls, which make the observation of the complex robust.

**Reviewer #1 (Recommendations For The Authors):**
I have only a few minor comments:1. In the supplemental data section I would include a document of code ( R script) used for the analysis. If this is too cumbersome then I would instead suggest that like done with proteomic data, the code should be deposited in a database that provides a DOI for access, instead of only being provided on request. This can be done by use of an electronic laboratory notebook or via Github.com or a similar service.

Zip files containing R code and CSVs have been included for the sub-minute resolution phosphoproteomics (Supplementary File 11) and thiophosphorylation (Supplementary File 12).

2. It would be useful to expand the discussion of the other 2 proteins identified in the HOOK complex TGGT1_316650 and 306920. Do these have homologs to proteins in other organisms? Based on HOOK in other eukaryotes can you provide a model of the 4 proteins in the complex that you identified? Was any work done on 316650 and 306920 with regards to genetic KO or auxin regulation to see if they also provided a similar phenotype to what was described with HOOK and FTS?

We have included the following information in our discussion:

“It also remains unknown how the HOOK complex binds to micronemes. In H. sapiens and *D. melanogaster*, RAB5 on vesicles interacts with FHIP in the HOOK complex (Bielska et al., 2014; Gillingham et al., 2014; Guo et al., 2016; Xu et al., 2008; Yao et al., 2014). We speculate that TGGT1_306920 may serve the role of FHIP within the HOOK complex as it is fitness conferring whereas TGGT1_316650 appears dispensable but the complex's binding partner on micronemes remains unknown. RAB5A and RAB5C have been implicated in the biogenesis of micronemes, but their roles during exocytosis have not been explored (Kremer et al., 2013). Understanding how micronemes are recognized may elucidate how cargo specificity is achieved and regulated.”

TGGT1_306920 is conserved amongst coccidians and shares similar localization to HOOK and FTS. TGGT1_316650 is conserved amongst apicomplexans and more broadly in subsets of other eukaryotic phyla. Given our IP-MS data, HOOK and FTS form a core complex that is either bound to TGGT1_316650 or TGGT1_306920. Given that TGGT1_306920 appears to be important for parasite fitness, based on genome-wide screening data (Sidik, Huet, et al. 2016), we speculate this could function to mediate the linkage to microneme organelles. At this time, we have no additional data to present on 316650 and 306920. Additional biochemical studies will be needed to characterize the stoichiometry of complexes and their function; however, we propose that HOOK and FTS are interacting as previously described in opisthokonts (Bielska et al., 2014, Guo et al., 2016 and Zhang et al., 2014).

3. The myristoylation data section ended with "additional studies will be required to understand how myristoylation influences CDPK1 activity". What studies are required to further this understanding? I assume these studies are difficult and that is why they were not part of this outstanding paper.

The effect of myristoylation is modest during acute phenotypes like egress (see Figure 2H). Moreover there were no significant differences between cWT and cMut that could explain the impact of CDPK1 on microneme secretion, which was the purpose of this study. Further studies would require a phosphoproteomic workup of the cWT and cMut, which is beyond the scope of the present study.

4. In the key resource table, in the first column reagent type I suggest you indicate this as *T. gondii* RH strain to make it clear the background strain (I know it is encoded in additional information but the first column should also be clear).

We have updated the key resources table to indicate the *T. gondii* strains used are of RH background.

**Reviewer #2 (Recommendations For The Authors):**
I have a few minor comments that could be addressed by modification of the current version of the manuscript.Line 290, where authors classify proteins phosphorylated in CDPK1 dependent manner into five groups, it would be helpful to list at least class 1 (five proteins) and class 2 (four proteins) in the text of the results section. Further since in the same paragraph, the authors are also describing figure 3G, it would be helpful if the groups are identified with roman numerals or as class A, B, C, D, and E. Currently, in fig 3G, the three columns (CDPK1 dependent, CDPK1 independent and fitness scores) are also identified as 1, 2 and 3 and these nomenclatures could be confused with the five different classes of putative substrates.

We thank the reviewer for their helpful suggestion. We have renamed the classes of CDPK1 targets using roman numerals I, II, III, IV, and V. We have also listed out the proteins in Class I and Class II in the results section as follows:

“Class I contains five proteins for which the same phosphorylated site was identified in both the time course and thiophosphorylation experiments and include: TGGT1_227610, TGGT1_221470, TGGT1_235160, TGGT1_273560 (KinesinB), and TGGT1_310060. Class II contains four proteins for which phosphorylated sites identified across both approaches were within 50 amino acid residues of one another and include: TGGT1_289100 (MIC18), TGGT1_309190 (AIP), TGGT1_254870, and TGGT1_259630.”

Line 398, the expansions of the abbreviations FTS and FHIP should be included.

We have included the expansions of the abbreviations for FTS and FHIP:

“In *D. melanogaster* and mammals, HOOK proteins have been shown to form dimers and bind Fused Toes (FTS) and FTS and HOOK-interacting protein (FHIP) via a C-terminal region that interacts with vesicular cargo (Christensen et al., 2021; Krämer and Phistry, 1996; Lee et al., 2018; Xu et al., 2008).”

The HOOK protein shows CDPK1-dependent phosphorylation at multiple sites S167, S177, and S189-191. In the discussion section, it would be helpful if the authors can speculate about the importance of these phosphorylated residues on the functioning of HOOK.

Prior to engaging parasite motility, micronemes are positioned at the apical third of the parasite, but after an increase in intracellular Ca_2+_, micronemes rapidly traffic to the apical tip of the parasite. Our results indicate that both CDPK1 kinase activity and HOOK are required for microneme trafficking. Given the association of micronemes with tubulin-based structures such as the cortical microtubules and conoid, activation of trafficking along such structures must be rapid, on the time scale of seconds. Cell-free reconstitution assays generated from opisthokonts indicate that activating adaptors like HOOK are necessary to activate processive dynein trafficking along microtubules in addition to conferring cargo selectivity. In intracellular non-motile parasites, HOOK is expressed and localized to the apical end and cytosol prior to the activation of rapid microneme trafficking, consistent with regulation of HOOK activity. We have included reference to this type of regulation and our expectation that CDPK1 activates the HOOK complex as part of the Discussion:

“Phosphorylation has been reported to regulate the function of activating adaptors. In HeLa cells, phosphorylation of BICD2 facilitates recruitment of dynein and dynactin (Gallisà-Suñé et al. 2023). Analogously, phosphorylation of JIP1 mediates the switch between kinesin and dynein motility of autophagosomes in murine neurons (Fu et al. 2014). We therefore speculate that phosphorylation of HOOK by CDPK1 may activate the adaptor by promoting its interaction with dynein and dynactin to initiate trafficking of micronemes.”

**Reviewer #3 (Recommendations For The Authors):**

1. CDPK1 myristoylation. The loss of myristoylation of CDPK1 appears to increase its interaction with ROM4 which also becomes cytosolic instead of localizing to the plasma membrane. As ROM4 is necessary for microneme discharge after proteolysis it would be interesting to investigate the specific relation between CDPK1 and ROM4 and to confirm the interaction by reciprocal IP.

Please see our response to Suggested Revision #1.

2. CDPK1 myristoylation, Figure 2D. It would be useful to compare the expression of the tagged and complemented (cWT and cMut) CDPK1 with the endogenous expression of the non-tagged and non-complemented gene.

Please see our response to Suggested Revision #2.

3. Thiophosphorylation. The authors used the bacterial toxin aerolysin to semi-permeabilize parasite membranes by forming 3-nm pores. Aerolysin affects the membrane integrity, however, the authors demonstrated that CDPK1 is possibly associated with membrane structures (Figure 2E/F). Could it be possible to transiently destabilize the membrane before to treat with KTPγS or ATP? If not, it would be necessary to confirm that aerolysin treatment does not impact myristoylation-dependent subcellular partitioning of CDPK1 before identifying proteins specifically labelled by CDPK1G and not by CDPK1M (Figure 3C).

Please see our response to Essential Revision #3.

4. IP-MS on HOOK-3xHA parasites. The authors' results suggest that HOOK and FTS form a functional complex implicated in microneme exocytosis. It would be interesting to know if HOOK knockdown can have an effect on FTS expression or localization and reciprocally.

While we agree with the reviewer that this is an interesting question, we focused this paper on the discovery of the complex in relation to CDPK1. Understanding the regulation and interaction of the complex components is the focus of ongoing work and will require generation of new strains and additional mass spectrometry. For those reasons we find these experiments fall beyond the scope of the present study.

5. FTS-Turbo-ID. (Line 443-444) Authors should confirm the interaction of TGGT1_306920 and TGGT1_316650 with the HOOK and FTS proteins, it will give strength to their conclusion. In fact, without confirmation, everything is based on suggestions that were also formulated but not confirmed in humans. The physical existence of this putative complex should be demonstrated by co-IP experiments. In addition, the missing player is a dynein candidate itself, which leaves the model vulnerable. Short of pursuing this experimentally, it should at least be commented on in the Discussion.

Please see our response to Sugegsted Revision #4. Our IP-MS experiments of HOOK-3xHA and FTS-3xHA indicate interactions with HOOK, FTS, TGGT1_316650, and TGGT1_306920. Our FTS-TurboID experiments also suggest an interaction between FTS, HOOK, TGGT1_316650 and TGGT1_306920. Furthermore, our TGGT1_306920 IP-MS data identifies HOOK and FTS, but not TGGT1_316650, suggesting distinct complexes with HOOK and FTS as core components.

6. MIC2 secretion (Fig 5J). The rep represented by the grey dot with a white outline seems like an outlier result compared to the other 2 reps. Basically, without this rep there at least is a strong trend that there is a difference in secretion without EtOH stimulation. That is what actually would be expected, for constitutive secretion! Please carefully reconsider these data - e.g. check for outlier statistics and/or add reps.

We present three independent biological replicates, showing a significant difference in microneme secretion following depletion of CDPK1, HOOK, or FTS. It is expected, based on our prior experience, that microneme secretion will vary day to day. However, the expected trend can be observed in all replicates. We are unclear what the reviewer means by constitutive secretion since some low-level of calcium-dependent microneme discharge is expected even in the absence of stimulation, barring BAPTA-AM treatment. Even in the absence of EtOH stimulation (left graph in Fig. 5J), the trend of diminished basal MIC2 release holds when CDPK1, HOOK, or FTS is knocked down.

7. Apical accumulation of micronemes. A similar observation was made upon manipulation of Ferlin1, which is a manuscript on BioRXivs. Since other BioRXiv manuscripts are cited in the presented work, this is an omission.

We apologize for this omission and have updated the manuscript accordingly:

“It therefore appears that the initial round of microneme discharge during egress depends on CDPK1, and only subsequent rounds require the HOOK complex. Indeed, a fraction of micronemes are already found docked at the apical complex prior to the transition from the replicative to the motile stages, and may constitute the first round of microneme exocytosis (Mageswaran et al., 2021; Sun et al., 2022). Ferlin 1 (FER1) was recently shown to be involved in microneme positioning and overexpression of FER1 was sufficient to initiate an initial round of microneme exocytosis and induce egress (Tagoe et al. 2020).”

Minor comments:1. Concerning the expression of the HOOK protein in Figures 4B, and C, could the author indicate why they performed the IFA after 24h of auxin treatment and the WB after 40h of treatment?

The difference in timing was for technical reasons. Our immunoblots and additional assays such as microneme secretion require more parasites, such that we harvest at the end of the lytic cycle to increase yields. For the IFAs, we perform these at 24 hrs, which allows for depletion and replication, but captures parasites in small vacuoles that show clear localization patterns. Furthermore, our microneme relocalization studies in Figure 6 were also performed after 24 hrs of auxin treatment, yet exhibit a trafficking defect following 24 hr HOOK depletion.

2. Fig 4H. The color of CDPK1-AID on the left and the HA on the top (HOOK) do correspond but indicate different proteins. Please label HOOK text in teal, not CDPK1.

We have changed the text color of the strain names on 4H to black to avoid confusion with the IFA channel labels.

3. I would like to suggest adding the "Key resources tables" in the supplementary data because it makes the materials & methods harder to read.

The key resources table was included at the beginning of the Materials and Methods section as indicated in eLife’s instructions to the authors.